# QUASI-TAYLOR SAMPLERS FOR DIFFUSION GENERATIVE MODELS BASED ON IDEAL DERIVATIVES

## ABSTRACT

Diffusion generative models have emerged as a new challenger to popular deep neural generative models such as GANs, but have the drawback that they often require a huge number of neural function evaluations (NFEs) during synthesis unless some sophisticated sampling strategies are employed. This paper proposes new efficient samplers based on the numerical schemes derived by the familiar Taylor expansion, which directly solves the ODE/SDE of interest. In general, it is not easy to compute the derivatives that are required in higher-order Taylor schemes, but in the case of diffusion models, this difficulty is alleviated by the trick that the authors call "ideal derivative substitution," in which the higher-order derivatives are replaced by tractable ones. To derive ideal derivatives, the authors argue the "single point approximation," in which the true score function is approximated by a conditional one, holds in many cases, and considered the derivatives of this approximation. Applying thus obtained new quasi-Taylor samplers to image generation tasks, the authors experimentally confirmed that the proposed samplers could synthesize plausible images in small number of NFEs, and that the performance was better or at the same level as DDIM and Runge-Kutta methods. The paper also argues the relevance of the proposed samplers to the existing ones mentioned above.

## 1 INTRODUCTION

Generative modeling based on deep neural networks is an important research subject for both fundamental and applied purposes, and has been a major trend in machine learning studies for several years. To date, various types of neural generative models have been studied including GANs (Goodfellow et al., 2014), VAEs (Kingma et al., 2021; Kingma & Welling, 2019), normalizing flows (Rezende & Mohamed, 2015), and autoregressive models (van den Oord et al., 2016b;a). In addition to these popular models, a class of novel generative models based on the idea of iteratively refinement using the diffusion process has been rapidly gaining attention recently as a challenger that rivals the classics above (Sohl-Dickstein et al., 2015; Song & Ermon, 2019; Song et al., 2020b; Song & Ermon, 2020; Ho et al., 2020; Dhariwal & Nichol, 2021). The diffusion-based generative models have recently been showing impressive results in many fields including image (Ho et al., 2020; Vahdat et al., 2021; Saharia et al., 2021; Ho et al., 2021; Sasaki et al., 2021), video (Ho et al., 2022), text-to-image (Nichol et al., 2021; Ramesh et al., 2022), speech (Chen et al., 2020; 2021; Kong et al., 2021; Popov et al., 2021; Kameoka et al., 2020), symbolic music (Mittal et al., 2021), natural language (Hoogeboom et al., 2021; Austin et al., 2021), chemoinformatics (Xu et al., 2022), etc.

However, while the diffusion models have good synthesis quality, it has been said that they have a fatal drawback that they often require a very large number of iterations (refinement steps) during synthesis, ranging from hundreds to a thousand. In particular, the increase in refinement steps critically reduces the synthesis speed, as each step involves at least one neural function evaluation (NFE). Therefore, it has been a common research question how to establish a systematic method to stably generate good data from diffusion models in a relatively small number of refinement steps, or NFEs in particular. From this motivation, there have already been some studies aiming at reducing the NFEs (See § 2). Among these, Probability Flow ODE (PF-ODE) (Song et al., 2020b) enable efficient and deterministic sampling, and is gaining attention. This framework has the merit of deriving a simple ODE by a straightforward conceptual manipulation of diffusion process. However, the ODE is eventually solved by using a black-box Runge-Kutta solver in the original paper, which requires several NFEs per step and is clearly costly. Another PF-ODE solver includes DDIM (Song et al., 2020a), and is also

commonly used. It is certainly efficient and can generate plausible images. However, it was not originally formulated as a PF-ODE solver, and the relationship between DDIM and PF-ODE is not straightforward.

From these motivations, we provide another sampler to solve the same ODE, which performs better than or on par with DDIM. The derivation outline is simple and intuitive: (1) consider the Taylor expansion of the given system, and (2) replace the derivatives in the Taylor series with appropriate functions; that's all.

The contribution of this paper would be as follows: (1) We propose novel samplers for diffusion models based on Taylor expansion of PF-ODE. They outperformed, or were on par with Runge-Kutta methods. (2) To derive our algorithms, we show that the derivatives of score function can be approximated by simple functions. We call this technique the *ideal derivative substitution*. (3) It has been known that the 1st order term of DDIM is same as the Euler method for PF-ODE. This paper gives further explanation for higher order terms of DDIM: we show that the proposed Quasi-Taylor method and DDIM are identical at least up to 3rd order terms. (4) The same idea can be naturally extended to derive a stochastic solver for a reverse-time SDE, which we call R-SDE in this paper.

## 2 BACKGROUND AND RELATED WORK

**Diffusion Process to draw a new data from a target density:** Let us first briefly summarize the framework of the diffusion-based generative models. Following Song et al. (2020b), we describe the mechanisms using the language of continuous-time diffusion process for later convenience. Let us consider "particles" $\{\mathbf{x}_t\}$ moving in a $d$-dim space obeying the following Itô diffusion,

$$\text{SDE:} \quad d\mathbf{x}_t = \mathbf{f}(\mathbf{x}_t, t)dt + g(\mathbf{x}_t, t)d\mathbf{B}_t, \tag{1}$$

where $\mathbf{B}_t$ is the $d$-dim Brownian motion whose temporal increments obeys the standard Gaussian. The drift $\mathbf{f}(\cdot, \cdot)$ is $d$-dim vector, and the diffusion coefficient $g(\cdot, \cdot)$ is scalar. The SDE describes the microscopic dynamics of each particle. On the other hand, the "population" of the particles obeying the above SDE, i.e. density function $p(\mathbf{x}_t, t \mid \mathbf{x}_s, s), (t > s)$, follows the following PDEs, which are known as Kolmogorov's forward and backward equations (KFE and KBE); the former is also known as the Fokker-Planck equation (FPE), see § E.2,

$$\text{FPE:} \qquad \partial_t p(\mathbf{x}_t, t \mid \mathbf{x}_s, s) = -\nabla_{\mathbf{x}_t} \cdot \mathbf{f}(\mathbf{x}_t, t)p(\mathbf{x}_t, t \mid \mathbf{x}_s, s) + \Delta_{\mathbf{x}_t} \frac{g(\mathbf{x}_t, t)^2}{2} p(\mathbf{x}_t, t \mid \mathbf{x}_s, s), \tag{2}$$

$$\text{KBE:} \quad -\partial_s p(\mathbf{x}_t, t \mid \mathbf{x}_s, s) = \mathbf{f}(\mathbf{x}_s, s) \cdot \nabla_{\mathbf{x}_s} p(\mathbf{x}_t, t \mid \mathbf{x}_s, s) + \frac{g(\mathbf{x}_s, s)^2}{2} \Delta_{\mathbf{x}_s} p(\mathbf{x}_t, t \mid \mathbf{x}_s, s), \tag{3}$$

where $\Delta_{\mathbf{x}} := \nabla_{\mathbf{x}} \cdot \nabla_{\mathbf{x}}$ is Laplacian. (FPE also holds for $p(\mathbf{x}_t, t)$; consider the expectation $\mathbb{E}_{p(\mathbf{x}_s, s)}[\cdot]$.) These PDEs enables us to understand the macroscopic behavior of the particle ensemble. For example, if $\mathbf{f}(\mathbf{x}, t) = -\nabla U(\mathbf{x}), g(\mathbf{x}, t) = \sqrt{2D}$, where $U(\mathbf{x})$ a certain potential and $D$ a constant, then we may verify that the stationary solution of FPE is $p(\mathbf{x}) \propto e^{-U(\mathbf{x})/D}$. It means that we may draw a sample $\mathbf{x}$ that follows the stationary density by evolving the SDE over time. This technique is often referred to as the Langevin Monte Carlo method (Rossky et al., 1978; Roberts & Tweedie, 1996). Some of the diffusion generative models are based on this framework, e.g. (Song & Ermon, 2019; 2020), in which the potential gradient $\nabla U(\mathbf{x})$ is approximated by a neural network.

Another systematic approach is considering the reverse-time dynamics (Song et al., 2020b). An approach is based on KBE eq. (3). Roughly speaking, FPE gives information about the future from the initial density, while KBE gives information about what the past states were likely to be from the terminal density. Here, instead of using KBE directly, it is useful to consider a variant of it which is transformed into the form of FPE, because it has an associated SDE that enables the particle-wise backward sampling (Stratonovich, 1965; Anderson, 1982); see also § E.3.2,

$$\text{R-FPE:} \quad -\partial_s p(\mathbf{x}_s, s \mid \mathbf{x}_t, t) = \nabla_{\mathbf{x}_s} \cdot \bar{\mathbf{f}}(\mathbf{x}_s, s)p(\mathbf{x}_s, s \mid \mathbf{x}_t, t) + \Delta_{\mathbf{x}_s} \frac{\bar{g}(\mathbf{x}_s, s)^2}{2} p(\mathbf{x}_s, s \mid \mathbf{x}_t, t) \tag{4}$$

$$\text{R-SDE:} \qquad\qquad d\mathbf{x}_s = -\bar{\mathbf{f}}(\mathbf{x}_s, s)(-ds) + \bar{g}(\mathbf{x}_s, s)d\bar{\mathbf{B}}_s. \tag{5}$$

Hereafter, let $g(x_t, t) = g(t)$ for simplicity. Then the specific forms of drift and diffusion coefficients are written as follows,

$$\text{R-SDE coeffs:} \quad \bar{\mathbf{f}}(\mathbf{x}_t, t) = \bar{\mathbf{f}}_\sharp(\mathbf{x}_t, t) := \mathbf{f}(\mathbf{x}_t, t) - g(t)^2 \nabla_{\mathbf{x}_t} \log p(\mathbf{x}_t, t), \quad \bar{g}(t) = g(t). \tag{6}$$

Starting from a certain random variable $\mathbf{x}_T$, then by evolving the R-SDE reverse in time, we may obtain a $\hat{\mathbf{x}}_0$ which follows $p(\mathbf{x}_0, 0 \mid \mathbf{x}_T, T)$ (i.e. the solution of R-FPE eq. (4)). Therefore, if the initial density $p(\mathbf{x}_0, 0)$ of the forward dynamics eq. (2) is the true density, then we may utilize this mechanism as a generative model to draw a new sample $\hat{\mathbf{x}}_0$ from it.

Another approach is based on FPE eq. (2). By formally eliminating the diffusion term of the FPE for the forward process, we can derive another backward FPE (see also § E.3.1). Being diffusion-free, the backward FPE yields a deterministic ODE, which is called the Probability Flow ODE (PF-ODE) (Song et al., 2020b), and is an example of neural ODEs (Chen et al., 2018). The population density obtained by evolving this system is exactly the same as the above R-SDE.

$$\text{PF-ODE coeffs:} \quad \bar{\mathbf{f}}(\mathbf{x}_t, t) = \bar{\mathbf{f}}_\flat(\mathbf{x}_t, t) := \mathbf{f}(\mathbf{x}_t, t) - \frac{1}{2}g(t)^2 \nabla_{\mathbf{x}_t} \log p(\mathbf{x}_t, t). \quad \bar{g}(t) = 0. \quad (7)$$

Some extensions of this framework include as follows. Dockhorn et al. (2021) introduced the velocity variable considering the Hamiltonian dynamics. Another extension is the introduction of a conditioning parameter, and guidance techniques using it (Dhariwal & Nichol, 2021; Ho & Salimans, 2021; Choi et al., 2021) to promote the dynamics to go to a specific class of images, which has achieved remarkable results in text-to-image tasks (Nichol et al., 2021; Ramesh et al., 2022).

**Variance-Preserving Model (VP-SDE Model):** The solution of unconditioned FPE is written as the convolution with the initial density $p(\mathbf{x}_0, 0)$ and the *fundamental solution*, or the *heat kernel*, $p(\mathbf{x}_t, t \mid \mathbf{x}_0, 0)$, which is the solution of the conditional FPE under the assumption that the initial density was delta function, $p(\mathbf{x}_0, 0) = \delta(\mathbf{x}_0 - \mathbf{x}_0^*)$. Although it is still intractable to solve this problem in general, a well-known exception is the (time-dependent) Ornstein-Uhlenbeck (OU) process where $\mathbf{f}(\mathbf{x}_t, t) = -\frac{1}{2}\beta_t \mathbf{x}_t$ and $g(\mathbf{x}_t, t) = \sqrt{\beta_t}$. $\beta_t = \beta(t)$ is a non-negative continuous function. The specific form of diffusion coefficient $\beta_t$ has some options: a simplest one would be the linear function, and another would be the cosine schedule proposed in (Nichol & Dhariwal, 2021); see also § D. In any cases, if it is the OU process, the heat kernel is simply written as follows,

$$p(\mathbf{x}_t, t \mid \mathbf{x}_0, 0) = \mathcal{N}(\mathbf{x}_t \mid \sqrt{1 - \sigma_t^2}\mathbf{x}_0, \sigma_t^2 \mathbf{I}), \quad \text{where} \ \ \sigma_t^2 = 1 - \exp\left(-\int_0^t \beta_{t'} dt'\right). \quad (8)$$

Hereafter, we denote the noise variance by $\nu_t := \sigma_t^2$. (In some literature, the signal level $\alpha_t := \sqrt{1 - \sigma_t^2}$ is used as a basic parameter instead of the variance.) This model is referred to as the variance-preserving (VP) model by Song et al. (2020b). It has good properties such as the scale of data $\|\mathbf{x}_t\|_2$ is almost homogeneous, which is advantageous in neural models. However, the variance exploding (VE) model (Song et al., 2020b) in which the norm increases is also practicable, and the theory can be developed in a similar manner.

**Training Objective:** In diffusion-based generative models, one estimates the *score function* $\nabla_{\mathbf{x}_t} \log p(\mathbf{x}_t, t) = \nabla_{\mathbf{x}_t} \log \mathbb{E}_{p(\mathbf{x}_0, 0)}[p(\mathbf{x}_t, t \mid \mathbf{x}_0, 0)]$ by a neural network $\mathbf{S}_\theta(\mathbf{x}_t, t)$. This sort of learning has been referred to as the score matching (Hyvärinen & Dayan, 2005; Vincent, 2011). However, the exact evaluation of this training target is clearly intractable because of the expectation $\mathbb{E}_{p(\mathbf{x}_0, 0)}[\cdot]$, so it has been common to consider a Variational Bayesian surrogate loss; Ho & Salimans (2021) showed that the following loss function approximates the negative ELBO,

$$\mathcal{L} := \mathbb{E}[\| -\sqrt{\nu_t} \nabla_{\mathbf{x}_t} \log p(\mathbf{x}_t, t \mid \mathbf{x}_0, 0) - \mathbf{S}_\theta(\mathbf{x}_t, t)\|_2^2] = \mathbb{E}[\| \tfrac{\mathbf{x}_t - \sqrt{1-\nu_t}\mathbf{x}_0}{\sqrt{\nu_t}} - \mathbf{S}_\theta(\mathbf{x}_t, t)\|_2^2] \quad (9)$$

$$= \mathbb{E}[\|\mathbf{w} - \mathbf{S}_\theta(\sqrt{1-\nu_t}\mathbf{x}_0 + \sqrt{\nu_t}\mathbf{w}, t)\|_2^2], \quad (10)$$

where the expectation in eq. (10) is taken w.r.t. $\mathbf{x}_0 \sim \mathcal{D}$, $\mathbf{w} \sim \mathcal{N}(\mathbf{0}, \mathbf{I})$, and $t \sim \text{Uniform}([0, T])$. Some variants of the score matching objectives are also studied. For example, Chen et al. (2020) reported that the $L_1$ loss gave better results than the $L_2$ loss in speech synthesis. Also, Kingma et al. (2021) argued that the weighted loss with SNR-based weights improves the performance.

It should be noted that the above loss function will actually be very close to the ideal score matching loss function in practice, where the probability is not conditioned on $\mathbf{x}_0$, i.e.,

$$\mathcal{L}_{\text{ideal}} = \mathbb{E}[\| -\sqrt{\nu_t} \nabla_{\mathbf{x}_t} \log p(\mathbf{x}_t, t) - \mathbf{S}_\theta(\mathbf{x}_t, t)\|_2^2]. \quad (11)$$

This is because there almost always exists a point $\mathbf{x}_0$ on the data manifold such that $\nabla_{\mathbf{x}_t} \log p(\mathbf{x}_t, t) \approx \nabla_{\mathbf{x}_t} \log p(\mathbf{x}_t, t \mid \mathbf{x}_0)$ holds with very high accuracy in very high-dim cases, because of the well-known "log-sum-exp $\approx$ max" law. For more details, see § 3.3 and § A.

**Sampling Schemes for R-SDE and PF-ODE:** Thus obtained $\mathbf{S}_\theta(\mathbf{x}_t, t)$ is expected to finely approximate $-\sqrt{\nu_t}\nabla_{\mathbf{x}_t}\log p(\mathbf{x}_t, t)$, and we may use it in eq. (5). One of the simplest numerical schemes for solving SDEs is the Euler-Maruyama method (Maruyama, 1955, Theorem. 1) as follows, and many diffusion generative models are actually using it.

$$\text{Euler-Maruyama:} \quad \mathbf{x}_{t-h} \leftarrow \mathbf{x}_t - h\bar{\mathbf{f}}_\sharp(\mathbf{x}_t, t) + \sqrt{h}g(t)\mathbf{w}, \quad \text{where } \mathbf{w} \sim \mathcal{N}(\mathbf{0}, \mathbf{I}) \quad (12)$$

where $h > 0$ is the step size. The error of the Euler-Maruyama method is the order of $O(\sqrt{h})$ in general, though it is actually $O(h)$ in our case; this is because $\nabla_{\mathbf{x}_t}g(t) = 0$. As a better solver for the R-SDE, the Predictor-Corrector (PC)-based sampler was proposed in (Song et al., 2020b). The PC sampler outperformed the Predictor-only strategy, but it requires many NFEs in the correction process, so we will exclude it in our discussion. Another R-SDE solver is the one proposed by Jolicoeur-Martineau et al. (2021), whose NFE per refinement step is 2.

On the other hand, there are also deterministic samplers for PF-ODE eqs. (5), (7) as follows,

$$\text{Euler:} \quad \mathbf{x}_{t-h} \leftarrow \mathbf{x}_t - h\bar{\mathbf{f}}_\flat(\mathbf{x}_t, t) \quad (13)$$

$$\text{Runge-Kutta:} \quad \mathbf{x}_{t-h} \leftarrow \mathbf{x}_t - h\sum_{i=1}^{m} b_i\mathbf{k}_i, \quad \text{where } \mathbf{k}_i = \bar{\mathbf{f}}_\flat(\mathbf{x}_t - h\sum_{j=1}^{i-1} a_{ij}\mathbf{k}_j, t - hc_i) \quad (14)$$

where $\{a_{ij}\}, \{b_i\}, \{c_i\}$ are coefficients of the Runge-Kutta (RK) method (see § E.5). The error of the Euler method is $O(h)$, and that of the RK method is $O(h^p), p \leq m$ in general (Press et al., 2007, § 16). Another deterministic sampler is DDIM (Song et al., 2020a, Eq. (13)), and is also understood as a PF-ODE solver (Salimans & Ho, 2022). Its NFE per step is only 1, and is capable of efficiently generate samples.

$$\text{DDIM:} \quad \mathbf{x}_{t-h} \leftarrow \frac{\alpha_{t-h}}{\alpha_t}\mathbf{x}_t + \left(\sigma_{t-h} - \frac{\alpha_{t-h}}{\alpha_t}\sigma_t\right)\mathbf{S}_\theta(\mathbf{x}_t, t). \quad (15)$$

In addition, as a concurrent work as ours, Lu et al. (2022) proposed the DPM-solver, which is based on the Taylor expansion of PF-ODE. However, as the gradient is evaluated using several different points, the NFE per step is greater than 1 in general. Liu et al. (2022) proposed a sampler based on the linear multi-step method, in which the NFE/step is reduced to 1 except initial 3 steps. Another PF-ODE solver is the DEIS (Zhang & Chen, 2022) which is based on the exponential integrator with some non-trivial approximations such as the polynomial interpolation of score function.

Other techniques that aimed to make sampling faster include as follows. Song & Ermon (2020) proposed a variety of techniques to accelerate the sampling. Watson et al. (2021) proposed a DP-based optimization method to tune noise schedules for faster sampling. Luhman & Luhman (2021) and Salimans & Ho (2022) proposed distilling the pretrained teacher model to a student model that can predict teacher's several steps in a single step, which is efficient during the sampling but extra training for distillation is required. Bao et al. (2022a;b) derived some analytic expressions of reverse dynamics to enable faster sampling.

## 3 PROPOSED METHOD: QUASI-TAYLOR SAMPLERS

### 3.1 MOTIVATION: HIGHER-ORDER STRAIGHTFORWARD SOLVERS FOR R-SDE AND PF-ODE

As mentioned above, DDIM already exists as an efficient solver for PF-ODE, but it can only be considered a PF-ODE solver up to first-order terms (Song et al., 2020a; Salimans & Ho, 2022), and it would not be clear enough whether it can be considered a higher-order solver for PF-ODE. Some other techniques (Lu et al., 2022; Liu et al., 2022; Zhang & Chen, 2022) were designed as higher-order PF-ODE solvers, though their derivations are rather sophisticated and less simple. Since PF-ODE and R-SDE provide the basis for the diffusion generative models, it would be beneficial to develop samplers that directly solve them through intuitive and straightforward arguments.

From these motivations, we propose a simple but efficient sampler based on the Taylor expansion, a very basic technique that is familiar to many researchers and practitioners. In general, Taylor methods are not very popular as numerical schemes because they require higher-order derivatives, which are not always tractable. However, in diffusion models, the derivatives are easily and effectively evaluated, albeit approximately. The validity of this approximation requires some consideration (see § A, § B), but once accepted, an efficient sampler can be derived simply by substituting this approximation formula into the Taylor series. This section describes the details of the idea, and derives solvers for both PF-ODE and R-SDE. Entire sampling procedures are summarized in § F.

## 3.2 Taylor Scheme for ODE and Itô-Taylor Scheme for SDE

**Taylor Scheme for Deterministic Systems** For simplicity, we consider the 1-dim case here, but we can easily generalized it to multidimensional cases. (See § E.1.1.) Given a ODE $\dot{x}_t = a(x_t, t)$, where the function $a$ is sufficiently smooth, then we can consider the Taylor expansion of it, using a differential operator $L_\flat := \left(\partial_t + a(t, x_t)\partial_{x_t}\right)$. We can write the Taylor expansion of the path $x_t$ as follows. Ignoring $o(h^p)$ terms of the series, we obtain a numerical scheme of order $p$.

$$x_{t+h} = x_t + ha(x_t, t) + \frac{h^2}{2!}L_\flat a(x_t, t) + \frac{h^3}{3!}L_\flat^2 a(x_t, t) + \cdots . \tag{16}$$

**Itô-Taylor Scheme for Stochastic Systems** In stochastic systems, the Taylor expansion requires modifications because of the relation $\mathbb{E}[dB_t^2] = dt$. If $x_t$ obeys a stochastic system $dx_t = a(x_t, t)dt + b(x_t, t)dB_t$, then the path is written in a stochastic version of Taylor-like series, which is often called the Itô-Taylor expansion, a.k.a. Wagner-Platen expansion (Platen & Wagner, 1982);(Kloeden et al., 1994, § 2.3.B);(Särkkä & Solin, 2019, § 8.2). The Itô-Taylor expansion is based on the following differential operators $L_\sharp, G_\sharp$, which are based on Itô's formula (Itô, 1944).

$$L_\sharp := \partial_t + a(x, t)\partial_x + \frac{1}{2}b(x, t)^2\partial_x^2, \quad G_\sharp := b(x, t)\partial_x \tag{17}$$

In (Kloeden & Platen, 1992), a number of higher order numerical schemes for SDEs based on the Itô-Taylor expansion are presented. One of the simplest of them is as follows. See also § E.1.2.

**Theorem 1** (Kloeden & Platen (1992, § 14.2): An Itô-Taylor scheme of weak order $\beta = 2$). *Let $x_t$ obeys the above SDE, and let the differential operators $L_\sharp, G_\sharp$ be given by eq. (17). Then, the following numerical scheme weakly converges with the order of $\beta = 2$ (see § E.4). Furthermore, in a special case where $G_\sharp^2 b \equiv 0$, the strong $\gamma = 1.5$ convergence is also guaranteed (Kloeden & Platen, 1992, § 10.4).*

$$x_{t+h} \leftarrow x_t + ha + \tilde{w}_t b + \frac{\tilde{w}_t^2 - h}{2}G_\sharp b + \frac{h^2}{2}L_\sharp a + (\tilde{w}_t h - \tilde{z}_t)L_\sharp b + \tilde{z}_t G_\sharp a \tag{18}$$

*where $\tilde{w}_t = \sqrt{h}w_t, \tilde{z}_t = h\sqrt{h}z_t$ are correlated Gaussian random variables, and $w_t, z_t$ are given by $w_t = u_1$ and $z_t = \frac{1}{2}u_1 + \frac{1}{2\sqrt{3}}u_2$, where $u_1, u_2 \sim \mathcal{N}(0, 1)$ (i.i.d.). The notations $a, L_\sharp a$, etc. are the abbreviations for $a(x_t, t), (L_\sharp a)(x_t, t)$, etc.*

## 3.3 Single Point Approximation of the Score Function

Before proceeding, let us introduce the single point approximation of score function that $\nabla_{\mathbf{x}_t} \log p(\mathbf{x}_t, t)$ almost certainly has a some point $\mathbf{x}_0$ on the data manifold such that the following approximation holds,

$$\nabla_{\mathbf{x}_t} \log p(\mathbf{x}_t, t) = \nabla_{\mathbf{x}_t} \log \int p(\mathbf{x}_t, t \mid \mathbf{x}_0, 0)p(\mathbf{x}_0, 0)d\mathbf{x}_0 \approx \nabla_{\mathbf{x}_t} \log p(\mathbf{x}_t, t \mid \mathbf{x}_0, 0). \tag{19}$$

To date, this approximation has often been understood as a tractable variational surrogate. However, the error between the integral and the single point approximation is actually very small in practical scenarios. More specifically, the following facts can be shown under some assumptions.

1. The relative $L_2$ distance between $\nabla_{\mathbf{x}_t} \log p(\mathbf{x}_t, t)$ and $\nabla_{\mathbf{x}_t} \log p(\mathbf{x}_t, t \mid \mathbf{x}_0, 0)$ is bounded above by $\sqrt{(1 - \nu_t)/\nu_t}$ for any point $\mathbf{x}_0$ on the "data manifold" in practical scenarios.

2. When the noise level is low $\nu_t \approx 0$, and the data space is sufficiently high-dimensional, the distant points far from $\mathbf{x}_t$ do not contribute to the integral. If the data manifold is locally a $k$-dim subspace of the entire $d$-dim data space, where $1 \ll k \ll d$, then the relative $L_2$ distance is bounded above by around $2\sqrt{k/d}$.

Of course, the single point approximation is not always valid. In fact, the approximation tends to break down when the noise level $\nu_t$ is around 0.9 (SNR $= (1 - \nu_t)/\nu_t$ is around 0.1). In this region, the single point approximation can deviates from the true gradient by about 20% in some cases. Conversely, however, it would be also said that the error is as small as this level even in the worst empirical cases. For more details on this approximation, see § A.

### 3.4 IDEAL DERIVATIVE SUBSTITUTION

In order to adopt the above Taylor schemes to our problem setting where the base SDE is eq. (5), and $\bar{\mathbf{f}}_\sharp, \bar{\mathbf{f}}_\flat$ are given by eqs. (6), (7), we need to consider the following differential operators. Note that the time evolves backward in time in our case, the temporal derivative should be $-\partial_t$,

$$L_\flat = -\partial_t - \left(\bar{\mathbf{f}}_\flat(\mathbf{x}_t, t) \cdot \nabla_{\mathbf{x}_t}\right), \quad L_\sharp = -\partial_t - \left(\bar{\mathbf{f}}_\sharp(\mathbf{x}_t, t) \cdot \nabla_{\mathbf{x}_t}\right) + \frac{\beta_t}{2}\Delta_{\mathbf{x}_t}, \quad G_\sharp = \sqrt{\beta_t}\left(\mathbf{1} \cdot \nabla_{\mathbf{x}_t}\right),$$

$$\text{where } \bar{\mathbf{f}}_\flat(\mathbf{x}_t, t) = -\frac{\beta_t}{2}\mathbf{x}_t + \frac{\beta_t}{2\sqrt{\nu_t}}\mathbf{S}_\theta(\mathbf{x}_t, t), \quad \bar{\mathbf{f}}_\sharp(\mathbf{x}_t, t) = -\frac{\beta_t}{2}\mathbf{x}_t + \frac{\beta_t}{\sqrt{\nu_t}}\mathbf{S}_\theta(\mathbf{x}_t, t). \quad (20)$$

It is not easy in general to evaluate expressions involving such many derivatives. Indeed, for example, $L_\flat(-\bar{\mathbf{f}}_\flat)$ has the derivatives of the learned score function, viz. $\partial_t \mathbf{S}_\theta(\mathbf{x}_t, t)$ and $(\bullet \cdot \nabla_{\mathbf{x}_t})\mathbf{S}_\theta(\mathbf{x}_t, t)$, which are costly to evaluate exactly, whether in approaches based on finite differences (as in (Lu et al., 2022)), back-propagation, or the JAX paradigm (Bradbury et al., 2018), because they eventually require extra evaluation of a deeply nested function other than $\mathbf{S}_\theta(\mathbf{x}_t, t)$, and extra memory consumption. Fortunately, however, by using the trick which the authors call the "*ideal derivative substitution*", we may write all of the derivatives as a simple combination of known values, only consisting of $\mathbf{x}_t, \mathbf{S}_\theta(\mathbf{x}_t, t), \nu_t, \beta_t$ and derivatives of $\beta_t$, and no extra computation is needed. Since the score function has a single point approximation eq. (19) we may assume that the derivatives should ideally hold following equalities. For derivation, see § B.1.

**Conjecture 1** (Ideal Derivatives). *Under assumptions in § A — i.e. the data space $\mathbb{R}^d$ is sufficiently high dimensional $d \gg 1$, the data manifold $\mathcal{M} \subset \mathbb{R}^d$ is also sufficiently high dimensional but much smaller than the entire space $(1 \ll \dim \mathcal{M} \ll d)$, $\mathcal{M}$ is bounded, $\mathcal{M}$ is sufficiently smooth locally, and the variance parameter $\nu_t$ is close to 0 or 1; — then it is likely that the following approximations hold, where $\mathbf{a} \in \mathbb{R}^d$ is an arbitrary vector. We call them the "ideal derivatives".*

$$(\mathbf{a} \cdot \nabla_{\mathbf{x}_t})\mathbf{S}_\theta(\mathbf{x}_t, t) = \frac{1}{\sqrt{\nu_t}}\mathbf{a}, \quad -\partial_t \mathbf{S}_\theta(\mathbf{x}_t, t) = -\frac{\beta_t}{2\sqrt{\nu_t}}\left(\mathbf{x}_t - \frac{\mathbf{S}_\theta(\mathbf{x}_t, t)}{\sqrt{\nu_t}}\right). \quad (21)$$

To confirm the accuracy of this approximation, we compared empirical and ideal derivatives using MNIST (LeCun et al., 2010) and CIFAR10 (Krizhevsky, 2009). As a result, it was confirmed that the approximation of spatial derivative, i.e. $(\mathbf{a} \cdot \nabla)$, is usually very accurate; the cosine similarity between the empirical and ideal derivatives is nearly always $> 0.99$ (Figure 10). On the other hand, for the time derivative $\partial_t$, it was confirmed that it is quite accurate when the time parameter $t$ (and the variance $\nu_t$) are small, but the error increases when the time parameter $t$ (and the variance $\nu_t$) become larger (Figure 9). See § B.2 for more details.

### 3.5 QUASI-TAYLOR AND QUASI-ITÔ-TAYLOR SCHEMES WITH IDEAL DERIVATIVES

As we can see in § B.2, the ideal derivative approximation is sometimes very accurate while sometimes not. In any case, however, the error in the ideal derivative only affects the second or higher order terms of Taylor series, and it will not be the dominant error in the whole. As there is an overall correlation between the true and ideal derivatives, the advantages will outweigh the disadvantages on average, and we can regularly use this approximation on a speculative basis, even though there exist some cases where the approximation is not accurate.

If we accept the ideal derivative approximation, we can formally compute the symbolic expressions for the derivatives $L_\flat(-\bar{\mathbf{f}}_\flat), L_\sharp(-\bar{\mathbf{f}}_\sharp), L_\sharp(g), G_\sharp(-\bar{\mathbf{f}}_\sharp)$ and $G_\sharp(g)$ that appear in the Taylor and Itô-Taylor series by routine calculations, which can be easily automated by computer algebra systems such as SymPy (Meurer et al., 2017) as shown in § B.3. By substituting thus obtained symbolic expressions into the above Taylor series, we can derive Taylor schemes for both PF-ODE and R-SDE as follows.

**Algorithm 1** (Quasi-Taylor Sampler with Ideal Derivatives for PF-ODE). *Starting from a Gaussian noise $\mathbf{x}_T \sim \mathcal{N}(\mathbf{0}, \mathbf{I})$, iterate the following refinement steps until $\mathbf{x}_0$ is obtained.*

$$\mathbf{x}_{t-h} = \rho_{t,h}^\flat \mathbf{x}_t + \mu_{t,h}^\flat \mathbf{S}_\theta(\mathbf{x}_t, t)/\sqrt{\nu_t}, \text{where} \quad (22)$$

$$\rho_{t,h}^\flat = 1 + \frac{\beta_t h}{2} + \frac{h^2}{4}\left(\frac{\beta_t^2}{2} - \dot{\beta}_t\right) + \frac{h^3}{4}\left(\frac{\beta_t^3}{12} - \frac{\beta_t \dot{\beta}_t}{2} + \frac{\ddot{\beta}_t}{3}\right) + \cdots, \quad (23)$$

$$\mu_{t,h}^\flat = -\frac{\beta_t h}{2} + \frac{h^2}{4}\left(\dot{\beta}_t - \frac{\beta_t^2}{2\nu_t}\right) + \frac{h^3}{4}\left(\frac{\beta_t^3(-\nu_t^2 + 3\nu_t - 3)}{12\nu_t^2} + \frac{\beta_t \dot{\beta}_t}{2\nu_t} - \frac{\ddot{\beta}_t}{3}\right) + \cdots. \quad (24)$$

Using terms up to $O(h^2)$, the sampler will have 2nd-order convergence (henceforth referred to as *Taylor 2nd*), and using terms up to $O(h^3)$, the sampler will 3rd-order convergent (similarly, *Taylor 3rd*). If we use up to the $O(h)$ terms, the algorithm is same as the Euler method.

**Algorithm 2** (Quasi-Itô-Taylor Sampler with Ideal Derivatives for R-SDE). *Starting from a Gaussian noise $\mathbf{x}_T \sim \mathcal{N}(\mathbf{0}, \mathbf{I})$, iterate the following refinement steps until $\mathbf{x}_0$ is obtained.*

$$\mathbf{x}_{t-h} = \rho_{t,h}^{\sharp} \mathbf{x}_t + \mu_{t,h}^{\sharp} \mathbf{S}_\theta(\mathbf{x}_t, t)/\sqrt{\nu_t} + \mathbf{n}_{t,h}^{\sharp}, where \tag{25}$$

$$\rho_{t,h}^{\sharp} = 1 + \frac{\beta_t}{2} h + \frac{h^2}{4}\left(\frac{\beta_t^2}{2} - \dot{\beta}_t\right), \qquad \mu_{t,h}^{\sharp} = -\beta_t h + \frac{\dot{\beta}_t h^2}{2}, \tag{26}$$

$$\mathbf{n}_{t,h}^{\sharp} = \sqrt{\beta_t}\sqrt{h}\mathbf{w}_t + h^{3/2}\left(-\frac{\dot{\beta}_t}{2\sqrt{\beta_t}}(\mathbf{w}_t - \mathbf{z}_t) + \frac{\beta_t^{3/2}(\nu_t - 2)}{2\nu_t}\mathbf{z}_\tau\right). \tag{27}$$

*The Gaussian variables $\mathbf{w}_t$ and $\mathbf{z}_t$ have dimension-wise correlations, and each dimension is sampled similarly to Theorem 1.*

**Computation Cost:** At first glance, these algorithms may appear to be very complex. However, the computational complexity hardly increases compared to the Euler or Euler-Maruyama methods, because almost all of the computational cost is accounted for by the neural network $\mathbf{S}_\theta(\mathbf{x}_t, t)$, and the costs for scalar values $\rho_{t,h}^{\bullet}, \mu_{t,h}^{\bullet}$ and noise generation $\mathbf{n}_{t,h}^{\sharp}$ are almost negligible. It should also be noted that these scalar values can be pre-computed and stored in the memory before synthesis. Thus the computational complexity of these methods are practically equal to Euler, Euler-Maruyama, and DDIM methods.

**Error from the Exact Solution of PF-ODE:** The numerical error of the Quasi-Taylor method from the exact solution increases depending on the following factors: (1) The truncation error of the Taylor series in each step, i.e. $O(h^{p+1})$, (2) The number of the steps i.e. $O(1/h)$, (3) The training and generalization error of the score function, i.e. $\approx \mathcal{L}$, and (4) The average error between the *true* and *ideal* derivatives of the score function $=: \|\delta\|$. If the factors 3 and 4 could be zero, then the numerical error is the order of $O(h^p)$. Otherwise, the expected numerical error is roughly evaluated as follows,

$$error = O\left(h^{-1}(h\mathcal{L} + h^2(\mathcal{L} + \|\delta\|) + h^3(\mathcal{L} + \|\delta\|) + \cdots + h^{p+1})\right)$$

$$= O\left(\mathcal{L} + h(\mathcal{L} + \|\delta\|) + h^2(\mathcal{L} + \|\delta\|) + \cdots + h^p\right). \tag{28}$$

That is, the error of Euler method is $O(\mathcal{L} + h)$, the Heun method (2nd order Runge-Kutta) will be $O(\mathcal{L} + h\mathcal{L} + h^2)$, and the Taylor-2nd method is $O(\mathcal{L} + h(\mathcal{L} + \|\delta\|) + h^2)$. As long as $\mathcal{L}, \|\delta\| > 0$, the predominant $O(h)$ term will not disappear. Therefore, the overall order of the error will not decrease even if we increase the order of Taylor series greater than $p \geq 3$. Nevertheless, beyond such an order evaluation, specific coefficients in higher order terms can still affect the performance, which should be validated empirically.

## 4 IMAGE SYNTHESIS EXPERIMENT

**Experimental Configuration:** In this section, we conduct experiments to verify the effectiveness of the methods developed in this paper. Specifically, we compare the performance of the Euler scheme eq. (13), **Taylor 2nd & Taylor 3rd** (Alg. 1), DDIM (Song et al., 2020a), and the Runge Kutta methods (Heun and RK4 § E.5; these are less efficient than others because of NFEs per step) for PF-ODE, as well as the Euler-Maruyama scheme eq. (12) and **Itô-Taylor** (Alg. 2) for R-SDE. The datasets we used were CIFAR-10 ($32 \times 32$) (Krizhevsky, 2009) and CelebA ($64 \times 64$) (Liu et al., 2015).

The network structure was not novel but was based on an existing open source implementation; we used the "NCSN++" implemented in the official PyTorch code by Song et al. (2020b). The network consisted of 4 levels of resolution, with the feature dimension of each level being $128 \to 128 \to 256 \to 256 \to 256$. Each level consisted of BigGAN-type ResBlocks, and the number of ResBlocks in each level was 8 (CIFAR-10) and 4 (CelebA). The loss function we used was the unweighted $L_2$ loss similarly to (Ho et al., 2020). The optimizer was Adam (Kingma & Ba, 2014). The machine used for training was an in-house Linux server dedicated to medium-scale machine learning training with four GPUs (NVIDIA Tesla V100). The batch size was 256. The number of training steps was 0.1 M steps, and the training took about a day for each dataset.

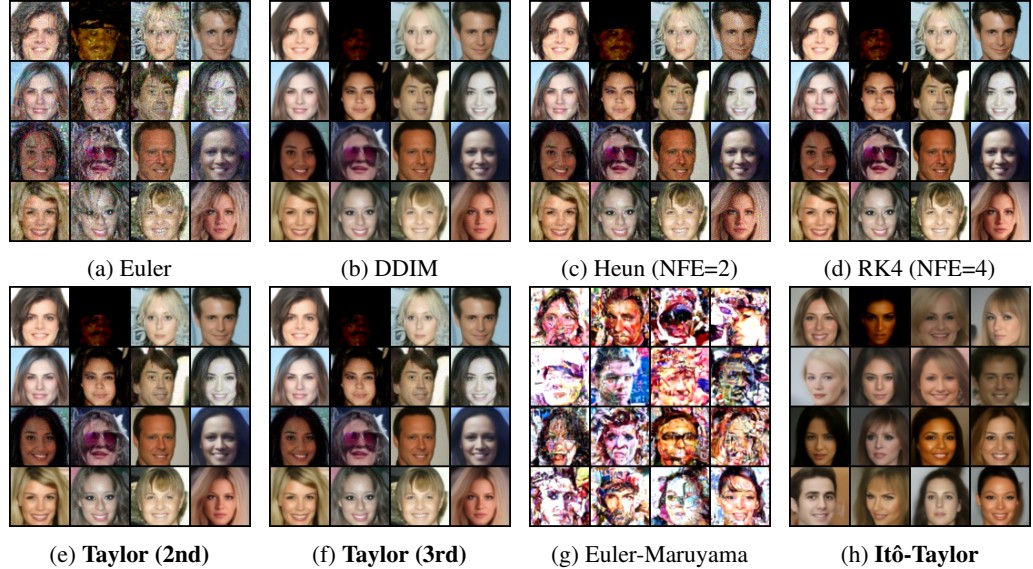

| (a) Euler | (b) DDIM | (c) Heun (NFE=2) | (d) RK4 (NFE=4) |

| (e) **Taylor (2nd)** | (f) **Taylor (3rd)** | (g) Euler-Maruyama | (h) **Itô-Taylor** |

Figure 1: Comparison of the synthesis results of CelebA ($64 \times 64$) data. The number of refinement steps is $N = 12$.

The noising schedule was also the same as the existing one, the default configuration of VP-SDE (Song et al., 2020b): $\beta_t = 0.1 + 19.9t$ and $\nu_t = 1 - \exp(-0.1t - 9.95t^2)$ eq. (76). The integration duration was $T = 1$, and the step size $h$ was constant, i.e. $h = T/N$ where $N$ is the number of refinement steps.

As a quality assessment metric, we used the Fréchet Inception Distance (FID) (Heusel et al., 2017). To evaluate FIDs, we used the pretrained Inception v3 checkpoint (Szegedy et al., 2016), and resized all images to $299 \times 299 \times 3$ by bilinear interpolation before feeding them to the Inception network. For each condition, 10,000 images were randomly generated to compute the FID score. Note that in this experiment, the computational resources for training were limited, and training was stopped before it fully converged (only 0.1 M steps, while in some other papers the number of training steps was e.g. 1.3 M steps in (Song et al., 2020b)). Therefore, it would be necessary to observe relative comparisons between samplers rather than directly comparing these FID value to those presented in other papers.

**Results:** Figure 1 and Figure 2 show random samples for each sampler. More examples are available in § G. The deterministic samplers considered in this paper generated plausible images much faster than the vanilla Euler-Maruyama sampler.

Figure 3a and Figure 3b reports the FID scores. From these figures, the following observations can be made. First, the proposed Quasi-Taylor methods have about the same or slightly better than DDIM. The reason for this is discussed in the next section § 5. We also found that the Runge-Kutta methods reduces FID in fewer steps overall. However, they also hit bottom faster. This may be due to the effect of the singularity at the time origin (see § D) in the final step. (This can be seen in Figure 16. In the second right column, the Runge-Kutta methods produce images similar to the other deterministic samplers, but the rightmost ones seem to be slightly noisier than the others).

Even though the ideal derivatives are only approximations and contain some errors, the convergence destinations of Quasi-Taylor methods were almost the same as the Runge-Kutta methods. This suggests that the error in the ideal derivatives is actually hardly a problem, because in regions where the approximation error is large, the state $\mathbf{x}_t$ is noisy to begin with (e.g. left 2/3 figures in Figure 16), and the approximation error is negligible compared to the noise that was originally there.

The proposed stochastic sampler (Itô-Taylor) also showed sufficiently competitive results, in terms of both FID scores and visual impression. Comparison of the figures in § G (e.g. Figure 21) confirms that the Itô-Taylor method empirically reaches almost the same target as Euler-Maruyama method

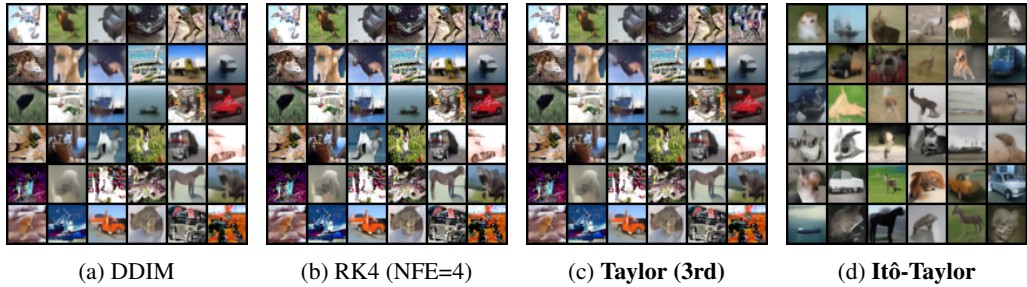

|  (a) DDIM | (b) RK4 (NFE=4) | (c) **Taylor (3rd)** | (d) **Itô-Taylor** |

Figure 2: Comparison of the synthesis results of CIFAR-10 ($32 \times 32$) data. The number of refinement steps is $N = 30$.

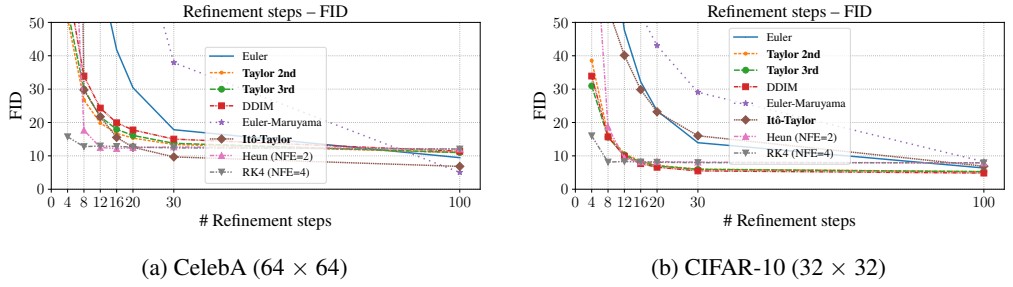

|  (a) CelebA ($64 \times 64$) | (b) CIFAR-10 ($32 \times 32$) |

Figure 3: Comparison of FID scores ($\downarrow$).

much more accurately, and it could be expected to be a safe alternative to Euler-Maruyama method when stochastic sampling is important.

## 5 DISCUSSION: RELATIONSHIP WITH DDIM

In the above experiment, the performance of the proposed Quasi-Taylor methods are found to be almost equivalent to that of DDIM. In fact, despite having distinctly different derivation logics, the proposed method and DDIM actually agree, at least up to the 3rd order terms of $h$. Therefore, it is not surprising the results are similar; and the smaller $h$ is, the closer the results are. This can be quickly verified by doing a Taylor expansion of the coefficients of eq. (15), i.e., $\frac{\alpha_{t-h}}{\alpha_t}$ and $(\sigma_{t-h} - \frac{\alpha_{t-h}}{\alpha_t}\sigma_t)$, w.r.t. $h$. Although it is tedious to perform this calculation by hand, the computer algebra systems e.g. SymPy immediately calculate it. For this computation, see § C.

This finding that truncating DDIM at the 2nd or 3rd order of $h$ yields exactly the same algorithms as the proposed Quasi-Taylor methods may be a useful insight for DDIM users, even if it does not lead them to switch the regular sampler from DDIM to Quasi-Taylor. That is, it offers an option of truncating the higher-order terms of DDIM.

## 6 CONCLUDING REMARKS

This paper proposed a Taylor-expansion approach for diffusion generative models, particularly the Probability Flow ODE (PF-ODE) and the reverse-time SDE (R-SDE) solvers. The assumptions to derive our sampler were minimalistic, and the derivation process was straightforward. We just substituted the derivatives in the Taylor series by ideal ones.

The obtained Quasi-Taylor and Quasi-Itô-Taylor samplers performed better than or on par with DDIM and Runge-Kutta methods. This fact implicitly supports the validity of our approximations. Conversely, if we could find some examples where the Quasi-Taylor methods, DDIM and RK methods gave decisively different results, we might be able to gain a deeper understanding of the structure of data manifold and the fundamentals of diffusion models by investigating the causes of discrepancy.

**Reproducibility Statement** Pseudocodes of the proposed methods are available in § F, and the derivation of the proposed method is described in § B.1, § B.3. The experiment is based on open source code with minimal modifications to match the proposed method, and all the data used in this paper are publicly available. Experimental conditions are elaborated in § 4.

**Ethics Statement** As a final note, negative aspects of generative models are generally pointed out, such as the risk of reproducing bias and discrimination in training data and the risk of being misused for deep fakes. Since this method only provides a solution to existing generative models, it does not take special measures against these problems. Maximum ethical care should be taken in the practical application of this method.

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

## A   On the Approximation $\nabla_{\mathbf{x}_t} \log p(\mathbf{x}_t, t) \approx \nabla_{\mathbf{x}_t} \log p(\mathbf{x}_t, t \mid \mathbf{x}_0, 0)$

Although $\nabla_{\mathbf{x}_t} \log p(\mathbf{x}_t, t)$ and $\nabla_{\mathbf{x}_t} \log p(\mathbf{x}_t, t \mid \mathbf{x}_0, 0)$ are clearly distinct concepts, they are nearly equivalent in practical situations that the data space is high-dimensional and the data are distributed in a small subset (low-dimensional manifold) of the space. That is, in such a case, the integrated gradient $\nabla_{\mathbf{x}_t} \log p(\mathbf{x}_t, t)$, given by

$$
\begin{aligned}
\nabla_{\mathbf{x}_t} \log p(\mathbf{x}_t, t) &= \nabla_{\mathbf{x}_t} \log \int p(\mathbf{x}_t, t \mid \mathbf{x}_0, 0) p(\mathbf{x}_0, 0) d\mathbf{x}_0 \\
&= \frac{\nabla_{\mathbf{x}_t} \int p(\mathbf{x}_t, t \mid \mathbf{x}_0, 0) p(\mathbf{x}_0, 0) d\mathbf{x}_0}{\int p(\mathbf{x}_t, t \mid \mathbf{x}_0, 0) p(\mathbf{x}_0, 0) d\mathbf{x}_0} \\
&= \frac{\mathbb{E}_{p(\mathbf{x}_0)} \left[ \nabla_{\mathbf{x}_t} p(\mathbf{x}_t, t \mid \mathbf{x}_0, 0) \right]}{\mathbb{E}_{p(\mathbf{x}_0)} \left[ p(\mathbf{x}_t, t \mid \mathbf{x}_0, 0) \right]} \\
&= \frac{\mathbb{E}_{p(\mathbf{x}_0)} \left[ -\frac{\mathbf{x}_t - \sqrt{1-\nu_t}\mathbf{x}_0}{\nu_t} p(\mathbf{x}_t, t \mid \mathbf{x}_0, 0) \right]}{\mathbb{E}_{p(\mathbf{x}_0)} \left[ p(\mathbf{x}_t, t \mid \mathbf{x}_0, 0) \right]} \\
&= \mathbb{E}_{p(\mathbf{x}_0)} \Bigg[ \underbrace{-\frac{\mathbf{x}_t - \sqrt{1-\nu_t}\mathbf{x}_0}{\nu_t}}_{=\nabla_{\mathbf{x}_t} \log p(\mathbf{x}_t, t \mid \mathbf{x}_0, 0)} \underbrace{\frac{e^{-\|\mathbf{x}_t - \sqrt{1-\nu_t}\mathbf{x}_0\|^2/2\nu_t}}{\mathbb{E}_{p(\mathbf{x}_0)} [e^{-\|\mathbf{x}_t - \sqrt{1-\nu_t}\mathbf{x}_0\|^2/2\nu_t}]}}_{=:q(\mathbf{x}_0 \mid \mathbf{x}_t)} \Bigg],
\end{aligned}
\tag{29}
$$

almost always has a single point approximation which is written as $\nabla_{\mathbf{x}_t} \log p(\mathbf{x}_t, t \mid \mathbf{x}_0^{(i)}, 0)$, where $\mathbf{x}_0^{(i)}$ is a certain point on the data manifold. There are two phases depending on the noise scale $\bar{\alpha}_t = 1 - \nu_t$, and both phases have different reasons for the validity of the approximation.

### A.1   Phase (1): Anyone can be a representative ($\bar{\alpha}_t = 1 - \nu_t \sim 0$)

If $\mathbf{x}_t$ is far from any of $\{\sqrt{1-\nu_t}\mathbf{x}_0^{(i)} \mid \mathbf{x}_0^{(i)} \sim p(\mathbf{x}_0)\}$, the gradients from each scaled data points are almost the same. Therefore, the integrated one viz. $\nabla \log p(\mathbf{x}_t, t)$ can be approximated by any of $\nabla \log p(\mathbf{x}_t, t \mid \mathbf{x}_0^{(i)}, 0)$ for any $\mathbf{x}_0^{(i)} \sim p(\mathbf{x}_0)$. Figure 4 intuitively illustrates the reason.

More quantitatively, noting that the weight function $q(\cdot)$ satisfies $\mathbb{E}_{p(\mathbf{x}_0)}[q(\mathbf{x}_0 \mid \mathbf{x}_t)] = 1$, the $L_2$ distance between $\nabla \log p(\mathbf{x}_t, t)$ and $\nabla \log p(\mathbf{x}_t, t \mid \mathbf{x}_0^{(i)}, 0)$ is bounded above as follows,

$$
\begin{aligned}
\|\nabla &\log p(\mathbf{x}_t, t) - \nabla \log p(\mathbf{x}_t, t \mid \mathbf{x}_0^{(i)}, 0)\|_2^2 \\
&= \left\| \mathbb{E}_{p(\mathbf{x}_0)} \left[ \frac{\mathbf{x}_t - \sqrt{1-\nu_t}\mathbf{x}_0}{\nu_t} q(\mathbf{x}_0 \mid \mathbf{x}_t) \right] - \frac{\mathbf{x}_t - \sqrt{1-\nu_t}\mathbf{x}_0^{(i)}}{\nu_t} \right\|_2^2 \\
&= \left\| \mathbb{E}_{p(\mathbf{x}_0)} \left[ \left( \frac{\mathbf{x}_t - \sqrt{1-\nu_t}\mathbf{x}_0}{\nu_t} - \frac{\mathbf{x}_t - \sqrt{1-\nu_t}\mathbf{x}_0^{(i)}}{\nu_t} \right) q(\mathbf{x}_0 \mid \mathbf{x}_t) \right] \right\|_2^2 \\
&= \frac{1-\nu_t}{\nu_t^2} \left\| \mathbb{E}_{p(\mathbf{x}_0)} \left[ (\mathbf{x}_0 - \mathbf{x}_0^{(i)}) q(\mathbf{x}_0 \mid \mathbf{x}_t) \right] \right\|_2^2 \\
&\leq \frac{1-\nu_t}{\nu_t^2} \mathbb{E}_{p(\mathbf{x}_0)} \left[ \|\mathbf{x}_0 - \mathbf{x}_0^{(i)}\|_2^2 q(\mathbf{x}_0 \mid \mathbf{x}_t) \right] \quad (\because \text{Jensen's inequality}) \\
&\leq \frac{1-\nu_t}{\nu_t^2} \left( \max_{\mathbf{x}_0} \|\mathbf{x}_0 - \mathbf{x}_0^{(i)}\|_2^2 \mathbb{E}_{p(\mathbf{x}_0)} [q(\mathbf{x}_0 \mid \mathbf{x}_t)] \right) \\
&= \frac{1-\nu_t}{\nu_t^2} \left( \max_{\mathbf{x}_0} \|\mathbf{x}_0 - \mathbf{x}_0^{(i)}\|_2^2 \right) \\
&\leq \frac{4(1-\nu_t)R_{\mathcal{M}}^2}{\nu_t^2},
\end{aligned}
\tag{30}
$$

where $R_{\mathcal{M}}$ is the radius of the smallest ball that covers the data manifold $\mathcal{M}$. The radius $R_{\mathcal{M}}$ will be a finite constant in most practical scenarios we are interested in. For example, if the data space is $d$-dim box, i.e., $\mathcal{M} \subset [0, 1]^d$, then $R_{\mathcal{M}}$ is bounded above by $R_{\mathcal{M}} \leq \sqrt{d}/2$.

Figure 4: Intuitive reasons why a single point approximation is valid. The case where $1 - \nu_t \approx 0$

Noting that $\|\nabla \log p(\mathbf{x}_t, t \mid \mathbf{x}_0^{(i)}, 0)\|_2 \approx \sqrt{d/\nu_t}$ (this is actually a random variable that follows the $\chi$-distribution[1]), the relative $L_2$ error is evaluated as follows,

$$\text{(relative } L_2 \text{ error)} = \frac{\|\nabla \log p(\mathbf{x}_t, t) - \nabla \log p(\mathbf{x}_t, t \mid \mathbf{x}_0^{(i)}, 0)\|_2}{\|\nabla \log p(\mathbf{x}_t, t \mid \mathbf{x}_0^{(i)}, 0)\|_2} \lessapprox \sqrt{\frac{1 - \nu_t}{\nu_t}}. \tag{31}$$

Similarly, the cosine similarity is evaluated as

$$\text{cossim}(\nabla \log p(\mathbf{x}_t, t), \nabla \log p(\mathbf{x}_t, t \mid \mathbf{x}_0^{(i)}, 0)) \gtrapprox 1 - \frac{1}{2}\left(1 + \sqrt{\frac{1 - \nu_t}{\nu_t}}\right)\frac{1 - \nu_t}{\nu_t}. \tag{32}$$

These bounds suggest that the relative error between the true gradient and the single point approximation approaches to 0, and the cosine similarity goes to 1, as $\nu_t \to 1$ ($\bar{\alpha}_t \to 0$). That is, the single point approximation is largely valid when the noise level $\nu_t$ is high.

**Proofs for some facts used in this section**

■ *Jensen's inequality*: A sketchy proof of Jensen's inequality for $L_2$-norm is as follows:

$$\left\|\sum_n w_n \mathbf{x}^{(n)}\right\|_2^2 = \sum_i \left[\sum_n w_n \mathbf{x}^{(n)}\right]_i^2 = \sum_i \left(\sum_n w_n x_i^{(n)}\right)^2$$

$$\leq \sum_i \sum_n w_n (x_i^{(n)})^2 = \sum_n w_n \sum_i (x_i^{(n)})^2 = \sum_n w_n \|\mathbf{x}^{(n)}\|_2^2. \tag{33}$$

■ *Cosine similarity*: Given the relative distance between $\mathbf{x}$ and $\mathbf{y}$ as $\|\mathbf{x} - \mathbf{y}\|_2/\|\mathbf{x}\|_2 = r$ (for simplicity, let $r \ll 1$) then the cosine similarity between the vectors are evaluated as follows,

$$\text{cossim}(\mathbf{x}, \mathbf{y}) = \frac{\langle \mathbf{x}, \mathbf{y} \rangle}{\|\mathbf{x}\|_2 \|\mathbf{y}\|_2} = \frac{\|\mathbf{x}\|_2}{\|\mathbf{y}\|_2} \cdot \frac{\langle \mathbf{x}, \mathbf{y} \rangle}{\|\mathbf{x}\|_2^2} = \frac{1}{2}\frac{\|\mathbf{x}\|_2}{\|\mathbf{y}\|_2} \cdot \frac{\|\mathbf{x}\|_2^2 + \|\mathbf{y}\|_2^2 - \|\mathbf{x} - \mathbf{y}\|_2^2}{\|\mathbf{x}\|_2^2}$$

$$= \frac{1}{2}\frac{\|\mathbf{x}\|_2}{\|\mathbf{y}\|_2} \cdot \left(1 + \frac{\|\mathbf{y}\|_2^2}{\|\mathbf{x}\|_2^2} - r^2\right) = \frac{1}{2}\left(\frac{\|\mathbf{x}\|_2}{\|\mathbf{y}\|_2} + \frac{\|\mathbf{y}\|_2}{\|\mathbf{x}\|_2} - \frac{\|\mathbf{x}\|_2}{\|\mathbf{y}\|_2}r^2\right). \tag{34}$$

Noting that the triangle inequality implies that the $L_2$ norm of $\mathbf{y}$ is bounded as follows,

$$(1 - r)\|\mathbf{x}\|_2 \leq \|\mathbf{y}\|_2 \leq (1 + r)\|\mathbf{x}\|_2, \tag{35}$$

we can put $\|\mathbf{y}\|_2/\|\mathbf{x}\|_2 = 1 + \delta, (0 < |\delta| < r \ll 1)$, and can evaluate the cosine similarity as follows,

$$\text{cossim}(\mathbf{x}, \mathbf{y}) = \frac{1}{2}\left(\frac{1}{1 + \delta} + (1 + \delta) - \frac{1}{1 + \delta}r^2\right)$$

$$\approx \frac{1}{2}\left((1 - \delta) + (1 + \delta) - (1 - \delta)r^2\right) > 1 - \frac{1}{2}(1 + r)r^2. \tag{36}$$

---

[1]Given Gaussian random variables $X_i \sim \mathcal{N}(0, 1)$, then the squared sum of them $\sum_{i=1}^d X_i^2$ follows the $\chi^2$-distribution with $d$ degrees of freedom. It is well known that $\chi^2$ distribution converges to $\mathcal{N}(d, 2d)$ as $d$ increases. Therefore, If $d$ is sufficiently large, $\|\mathbf{x}\|_2 := (\sum_{i=1}^d X_i^2)^{1/2} \approx \sqrt{d \pm \sqrt{2d}} \approx \sqrt{d} \pm \sqrt{1/2}$. Thus the $L_2$ norm of $d$-dim Gaussian variable is approximated by $\sqrt{d}$, and the error is as small as the scale of $\sqrt{1/2}$. It implies that Gaussian distribution looks like a sphere in high-dimensional space, contrary to our low-dimensional intuition. Indeed, the following inequality holds (Laurent & Massart, 2000, Lem. 1): $p((1 - 2\sqrt{y/d})^{1/2} \leq \|\mathbf{x}\|_2/\sqrt{d} \leq (1 + 2\sqrt{y/d} + 2y/d)^{1/2}) \geq 1 - 2e^{-y}$.

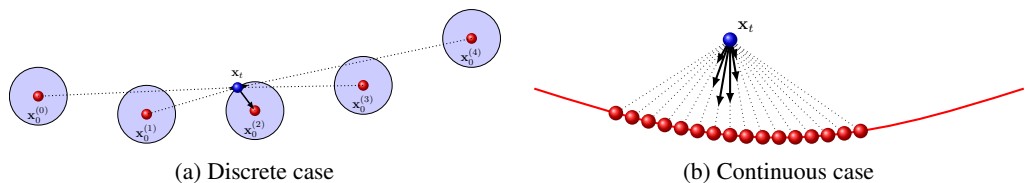

(a) Discrete case                                   (b) Continuous case

Figure 5: Intuitive reasons why a single point approximation is valid. The case where $1 - \nu_t \approx 1$

## A.2  PHASE (2): WINNER TAKES ALL $(\bar{\alpha}_t = 1 - \nu_t \gg 0)$

The above bounds suggest that the single point approximation is valid only when the noise level $\nu_t$ is high ($\nu_t \approx 1$). However, we can also show that the approximation is also valid because of another reason when the noise level $\nu_t$ is low ($\nu_t \approx 0$).

If $p(\mathbf{x}_0)$ is a discrete distribution, and if $\nu_t \approx 0$, the weight factor $q(\mathbf{x}_0^{(i)} \mid \mathbf{x}_t)$ in eq. (29) is almost certainly a "one-hot vector" such that

$$q(\mathbf{x}_0^{(i)} \mid \mathbf{x}_t) \approx 1, \quad \text{and} \quad q(\mathbf{x}_0^{(j)} \mid \mathbf{x}_t) \approx 0 \ (j \neq i). \tag{37}$$

That is, the entropy of $q(\mathbf{x}_0 \mid \mathbf{x}_t)$ is nearly zero (see Figure 8). It is almost certain (prob $\approx 1$), since the converse is quite rare; i.e., there exists another data point $\mathbf{x}_0^{(j)}$ near $\mathbf{x}_t \sim p(\mathbf{x}_t, t \mid \mathbf{x}_0^{(j)}, 0)$ in such a situation[2]. Figure 5a intuitively shows the reason.

If $p(\mathbf{x}_0)$ is a continuous distribution, very small area (almost a single point) of the data manifold contributes to the integration eq. (29). Figure 5b intuitively shows the situation. The weight function will become nearly the delta function, and the integration will be almost the same as the single point approximation,

$$q(\mathbf{x}_0 \mid \mathbf{x}_t) \approx \delta(\mathbf{x}_0 - \mathbf{x}_0^*). \tag{38}$$

where $\mathbf{x}_0^*$ is a certain point close to the perpendicular projection of $\mathbf{x}_t$ on the data manifold. These are related to the well-known formulae $\varepsilon \log \sum \exp(\cdot/\varepsilon)$ and $\varepsilon \log \int \exp(\cdot/\varepsilon)$ goes to $\max(\cdot)$ when $\varepsilon \to 0$. Naturally, this is more likely to be true for high-dimensional space and is expected to break down for low-dimensional toy data.

Now let us elaborate the above discussion. As the scale we are interested in now is very small, the data manifold is approximated as a flat $k$-dim space. In addition, as $q(\cdot \mid \mathbf{x}_t)$ is a Gaussian function, it decays rapidly with distance from the center (perpendicular projection of $\mathbf{x}_t \in \mathbb{R}^d$ to the $k$-dim

---

[2]Let us evaluate the probability more qualitatively using a toy model. Let us denote a $k$-dim ball of radius $r$ by $B^k(r)$, and let $D_i$ be the $L_2$ distance between $\mathbf{x}_t$ and $\sqrt{1 - \nu_t}\mathbf{x}_0^{(i)}$, i.e., $D_i := \|\mathbf{x}_t - \sqrt{1 - \nu_t}\mathbf{x}_0^{(i)}\|_2 \approx \sqrt{\nu_t d}$. The question is whether there exists another data point $\mathbf{x}_0^{(j)}(j \neq i)$ in the discrete data distribution $\mathcal{D} = \{\mathbf{x}_0^{(i)}\}_i$ such that $D_j < D_i$. In other words, whether there exists another data point in $B^d(\sqrt{\nu_t d})$ centered at $\sqrt{1 - \nu_t}\mathbf{x}_0^{(i)}$. If it is the case, $q(\mathbf{x}_0^{(j)} \mid \mathbf{x}_t)$ has significantly large value compared to $q(\mathbf{x}_0^{(i)} \mid \mathbf{x}_t)$, and it will have strong effects on the result of integration eq. (29). Now let us show that it is unlikely in a high-dimensional data space. As it is difficult evaluating the probability for general cases, let us consider a toy model that the all the $n$ points are uniformly distributed in a $k$-dim ball ($k \ll d$). That is, we assume that the data manifold is $\mathcal{M} = B^k(\sqrt{k})$ which satisfies $\mathcal{D} \subset \mathcal{M} \subset \mathbb{R}^d$. Then, the probability we are interested in is roughly evaluated as

$$p(\exists j \neq i, D_j < D_i) \approx 1 - \left(1 - \frac{\text{vol}[B^k(\sqrt{\nu_t k})]}{\text{vol}[\sqrt{1 - \nu_t}\mathcal{M}]}\right)^n = 1 - \left(1 - \frac{\text{vol}[B^k(\sqrt{\nu_t k})]}{\text{vol}[B^k(\sqrt{(1 - \nu_t)k})]}\right)^n.$$

Remembering that the volume of $k$-dim ball is given by $\text{vol}[B^k(r)] = \frac{\pi^{k/2}}{\Gamma(k/2+1)}r^k$, the probability is approximated as follows,

$$\text{r.h.s} = 1 - \left(1 - \left(\frac{\nu_t k}{(1 - \nu_t)k}\right)^{k/2}\right)^n \approx n\left(\frac{\nu_t}{1 - \nu_t}\right)^{k/2}, \quad \text{if } \nu_t \approx 0, k \gg 1.$$

Thus, when $\nu_t$ is sufficiently small and the data manifold is sufficiently high-dimensional, it is almost unlikely that there exists such a point $\mathbf{x}_0^{(j)}$ ($j \neq i$) unless $n$ is very large.

subspace). For this reason, only a small ball around the perpendicular foot of $\mathbf{x}_t$ contributes to the integration. As the majority of $k$-dim Gaussian variables of variance $\sigma^2 \mathbf{I}$ are distributed on a sphere of radius $\sqrt{k}\sigma$, it would be sufficient to consider a ball of radius $(\sqrt{k}+2)\sqrt{\nu_t}$.

Now the integration eq. (29) is approximated as follows,

$$\text{eq. (29)} \approx \int_{\mathcal{B}} \left[ \nabla_{\mathbf{x}_t} \log p(\mathbf{x}_t, t \mid \mathbf{x}_0, 0) q(\mathbf{x}_0 \mid \mathbf{x}_t) p(\mathbf{x}_0) \right] d\mathbf{x}_0,$$

$$\text{where } \mathcal{B} = \{\mathbf{x}_0 \mid \|\sqrt{1-\nu_t}\mathbf{x}_0 - \mathbf{x}_t\|_2 < (\sqrt{k}+2)\sqrt{\nu_t}\}. \quad (39)$$

As $\mathcal{B}$ is small, the gradient vector $\nabla_{\mathbf{x}_t} \log p(\mathbf{x}_t, t \mid \mathbf{x}_0, 0)$ is almost constant in this region. Quantitatively, let $\mathbf{x}_0^{(1)}, \mathbf{x}_0^{(2)} \in \mathcal{B}$, then the distance between these points is evidently bounded above by $\|\mathbf{x}_0^{(1)} - \mathbf{x}_0^{(2)}\|_2 \leq 2(\sqrt{k}+2)\sqrt{\nu_t}$, which implies

$$\left\| \nabla \log p(\mathbf{x}_t, t \mid \mathbf{x}_0^{(1)}, 0) - \nabla \log p(\mathbf{x}_t, t \mid \mathbf{x}_0^{(2)}, 0) \right\|_2$$

$$= \left\| \frac{\mathbf{x}_t - \sqrt{1-\nu_t}\mathbf{x}_0^{(1)}}{\nu_t} - \frac{\mathbf{x}_t - \sqrt{1-\nu_t}\mathbf{x}_0^{(2)}}{\nu_t} \right\|_2$$

$$= \left\| \frac{\sqrt{1-\nu}(\mathbf{x}_0^{(1)} - \mathbf{x}_0^{(2)})}{\nu_t} \right\|_2 \leq \frac{\sqrt{1-\nu_t}}{\nu_t} \cdot 2(\sqrt{k}+2)\sqrt{\nu_t}$$

$$= 2(\sqrt{k}+2)\sqrt{\frac{1-\nu_t}{\nu_t}}. \quad (40)$$

Noting that $\|\nabla \log p(\mathbf{x}_t, t \mid \mathbf{x}_0, 0)\|_2 \approx \sqrt{d/\nu_t}$, the relative $L_2$ error between two gradients is bounded above by

$$\frac{\|\nabla \log p(\mathbf{x}_t, t \mid \mathbf{x}_0^{(1)}, 0) - \nabla \log p(\mathbf{x}_t, t \mid \mathbf{x}_0^{(2)}, 0)\|_2}{\|\nabla \log p(\mathbf{x}_t, t \mid \mathbf{x}_0^{(2)}, 0)\|_2} \lessapprox 2\frac{(\sqrt{k}+2)\sqrt{1-\nu_t}}{\sqrt{d}} \lessapprox 2\sqrt{\frac{k}{d}}. \quad (41)$$

By integrating both sides [3] w.r.t. $\mathbf{x}_0^{(1)}$, i.e. $\int (\cdot) q(\mathbf{x}_0^{(1)} \mid \mathbf{x}_t) p(\mathbf{x}_0^{(1)}) d\mathbf{x}_0^{(1)}$, we obtain the following inequality for any point $\mathbf{x}_0^* \in \mathcal{B}$,

$$\frac{\|\nabla \log p(\mathbf{x}_t, t) - \nabla \log p(\mathbf{x}_t, t \mid \mathbf{x}_0^*, 0)\|_2}{\|\nabla \log p(\mathbf{x}_t, t \mid \mathbf{x}_0^*, 0)\|_2} \lessapprox 2\sqrt{\frac{k}{d}}. \quad (42)$$

Now, let us recall that a noised data $\mathbf{x}_t \sim p(\mathbf{x}_t, t)$ is generated in the following procedure.

1. Draw a data $\mathbf{x}_0^{(i)}$ from $p(\mathbf{x}_0, 0)$.
2. Draw a Gaussian $\mathbf{x}_t \sim p(\mathbf{x}_t, t \mid \mathbf{x}_0^{(i)}, 0) = \mathcal{N}(\sqrt{1-\nu_t}\mathbf{x}_0^{(i)}, \nu_t \mathbf{I}_d)$.

Here, the data point $\mathbf{x}_0^{(i)} \in \mathcal{M}$ is in the ball $\mathcal{B}$, because the distance between $\sqrt{1-\nu_t}\mathbf{x}_0^{(i)}$ and the perpendicular foot of $\mathbf{x}_t$ is about $\sqrt{\nu_t k}$, which is clearly smaller than $(\sqrt{k}+2)\sqrt{\nu_t}$. Thus we can write the following inequality.

$$\frac{\|\nabla \log p(\mathbf{x}_t, t) - \nabla \log p(\mathbf{x}_t, t \mid \mathbf{x}_0^{(i)}, 0)\|_2}{\|\nabla \log p(\mathbf{x}_t, t \mid \mathbf{x}_0^{(i)}, 0)\|_2} \lessapprox 2\sqrt{\frac{k}{d}}, \quad (43)$$

and we may conclude that the following approximation is largely valid if $\nu_t \approx 0, k \ll d$,

$$\nabla \log p(\mathbf{x}_t, t) \approx \nabla \log p(\mathbf{x}_t, t \mid \mathbf{x}_0^{(i)}, 0). \quad (44)$$

---

[3] We used the following relation: $\|\mathbb{E}_{\mathbf{x}}[f(\mathbf{x})] - \mathbf{y}\|_2 = \|\mathbb{E}_{\mathbf{x}}[f(\mathbf{x}) - \mathbf{y}]\|_2 \leq \mathbb{E}_{\mathbf{x}}[\|f(\mathbf{x}) - \mathbf{y}\|_2]$.

### A.3 COMPARISON OF THE EMPIRICAL SCORE FUNCTION AND THE SINGLE POINT APPROXIMATION

Let us empirically validate the accuracy of single point approximation using real data as follows,

- $\mathcal{D} = \{$MNIST (LeCun et al., 2010) 60,000 samples$\}$,
- $\mathcal{D} = \{$CIFAR-10 (Krizhevsky, 2009) 50,000 samples$\}$.

Since the true score function cannot be determined without knowing the true density (which will be possible with synthetic data, but discussing such data will not be very interesting here), the empirical score function was calculated using the real data $\mathcal{D}$ above as follows,

$$
\begin{aligned}
\textit{True Score} &= \nabla \log p(\mathbf{x}_t, t) \\
&= \mathbb{E}_{p(\mathbf{x}_0)}[q(\mathbf{x}_0 \mid \mathbf{x}_t) \nabla \log p(\mathbf{x}_t, t \mid \mathbf{x}_0, 0)] \\
&\approx \frac{1}{|\mathcal{D}|} \sum_{\mathbf{x}_0 \in \mathcal{D}} [q(\mathbf{x}_0 \mid \mathbf{x}_t) \nabla \log p(\mathbf{x}_t, t \mid \mathbf{x}_0, 0)] =: \textit{Empirical Score}.
\end{aligned} \tag{45}
$$

The evaluation of empirical score function using the entire dataset is unrealistic if the dataset $\mathcal{D}$ is large, but it is feasible if $\mathcal{D}$ is a small dataset like MNIST and CIFAR-10.

In order to evaluate the accuracy of single point approximation, we evaluated following three metrics.

- Relative $L_2$ error between the empirical score function and $\nabla \log p(\mathbf{x}_t, t \mid \mathbf{x}_0, 0)$,
- Cosine similarity between the empirical score function and $\nabla \log p(\mathbf{x}_t, t \mid \mathbf{x}_0, 0)$,
- Entropy of $q(\mathbf{x}_0 \mid \mathbf{x}_t)$.

Figure 6 shows the relative $L_2$ distance, for both datasets. Figure 7 similarly show the distribution (random 10,000 trials) of the cosine similarity, and Figure 8 shows the entropy. Dashed curves indicate the bounds evaluated in eq. (31) and eq. (32).

These figures show that the range of intermediate region between Phase (1) and Phase (2) will not have impact in practical situations since we do not evaluate the neural network $\mathbf{S}_\theta(\cdot, \cdot)$ in this range so many times (i.e., $\bar{\alpha}_t \sim 10^{-3}$ to $10^{-1} \Leftrightarrow \nu_t \sim 0.999$ to $0.9$). Moreover, the approximation accuracy is still very high even in this region. Furthermore, although MNIST and CIFAR-10 are quite "low-dimensional" for real-world images, approximations are established with such high accuracy. Therefore, it is expected to be established with higher accuracy for more realistic images.

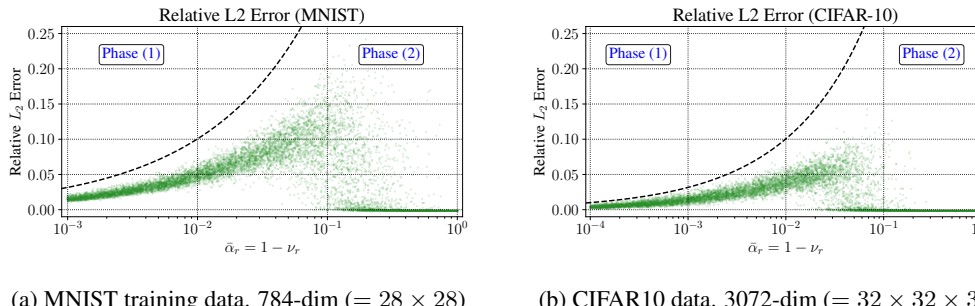

(a) MNIST training data. 784-dim $(= 28 \times 28)$    (b) CIFAR10 data. 3072-dim $(= 32 \times 32 \times 3)$

Figure 6: Relative $L_2$ distance ($\downarrow$) between the empirical score function eq. (45) and the single point approximation, depending on the noise level $\nu_t$. Dashed curves indicate the upper bounds evaluated in eq. (31).

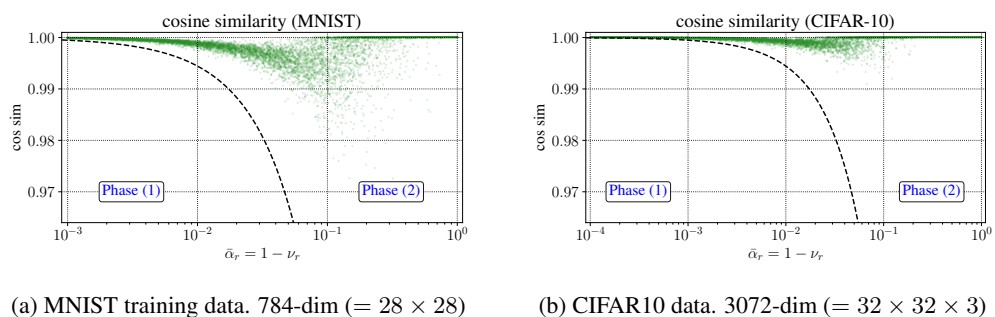

(a) MNIST training data. 784-dim $(= 28 \times 28)$    (b) CIFAR10 data. 3072-dim $(= 32 \times 32 \times 3)$

Figure 7: Cosine similarity ($\uparrow$) between the empirical score function and the single point approximation depending on the noise level $\nu_t$. Dashed curves indicate the lower bounds evaluated in eq. (32).

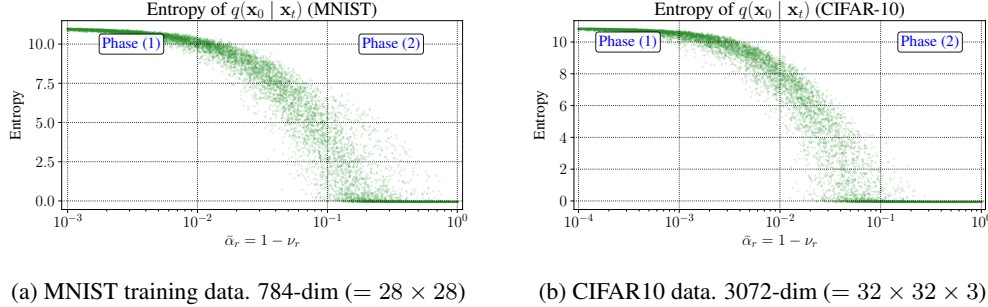

(a) MNIST training data. 784-dim $(= 28 \times 28)$    (b) CIFAR10 data. 3072-dim $(= 32 \times 32 \times 3)$

Figure 8: Entropy of $q(\mathbf{x}_0 \mid \mathbf{x}_t)$ depending on the noise level $\nu_t$.

## B    ON THE IDEAL DERIVATIVE APPROXIMATION

Thus, we can assume that the single point approximation almost always holds practically.

$$-\frac{\mathbf{S}_\theta(\mathbf{x}_t, t)}{\sqrt{\nu_t}} \stackrel{\text{model}}{\approx} \nabla_{\mathbf{x}_t} \log p(\mathbf{x}_t, t) \stackrel{\text{almost equal}}{\approx} \nabla_{\mathbf{x}_t} \log p(\mathbf{x}_t, t \mid \mathbf{x}_0^{(i)}, 0) = -\frac{\mathbf{x}_t - \sqrt{1-\nu_t}\mathbf{x}_0^{(i)}}{\nu_t}.$$

Therefore, we may also expect that the similar approximation will be valid for their derivatives.

Of course, strictly speaking, such an expectation is mathematically incorrect. For example, let $g(x) = f(x) + \varepsilon \sin \omega x$, then the difference $g(x) - f(x) = \varepsilon \sin \omega x$ goes to zero as $\varepsilon \to 0$, but the difference of derivatives $g'(x) - f'(x) = \varepsilon\omega \cos \omega x$ does not if $\omega \to \infty$ faster than $1/\varepsilon$. If the error between them in the Fourier domain is written as $E(\omega) = G(\omega) - F(\omega)$, then the $L_2$ error between the derivatives is $\|g'(x) - f'(x)\|_2^2 = \|\omega E(\omega)\|_2^2 \times \text{const}$ (Parseval's theorem). In other words, the single point approximation does not necessarily imply the ideal derivative approximation. If it is to be mathematically rigorous, it must be supported by other nontrivial knowledge on the data manifold. This nontrivial leap is the most important "conjecture" made in this paper and its theoretical background should be more closely evaluated in the future.

### B.1    DERIVATION OF THE "IDEAL DERIVATIVES"

Because of the discussion in § A, the true score function $\nabla_{\mathbf{x}_t} \log p(\mathbf{x}_t, t)$ is finely approximated by a single point approximation $\nabla_{\mathbf{x}_t} \log p(\mathbf{x}_t, t \mid \mathbf{x}_0, 0)$. Now we may also assume that the derivatives of both will also be close.

In this paper, we are interested in the Taylor expansion of the following form (see also § E.1.1),

$$\boldsymbol{\psi}(\mathbf{x}_h, h) = \boldsymbol{\psi}(\mathbf{x}_0, 0) + \sum_{k=1}^{\infty} \frac{h^k}{k!} \left(\partial_t + \mathbf{a}(\mathbf{x}_t, t) \cdot \nabla_{\mathbf{x}_t}\right)^k \boldsymbol{\psi}(\mathbf{x}_t, t)\bigg|_{t=0}. \tag{46}$$

If the function $\boldsymbol{\psi}(\mathbf{x}_t, t)$ is separable in each dimension (i.e., $\partial_{x_i}\psi_j = 0$ for $i \neq j$), the following relation holds,

$$(\mathbf{a}(\mathbf{x}_t, t) \cdot \nabla_{\mathbf{x}_t})\,\boldsymbol{\psi}(\mathbf{x}_t, t) = \mathbf{a}(\mathbf{x}_t, t) \odot \nabla_{\mathbf{x}_t} \odot \boldsymbol{\psi}(\mathbf{x}_t, t), \tag{47}$$

where $\odot$ is the element-wise product or operation. If $\mathbf{a}(\mathbf{x}_t, t)$ is also separable in each dimension[4] the Taylor series is formally rewritten as follows,

$$\boldsymbol{\psi}(\mathbf{x}_t, t) = \boldsymbol{\psi}(\mathbf{x}_0, 0) + \sum_{k=1}^{\infty} \frac{t^k}{k!} \left(\mathbf{1}\partial_t + \mathbf{a}(\mathbf{x}_t, t) \odot \partial_{\mathbf{x}_t}\right)^k \boldsymbol{\psi}(\mathbf{x}_t, t)\bigg|_{t=0} \tag{48}$$

where $\partial_{\mathbf{x}_t} := \nabla_{\mathbf{x}_t} \odot$ is the element-wise derivative operator. This is formally the same as the 1-dim Taylor series. Therefore, it is sufficient to consider the 1-dim Taylor series first, and parallelize each dimension later. Thus the derivatives we actually need are the following two.

$$\partial_{\mathbf{x}_t}\mathbf{S}_\theta(\mathbf{x}_t, t) = \nabla_{\mathbf{x}_t} \odot \mathbf{S}_\theta(\mathbf{x}_t, t), \quad \partial_t \mathbf{S}_\theta(\mathbf{x}_t, t) = (\mathbf{1}\partial_t) \odot \mathbf{S}_\theta(\mathbf{x}_t, t). \tag{49}$$

### B.1.1    SPATIAL DERIVATIVE $\partial_{\mathbf{x}_t}\mathbf{S}_\theta(\mathbf{x}_t, t) := \nabla_{\mathbf{x}_t} \odot \mathbf{S}_\theta(\mathbf{x}_t, t)$

Let us first compute the spatial derivative of the conditional score function.

$$(\mathbf{a} \cdot \nabla_{\mathbf{x}_t})(-\sqrt{\nu_t}\nabla_{\mathbf{x}_t} \log p(\mathbf{x}_t, t \mid \mathbf{x}_0, 0)) = \left(\sum_i a_i \partial_{x_t{}^i}\right) \frac{\mathbf{x}_t - \sqrt{1-\nu_t}\mathbf{x}_0}{\sqrt{\nu_t}}$$

---

[4]In general, $(\mathbf{a} \cdot \nabla)^2 = (\sum_i a_i \partial_i)^2 = (\sum_i a_i \partial_i)(\sum_j a_j \partial_j) = \sum_i a_i \sum_j (\partial_i a_j + a_j \partial_i \partial_j)$. If $\mathbf{a}$ is separable in each dimension, the $\partial_i a_j (i \neq j)$ terms vanish, and $(\mathbf{a} \cdot \nabla)^2 = \sum_i (a_i \partial_i a_i + \sum_j a_i a_j \partial_i \partial_j)$. If the function $\boldsymbol{\psi}(\mathbf{x}_t, t)$ is separable in each dimension, then $(\mathbf{a} \cdot \nabla)^2 \psi_k = \sum_i (a_i \partial_i a_i + \sum_j a_i a_j \partial_i \partial_j)\psi_k = (a_k \partial_k a_k + a_k^2 \partial_k^2)\psi_k$. Thus we can formally write $(\mathbf{a} \cdot \nabla)^2 \boldsymbol{\psi} = (\mathbf{a} \odot \nabla \odot \mathbf{a} + \mathbf{a} \odot \mathbf{a} \odot \nabla \odot \nabla) \odot \boldsymbol{\psi} = \mathbf{a} \odot (\nabla \odot \mathbf{a} + \mathbf{a} \odot \nabla \odot \nabla) \odot \boldsymbol{\psi} = \mathbf{a} \odot \nabla \odot (\mathbf{a} \odot \nabla) \odot \boldsymbol{\psi} = (\mathbf{a} \odot \nabla\odot)^2 \boldsymbol{\psi} = (\mathbf{a} \odot \partial_\mathbf{x})^2 \boldsymbol{\psi}$. (Note that the operator $(\mathbf{a} \cdot \nabla)$ is scalar while $(\mathbf{a} \odot \partial_\mathbf{x})$ is $d$-dim vector.) We can similarly show $(\mathbf{a} \cdot \nabla)^k \boldsymbol{\psi} = (\mathbf{a} \odot \partial_\mathbf{x})^k \boldsymbol{\psi}$ for $k \geq 3$.

$$
= \frac{1}{\sqrt{\nu_t}} \begin{bmatrix} \left(\sum_i a_i \partial_{x_t{}^i}\right) (\mathbf{x}_t - \sqrt{1 - \nu_t}\mathbf{x}_0)^1 \\ \vdots \\ \left(\sum_i a_i \partial_{x_t{}^i}\right) (\mathbf{x}_t - \sqrt{1 - \nu_t}\mathbf{x}_0)^d \end{bmatrix}
$$

$$
= \frac{1}{\sqrt{\nu_t}} \begin{bmatrix} \left(\sum_i a_i \partial_{x_t{}^i}\right) (x_t{}^1 - \sqrt{1 - \nu_t}x_0{}^1) \\ \vdots \\ \left(\sum_i a_i \partial_{x_t{}^i}\right) (x_t{}^d - \sqrt{1 - \nu_t}x_0{}^d) \end{bmatrix}
$$

$$
= \frac{1}{\sqrt{\nu_t}} \begin{bmatrix} \left(a_1 \partial_{x_t{}^1}\right) (x_t{}^1 - \sqrt{1 - \nu_t}x_0{}^1) \\ \vdots \\ \left(a_d \partial_{x_t{}^d}\right) (x_t{}^d - \sqrt{1 - \nu_t}x_0{}^d) \end{bmatrix}
$$

$$
= \frac{1}{\sqrt{\nu_t}} \begin{bmatrix} a_1 \\ \vdots \\ a_d \end{bmatrix} = \frac{1}{\sqrt{\nu_t}}\mathbf{a} = \mathbf{a} \odot \frac{1}{\sqrt{\nu_t}}\mathbf{1}. \tag{50}
$$

Here, we used the notation $x_t{}^i$ to denotes the $i$-th component of a vector $\mathbf{x}_t$. Note that up to this point in the discussion, there have been no approximations, but strict ones.

Now let us consider the approximation. Because of the single point approximation, we may assume that the derivative of the integrated score function will also be approximated by the derivative of the conditional score function, i.e.,

$$
(\mathbf{a} \cdot \nabla_{\mathbf{x}_t})(-\sqrt{\nu_t}\nabla_{\mathbf{x}_t} \log p(\mathbf{x}_t, t)) \approx (\mathbf{a} \cdot \nabla_{\mathbf{x}_t})(-\sqrt{\nu_t}\nabla_{\mathbf{x}_t} \log p(\mathbf{x}_t, t \mid \mathbf{x}_0, 0)). \tag{51}
$$

As the neural network $\mathbf{S}_\theta(\mathbf{x}_t, t)$ is trained so that it approximates the integrated score function, we can also assume the following relation,

$$
(\mathbf{a} \cdot \nabla_{\mathbf{x}_t})\mathbf{S}_\theta(\mathbf{x}_t, t) \approx (\mathbf{a} \cdot \nabla_{\mathbf{x}_t})(-\sqrt{\nu_t}\nabla_{\mathbf{x}_t} \log p(\mathbf{x}_t, t \mid \mathbf{x}_0, 0)) = \frac{1}{\sqrt{\nu_t}}\mathbf{a}. \tag{52}
$$

Thus we have obtained the ideal spatial derivative of the neural network.

We can also formally write the spatial derivative as follows using the above notation,

$$
\mathbf{a} \odot (\partial_{\mathbf{x}_t}\mathbf{S}_\theta(\mathbf{x}_t, t)) = \mathbf{a} \odot \frac{1}{\sqrt{\nu_t}}\mathbf{1}. \tag{53}
$$

We can also write it as

$$
\partial_{\mathbf{x}_t}\mathbf{S}_\theta(\mathbf{x}_t, t) = \frac{1}{\sqrt{\nu_t}}\mathbf{1}. \tag{54}
$$

### B.1.2 Time Derivative $-\partial_t\mathbf{S}_\theta(\mathbf{x}_t, t)$

Next, let us compute $-\partial_t(-\sqrt{\nu_t}\nabla_{\mathbf{x}_t} \log p(\mathbf{x}_t, t \mid \mathbf{x}_0, 0))$. During the computation, $\mathbf{x}_0$ is replaced by the relation

$$
\mathbf{x}_0 = \frac{1}{\sqrt{1 - \nu_t}} (\mathbf{x}_t + \nu_t\nabla_{\mathbf{x}_t} \log p(\mathbf{x}_t, t \mid \mathbf{x}_0, 0)). \tag{55}
$$

We also use the following relations between $\nu_t, \beta_t$, which is immediately obtained from the definition of $\nu_t$,

$$
\dot{\nu}_t = (1 - \nu_t)\beta_t. \tag{56}
$$

Using the above information, we may compute the temporal derivative of the conditional score function as follows.

$$
-\partial_t(-\sqrt{\nu_t}\nabla_{\mathbf{x}_t} \log p(\mathbf{x}_t, t \mid \mathbf{x}_0, 0))
$$

$$
= -\partial_t \frac{\mathbf{x}_t - \sqrt{1 - \nu_t}\mathbf{x}_0}{\sqrt{\nu_t}}
$$

$$= -\frac{1}{\sqrt{\nu_t}} \left( \frac{1}{2}\dot{\nu}_t(1-\nu_t)^{-1/2}\mathbf{x}_0 \right) - (\mathbf{x}_t - \sqrt{1-\nu_t}\mathbf{x}_0) \left( -\frac{1}{2}\dot{\nu}_t\nu_t^{-3/2} \right)$$

$$= -\frac{\dot{\nu}_t}{2\nu_t^{3/2}} \left( \frac{\nu_t}{\sqrt{1-\nu_t}}\mathbf{x}_0 - (\mathbf{x}_t - \sqrt{1-\nu_t}\mathbf{x}_0) \right)$$

$$= -\frac{\dot{\nu}_t}{2\nu_t^{3/2}} \left( -\mathbf{x}_t + \frac{1}{\sqrt{1-\nu_t}}\mathbf{x}_0 \right)$$

$$= -\frac{\dot{\nu}_t}{2\nu_t^{3/2}} \left( -\mathbf{x}_t + \frac{1}{\sqrt{1-\nu_t}} \frac{1}{\sqrt{1-\nu_t}} \left( \mathbf{x}_t + \nu_t\nabla_{\mathbf{x}_t}\log p(\mathbf{x}_t, t \mid \mathbf{x}_0, 0) \right) \right)$$

$$= -\frac{\dot{\nu}_t}{2\nu_t^{3/2}} \left( \left( -1 + \frac{1}{1-\nu_t} \right)\mathbf{x}_t + \frac{1}{1-\nu_t} \left( \nu_t\nabla_{\mathbf{x}_t}\log p(\mathbf{x}_t, t \mid \mathbf{x}_0, 0) \right) \right)$$

$$= -\frac{1}{2\nu_t^{3/2}} \frac{\dot{\nu}_t}{1-\nu_t} \left( \nu_t\mathbf{x}_t + \nu_t\nabla_{\mathbf{x}_t}\log p(\mathbf{x}_t, t \mid \mathbf{x}_0, 0) \right)$$

$$= -\frac{1}{2\nu_t^{3/2}}\beta_t \left( \nu_t\mathbf{x}_t + \nu_t\nabla_{\mathbf{x}_t}\log p(\mathbf{x}_t, t \mid \mathbf{x}_0, 0) \right)$$

$$= -\frac{\beta_t}{2\sqrt{\nu_t}} \left( \mathbf{x}_t + \nabla_{\mathbf{x}_t}\log p(\mathbf{x}_t, t \mid \mathbf{x}_0, 0) \right). \tag{57}$$

(Note that this calculation is exact, and no approximation is injected.) Because of the single point approximation, we may assume

$$-\partial_t(-\sqrt{\nu_t}\nabla_{\mathbf{x}_t}\log p(\mathbf{x}_t, t)) \approx -\partial_t(-\sqrt{\nu_t}\nabla_{\mathbf{x}_t}\log p(\mathbf{x}_t, t \mid \mathbf{x}_0, 0))$$

$$= -\frac{\beta_t}{2\sqrt{\nu_t}} \left( \mathbf{x}_t + \nabla_{\mathbf{x}_t}\log p(\mathbf{x}_t, t \mid \mathbf{x}_0, 0) \right)$$

$$\approx -\frac{\beta_t}{2\sqrt{\nu_t}} \left( \mathbf{x}_t + \nabla_{\mathbf{x}_t}\log p(\mathbf{x}_t, t) \right), \tag{58}$$

and therefore, we can also assume that the temporal derivative of the neural network is approximated as

$$-\partial_t\mathbf{S}_\theta(\mathbf{x}_t, t) \approx -\frac{\beta_t}{2\sqrt{\nu_t}} \left( \mathbf{x}_t - \frac{1}{\sqrt{\nu_t}}\mathbf{S}_\theta(\mathbf{x}_t, t) \right). \tag{59}$$

The "derivatives" have some good points. For example, the partial derivatives commute,

$$\partial_{\mathbf{x}_t}\partial_t\mathbf{S}_\theta(\mathbf{x}_t, t) = \partial_t\partial_{\mathbf{x}_t}\mathbf{S}_\theta(\mathbf{x}_t, t). \tag{60}$$

## B.2 Comparison of the Empirical Score Derivatives and Ideal Derivatives

Let us empirically validate that idela approximation using real data similarly as above. However, since the equations will become very complicated if we evaluate the exact empirical score derivatives, we instead used finite differences as the ground truths. That is, let $S(\mathbf{x}, t)$ be the routine that computes the empirical score function as follows,

$$S(\mathbf{x}, t) = -\frac{\sqrt{\nu_t}}{|\mathcal{D}|} \sum_{\mathbf{x}_0 \in \mathcal{D}} [q(\mathbf{x}_0 \mid \mathbf{x}_t) \nabla \log p(\mathbf{x}_t, t \mid \mathbf{x}_0, 0)], \tag{61}$$

and we evaluated the empirical score derivatives by the finite differences as follows[5],

$$\text{Empirical } t \text{ Deriv:} \qquad \partial_t S \approx \frac{S(\mathbf{x}_t, t + \varepsilon) - S(\mathbf{x}_t, t)}{\varepsilon} \tag{62}$$

$$\text{Empirical } \mathbf{x}_t \text{ Deriv:} \qquad (\mathbf{a} \cdot \nabla_{\mathbf{x}_t}) S \approx \frac{S(\mathbf{x}_t + \varepsilon \mathbf{a}, t) - S(\mathbf{x}_t, t)}{\varepsilon}, \text{ where } \mathbf{a} \sim \mathcal{N}(\mathbf{0}, \mathbf{I}). \tag{63}$$

where $\varepsilon$ should be a sufficiently small value, and we used $\varepsilon = 10^{-3}$ here. We compared these empirical derivatives with the ideal derivatives using MNIST and CIFAR-10.

$$\text{Ideal } t \text{ Deriv:} \qquad \partial_t \mathbf{S}_\theta = \frac{\beta_t}{2\sqrt{\nu_t}} \left( \mathbf{x}_t - \frac{1}{\sqrt{\nu_t}} \mathbf{S}_\theta(\mathbf{x}_t, t) \right) = \frac{\beta_t}{2\sqrt{\nu_t}} \left( \mathbf{x}_t - \frac{\mathbf{x}_t - \sqrt{1 - \nu_t}\mathbf{x}_0}{\nu_t} \right)$$

$$\text{Ideal } \mathbf{x}_t \text{ Deriv:} \qquad (\mathbf{a} \cdot \nabla_{\mathbf{x}_t}) \mathbf{S}_\theta = \frac{1}{\sqrt{\nu_t}} \mathbf{a}$$

As the ideal derivatives require the specific function forms of diffusion and variance schedules, we tested on following two noise schedules.

**Linear schedule** We first tested on the linear schedule eq. (76), where $\beta_0 = 0.1$ and $\beta_1 = 9.95$. This is the same schedule as the one used in the main text.

Figure 9 shows the relative $L_2$ error and the cosine similarity between the ideal $t$ derivative eq. (21) and the empirical $t$ derivative eq. (62), in which it is observed that they are very close when $0 \lesssim t \lesssim 0.5$, while the approximation accuracy decreases as $t$ increases. However, even in that case, there tends to be an overall positive correlation. It can also be observed that there is an error that seems to originate from the singularity of time origin when $t \approx 0$. (See also § D.2.)

For the $\mathbf{x}$ derivative (Figure 9), on the other hand, we can confirm that the errors between the ideal $\mathbf{x}$ derivative eq. (21) and empirical $\mathbf{x}$ derivative eq. (62) are generally very highly correlated, except around $t \approx 0.5$.

**Modified tanh schedule** We also tested on another noise schedule, the modified tanh schedule eq. (79) which does not have the singularity at the time origin. The parameters $A, k$ were determined so that $\nu_0 = 0.001$ and $\nu_1 = 0.999$. Figure 11 and Figure 12 show the results. In this case, the overall trend is similar to the linear schedule, but we can observe that the singularity of the time origin of the $t$ derivative is eliminated.

---

[5]To verify the empirical $\mathbf{x}_t$ derivative, let us consider a simple case of three-variable function $f(x, y, z)$. As its total derivative is $df = \partial_x f dx + \partial_y f dy + \partial_z f dz$, we have $f(x + a, y + b, z + c) - f(x, y, z) = (a\partial_x + b\partial_y + c\partial_z)f(x, y, z)$ for small $a, b, c$. Let $a = \varepsilon a', b = \varepsilon b'$ and $c = \varepsilon c'$, then $f(x + \varepsilon a', y + \varepsilon b', z + \varepsilon c') - f(x, y, z) = \varepsilon(a'\partial_x + b'\partial_y + c'\partial_z)f(x, y, z)$. Therefore, we can write the spatial derivative as $(a'\partial_x + b'\partial_y + c'\partial_z)f(x, y, z) = \lim_{\varepsilon \to 0} \frac{1}{\varepsilon}(f(x + \varepsilon a', y + \varepsilon b', z + \varepsilon c') - f(x, y, z))$.

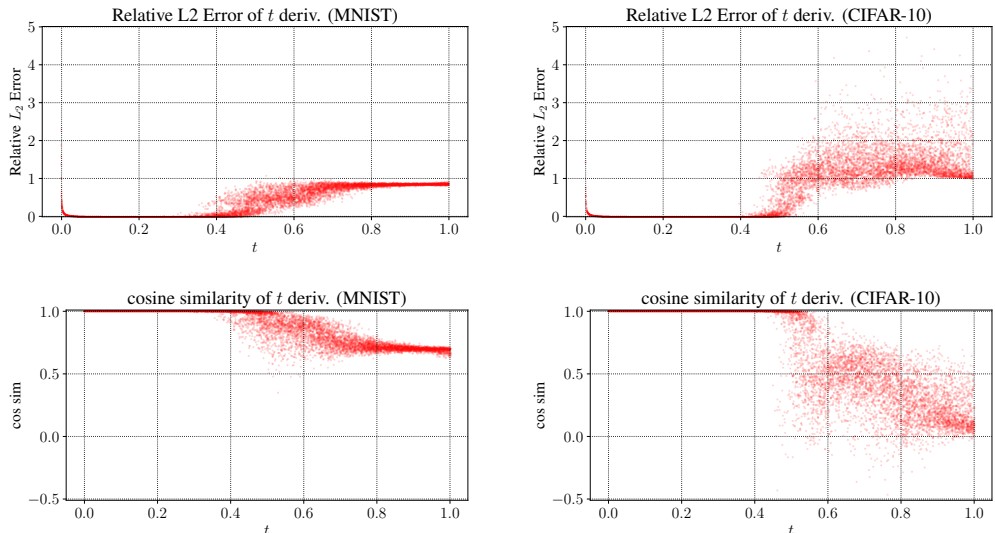

Figure 9: Relative $L_2$ Error ($\downarrow$) and Cosine similarity ($\uparrow$) between the empirical $t$ derivative and the ideal $t$ derivative. (Linear schedule.)

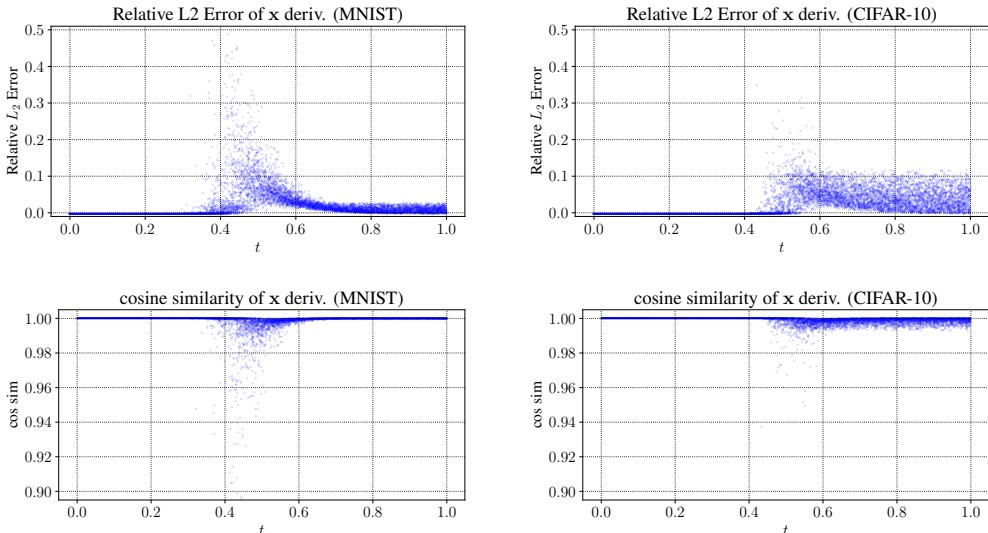

Figure 10: Relative $L_2$ Error ($\downarrow$) and Cosine similarity ($\uparrow$) between the empirical $\mathbf{x}$ derivative and the ideal $\mathbf{x}$ derivative. (Linear schedule.)

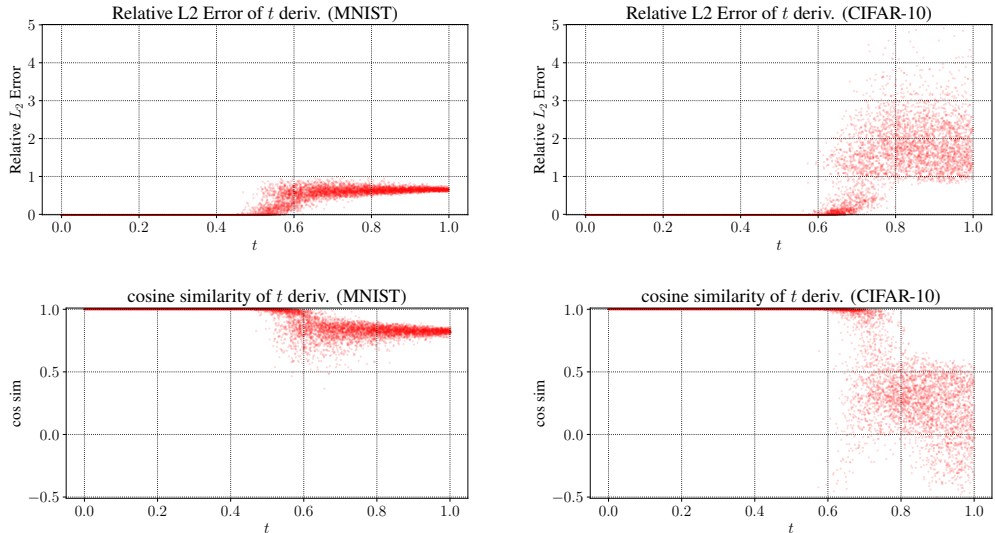

Figure 11: Relative $L_2$ Error ($\downarrow$) and Cosine similarity ($\uparrow$) between the empirical $t$ derivative and the ideal $t$ derivative. (Modified tanh schedule.)

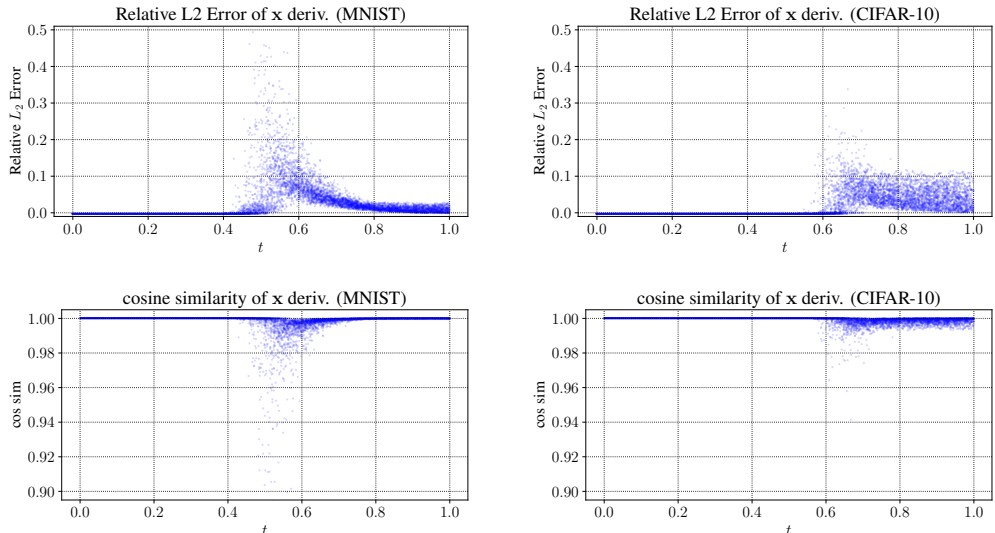

Figure 12: Relative $L_2$ Error ($\downarrow$) and Cosine similarity ($\uparrow$) between the empirical $\mathbf{x}$ derivative and the ideal $\mathbf{x}$ derivative. (Modified tanh schedule.)

## B.3 The Derivatives $L_\flat(-\bar{\mathbf{f}}_\flat)$, $L_\sharp(-\bar{\mathbf{f}}_\sharp)$, $L_\sharp(g)$, $G_\sharp(-\bar{\mathbf{f}}_\sharp)$, $G_\sharp(g)$

The computation of the derivative $L_\flat(-\bar{\mathbf{f}}_\flat)$, $L_\sharp(-\bar{\mathbf{f}}_\sharp)$, $L_\sharp(g)$, $G_\sharp(-\bar{\mathbf{f}}_\sharp)$, $G_\sharp(g)$ does not require any particular nontrivial process. All we have to do is rewrite a term every time we encounter a derivative of $\mathbf{S}_\theta(x_t, t)$ or $\nu_t$, and the rest is at the level of elementary exercises in introductory calculus. To execute this symbolic computation, the use of computer algebra systems will be a good option. It should be noted, however, that some implementation tricks to process such custom derivatives are required (in other words, the term-rewriting system should be customized).

The results are shown below. Although these expressions appear complex at first glance, the code generation system can automatically generate code for such expressions.

$$L_\flat(-\bar{\mathbf{f}}_\flat)(x_t, t) = \left( \frac{\beta_t^2}{4} - \frac{\dot{\beta}_t}{2} \right) x_t + \left( \frac{\dot{\beta}_t}{2\sqrt{\nu_t}} - \frac{\beta_t^2}{4\nu_t^{3/2}} \right) \mathbf{S}_\theta(x_t, t) \tag{64}$$

$$L_\sharp(-\bar{\mathbf{f}}_\sharp)(x_t, t) = \left( \frac{\beta_t^2}{4} - \frac{\dot{\beta}_t}{2} \right) x_t + \frac{\dot{\beta}_t}{\sqrt{\nu_t}} \mathbf{S}_\theta(x_t, t) \tag{65}$$

$$G_\sharp(-\bar{\mathbf{f}}_\sharp)(x_t, t) = \left( \frac{1}{2} - \frac{1}{\nu_t} \right) \beta_t^{3/2} \tag{66}$$

$$L_\sharp g(t) = -\frac{\dot{\beta}_t}{2\sqrt{\beta_t}} \tag{67}$$

$$G_\sharp g(t) = 0. \tag{68}$$

We may also compute higher order derivatives, though we do not use them in this paper except $L_\flat L_\flat(-\bar{\mathbf{f}}_\flat)$,

$$L_\flat L_\flat(-\bar{\mathbf{f}}_\flat)(x_t, t) = \left( \frac{\beta_t^3}{8} - \frac{3\beta_t\dot{\beta}_t}{4} + \frac{\ddot{\beta}_t}{2} \right) x_t$$
$$+ \left( \frac{\beta_t^3(-\nu_t^2 + 3\nu_t - 3)}{8\nu_t^{5/2}} + \frac{3\beta_t\dot{\beta}_t}{4\nu_t^{3/2}} - \frac{\ddot{\beta}_t}{2\sqrt{\nu_t}} \right) \mathbf{S}_\theta(x_t, t) \tag{69}$$

$$L_\sharp L_\sharp(-\bar{\mathbf{f}}_\sharp)(x_t, t) = \left( \frac{\beta_t^3}{8} - \frac{3\beta_t\dot{\beta}_t}{4} + \frac{\ddot{\beta}_t}{2} \right) x_t - \frac{\beta_t^3 + 4\ddot{\beta}_t}{4\sqrt{\nu_t}} \mathbf{S}_\theta(x_t, t)$$

$$L_\sharp G_\sharp(-\bar{\mathbf{f}}_\sharp)(x_t, t) = \frac{\sqrt{\beta_t}}{\nu_t^2} \left( \frac{\nu_t(2\beta_t^2 + 3\dot{\beta}_t)}{2} - \beta_t^2 - \frac{3\nu_t^2\dot{\beta}_t}{4} \right)$$

$$G_\sharp L_\sharp(-\bar{\mathbf{f}}_\sharp)(x_t, t) = \sqrt{\beta_t} \left( \frac{\beta_t^2}{4} - \frac{\dot{\beta}_t}{2} + \frac{\dot{\beta}_t}{\nu_t} \right)$$

$$G_\sharp G_\sharp(-\bar{\mathbf{f}}_\sharp)(x_t, t) = 0$$

$$L_\sharp L_\sharp g(t) = \frac{2\beta_t\ddot{\beta}_t - \dot{\beta}_t^2}{4\beta_t^{3/2}}$$

$$L_\sharp G_\sharp g(t) = 0$$
$$G_\sharp L_\sharp g(t) = 0$$
$$G_\sharp G_\sharp g(t) = 0.$$

As we can see, no factors other than integers, $x_t$, $\mathbf{S}_\theta(x_t, t)$, $\nu_t$, $\beta_t$ and derivatives of $\beta_t$ appear. This is also true for higher order derivatives, which can be easily shown.

**SymPy Code Snippet for Automatic Symbolic Computation of Derivatives** The following code snippet is a minimalistic example of SymPy code to compute the above derivatives using the customized derivative method. We used SymPy 1.11 to test the following code snippet.

```python
from sympy import Function, symbols, sqrt, simplify

x, t = symbols('x t') # x, t
B = Function('beta') # β_t

# define customized derivatives of ν_t
class nu(Function):
    def fdiff(self, argindex=1):
        t, = self.args
        return (1 - nu(t)) * B(t) # ν̇_t = (1 − ν_t)β_t

# define customized derivatives of S_θ(x, t)
class S_theta(Function):
    def fdiff(self, argindex=1):
        x, t = self.args
        if argindex == 1: # ∂/∂x
            d = 1 / sqrt(nu(t))
        elif argindex == 2: # ∂/∂t
            d = (x - S_theta(x, t)/sqrt(nu(t))) * B(t) / (2 * sqrt(nu(t)))
        return d

# define f̄_♭
class f_flat(Function):
    @classmethod
    def eval(cls, x, t):
        return - B(t) * x / 2 + S_theta(x, t) * B(t) / (2 * sqrt(nu(t)))

# define differential operator L_♭
class L_flat(Function):
    @classmethod
    def eval(cls, fxt):
        return -fxt.diff(t) - f_flat(x, t) * fxt.diff(x)

# show each derivative
print(f_flat(x, t))
print(simplify(L_flat(f_flat(x,t)))) # L_♭f̄_♭(x_t,t); see eq. (64)
print(simplify(L_flat(L_flat(f_flat(x,t))))) # L_♭L_♭f̄_♭(x_t,t); see eq. (69),

# we can similarly define f̄_♯, L_♯, G_♯ and compute other derivatives.
```

The result will look like

$$[\text{Out 1}] \quad -\frac{x\beta(t)}{2} + \frac{S_\theta(x,t)\beta(t)}{2\sqrt{\nu(t)}}$$

$$[\text{Out 2}] \quad -\frac{x\beta^2(t)}{4} + \frac{x\frac{d}{dt}\beta(t)}{2} + \frac{S_\theta(x,t)\beta^2(t)}{4\nu^{\frac{3}{2}}(t)} - \frac{S_\theta(x,t)\frac{d}{dt}\beta(t)}{2\sqrt{\nu(t)}}$$

$$[\text{Out 3}] \quad -\frac{x\beta^3(t)}{8} + \frac{3x\beta(t)\frac{d}{dt}\beta(t)}{4} - \frac{x\frac{d^2}{dt^2}\beta(t)}{2} + \frac{S_\theta(x,t)\beta^3(t)}{8\sqrt{\nu(t)}}$$

$$- \frac{3S_\theta(x,t)\beta^3(t)}{8\nu^{\frac{3}{2}}(t)} + \frac{3S_\theta(x,t)\beta^3(t)}{8\nu^{\frac{5}{2}}(t)} - \frac{3S_\theta(x,t)\beta(t)\frac{d}{dt}\beta(t)}{4\nu^{\frac{3}{2}}(t)} + \frac{S_\theta(x,t)\frac{d^2}{dt^2}\beta(t)}{2\sqrt{\nu(t)}}$$

and so on. Some additional coding techniques can further improve the readability of these expressions, but there will be no need to go any deeper into such subsidiary issues here.

Thus obtained symbolic expressions can be automatically converted into executable code in practical programming languages including Python and C++ using a code generator, though the authors hand-coded the obtained expressions in Python for the experiments in this paper.

## C  TRUNCATED DDIM IS EQUIVALENT TO THE QUASI-TAYLOR SAMPLER

Using SymPy, we can easily compute the Taylor expansion of a given function. For example, the following code

```
sympy.series(B(t+h), h, 0, 4)
```

yields the result like

$$\beta(t) + h \left.\frac{d}{d\xi_1}\beta(\xi_1)\right|_{\xi_1=t} + \frac{h^2 \left.\frac{d^2}{d\xi_1^2}\beta(\xi_1)\right|_{\xi_1=t}}{2} + \frac{h^3 \left.\frac{d^3}{d\xi_1^3}\beta(\xi_1)\right|_{\xi_1=t}}{6} + O\left(h^4\right).$$

Similarly, using the relation $\dot{\nu}_t = (1 - \nu_t)\beta_t$, we can easily compute the Taylor expansion of $\nu_{t-h}$ as follows.

```
sympy.series(nu(t-h), h, 0, 3)
```

$$\nu_{t-h} = \nu(t) + h\left(\beta(t)\nu(t) - \beta(t)\right) + h^2 \left(\frac{\beta^2(t)\nu(t)}{2} - \frac{\beta^2(t)}{2} - \frac{\nu(t)\left.\frac{d}{d\xi_1}\beta(\xi_1)\right|_{\xi_1=t}}{2} + \frac{\left.\frac{d}{d\xi_1}\beta(\xi_1)\right|_{\xi_1=t}}{2}\right) + O\left(h^3\right)$$

Using this functionality of SymPy, we can easily compute the Taylor expansion of the DDIM (Song et al., 2020a). Let us recall that the DDIM algorithm is given by eq. (15), and using our notation $\alpha = \sqrt{1-\nu}$ and $\sigma = \sqrt{\nu}$, it can be written as follows,

$$\text{DDIM:} \quad \mathbf{x}_{t-h} \leftarrow \underbrace{\sqrt{\frac{1-\nu_{t-h}}{1-\nu_t}}}_{=:\rho_{t,h}^{\text{DDIM}}} \mathbf{x}_t + \underbrace{\left(\sqrt{\nu_{t-h}} - \sqrt{\frac{1-\nu_{t-h}}{1-\nu_t}}\nu_t\right)}_{=:\mu_{t,h}^{\text{DDIM}}} \mathbf{S}_\theta(\mathbf{x}_t, t).$$

Then using SymPy, the Taylor expansion of $\rho_{t,h}^{\text{DDIM}}$ and $\mu_{t,h}^{\text{DDIM}}$ are computed as follows,

$$\rho_{t,h}^{\text{DDIM}} = 1 + \frac{\beta_t}{2}h - \frac{h^2}{4}\left(\frac{\beta_t^2}{2} - \dot{\beta}_t\right) + \frac{h^3}{4}\left(\frac{\beta_t^3}{12} - \frac{\beta_t\dot{\beta}_t}{2} + \frac{\ddot{\beta}_t}{3}\right) + o(h^3), \tag{70}$$

$$\sqrt{\nu_t}\mu_{t,h}^{\text{DDIM}} = -\frac{\beta_t}{2}h + \frac{h^2}{4}\left(\dot{\beta}_t - \frac{\beta_t^2}{2\nu_t}\right) + \frac{h^3}{4}\left(-\frac{\beta_t^3}{12} + \frac{\beta_t^3}{4\nu_t} - \frac{\beta_t^3}{4\nu_t^2} + \frac{\beta_t\dot{\beta}_t}{2\nu_t} - \frac{\ddot{\beta}_t}{3}\right) + o(h^3). \tag{71}$$

Although it has been known that DDIM corresponds to the Euler method up to 1st order terms (Song et al., 2020a; Salimans & Ho, 2022), this expansion gives better understanding of higher order terms. That is, these are exactly equivalent to our deterministic Quasi-Taylor sampler eq. (23) and eq. (24) up to 3rd-order terms. This fact may suggest that the assumptions behind the DDIM derivation will be logically equivalent to our assumptions of ideal derivatives.

The advantage of the proposed Quasi-Taylor method is that we can decide the hyperparameter at which order the Taylor expansion is truncated. On the other hand, DDIM automatically incorporates terms of much higher order, leaving no room for order tuning.

# D ON THE NOISE SCHEDULE

## D.1 BACKGROUND: PICARD-LINDELÖF THEOREM

Let us consider a 1-dim deterministic system $\dot{x}(t) = a(x(t), t)$. It is well known that this ODE has a unique solution if $a(x, t)$ is Lipschitz continuous w.r.t. $x$ and continuous w.r.t. $t$ (Picard-Lindelöf Theorem). Otherwise, ODEs often behave less favorably. (Similar Lipschitz conditions are also required for SDEs.)

**Example 1.** *For example, the ODE $\dot{x} = x^2, x(0) = 1$ has the solution $x = 1/(1 - t)$ when $t < 1$, and it blows up at $t = 1$. It is usually impossible to consider what happens after $t > 1$ in ordinary contexts.*

**Example 2.** *Another well-known example is $\dot{x} = \sqrt{x}, x(0) = 0$. It has a solution $x = t^2/4$, but $x \equiv 0$ is also a solution. It actually has infinitely many solutions $x = 0$ (if $t \leq t_0$), $x = (t - t_0)^2/4$ (if $t > t_0$), where $t_0 \geq 0$ is an arbitrary constant.*

**Example 3.** *Let us consider the following ODE*

$$\dot{x} = -\frac{t - 1}{1 - e^{-(t-1)^2}} x, \quad x(0) = 1, \tag{72}$$

*which is a simplified model of the Linear schedule eq. (76). The exact solution is as follows,*

$$x = \frac{\sqrt{e - 1}}{\sqrt{e^{(t-1)^2} - 1}}, \tag{73}$$

*which diverges at $t = 1$. In this case, $a(x, t) = -x \cdot (t - 1)/(1 - e^{-(t-1)^2})$ is not Lipschitz continuous, as the Taylor expansion of the denominator is $1 - e^{-(t-1)^2} = (t - 1)^2 + O((t - 1)^4)$, and $a(x, t)$ is approximately $-x/(t - 1)$ near $t = 1$.*

In these cases, the coefficient $a(\cdot, \cdot)$ is not Lipschitz continuous. Even these seemingly simplest ODEs behave very complexly unless the coefficients are carefully designed.

In PF-ODE, the Lipschitz condition is written as follows,

$$\text{Lip}(\bar{\mathbf{f}}_\flat) = \left| \partial_{x_t} \left( \frac{\beta_t}{2} x_t - \frac{\beta_t}{2\sqrt{\nu_t}} \mathbf{S}_\theta(x_t, t) \right) \right| < \infty. \tag{74}$$

Using the ideal derivative of $\mathbf{S}_\theta(x_t, t)$, this condition translates as

$$\text{Lip}(\bar{\mathbf{f}}_\flat) = |\beta_t(1 - 1/\nu_t)| = \left| \frac{\dot{\nu}_t}{\nu_t} \right| < \infty. \tag{75}$$

## D.2 SPECIFIC SCHEDULES

Including this point, the necessary conditions for a variance schedule $\nu_t$ will be summarized as follows.

1. $\nu_0 \approx 0$ so that the initial density $p(\mathbf{x}_0, 0)$ is close to the true data density.
2. $\nu_T \approx 1$ so that the terminal density $p(\mathbf{x}_T, T)$ is close to the Gaussian.
3. Sufficiently smooth so that $\beta_t = -\frac{d}{dt} \log(1 - \nu_t)$ is well defined.
   - In addition, $\beta_t$ should also be smooth so that the Taylor schemes can be used.
4. Monotonic ($s < t \implies \nu_s \leq \nu_t$) to make $\beta_t$ non-negative.
5. Preferably, make the drift coefficient $\bar{\mathbf{f}}_\flat$ Lipschitz continuous so that PF-ODE has a unique solution, i.e., $\text{Lip}(\bar{\mathbf{f}}_\flat) \approx |\dot{\nu}_t/\nu_t| < \infty$.

The following two scheduling functions which are common in diffusion generative models satisfy the conditions 1, 2, 4 above (the linear schedule also satisfies the 3rd condition),

$$\text{Linear:} \quad \nu_t = 1 - e^{-\beta_0 t - \beta_1 t^2}, \quad \beta_t = \beta_0 + 2\beta_1 t, \tag{76}$$

$$\text{Cosine:} \quad \nu_t = 1 - C\cos^2\left(\frac{\pi}{2}\frac{t/T + \varsigma}{1 + \varsigma}\right), \quad \beta_t = \begin{cases} \frac{\pi}{T}\tan\left(\frac{\pi}{2}\frac{t/T + \varsigma}{1 + \varsigma}\right) & \text{if } 0 \leq t \leq T' \\ \Theta & \text{if } T' < t \leq T \end{cases}. \tag{77}$$

where $\varsigma > 0$ is a small constant, $C = 1/\cos^2(\pi\varsigma/2(1+\varsigma))$ is a constant to make $\nu_0 = 0$, and the threshold constant is $\Theta = \beta_{T'}$.

However, these common schedules do not satisfy the 5th condition that the drift coefficient $\bar{\mathbf{f}}_\flat$ is Lipschitz continuous. Indeed, it is easily verified that $\lim_{t\to 0}\dot{\nu}_t/\nu_t = \infty$ in both cases, since $\nu_0 = 0$ but $\dot{\nu}_0 > 0$. Nevertheless, $t = 0$ is the only singular point, and since no function value or derivative at $t = 0$ is evaluated by numerical methods (except by the Runge-Kutta method), this point can practically be ignored.

Note that, we can also consider some other schedule functions such as the sigmoid function and the hyperbolic tangent, which satisfy the condition 2, 3, 4, 5 but do not satisfy the 1st condition rigorously (but if $\nu_0$ is less than or equal to the level of the quantization error in the data, we may consider the first condition to be essentially satisfied),

$$\text{Sigmoid:} \qquad \nu_t = \frac{1}{1 + e^{-A(t-k)}}, \qquad \beta_t = A\nu_t, \qquad (78)$$

$$\text{Modified Tanh:} \qquad \nu_t = \tanh^2(\lambda(t)/2), \qquad \beta_t = \dot{\lambda}(t)\tanh(\lambda(t)/2), \qquad (79)$$

where the parameter function $\lambda(t)$ has some options, such as $\lambda(t) = \log(1 + Ae^{kt})$, and $A > 0$, $k > 0$ are hyperparameters.

## D.3  HOW TO AVOID THE TIME ORIGIN SINGULARITY IN THE RUNGE-KUTTA METHODS

When using the Heun and Classical RK4 methods, the function $\bar{\mathbf{f}}_\flat(\mathbf{x}_t, t)$ is evaluated at time $t = 0$. However, since the function $\bar{\mathbf{f}}_\flat(\mathbf{x}_t, t)$ contains the term proportional to $1/\sqrt{\nu_t}$, it will diverge at time $t = 0$ if the linear eq. (76) or cosine schedule eq. (77) is used. The simplest way to avoid this is to replace the function $\bar{\mathbf{f}}_\flat(\mathbf{x}_0, 0)$ with $\bar{\mathbf{f}}_\flat(\mathbf{x}_\varepsilon, \varepsilon)$ where $\varepsilon > 0$ is a sufficiently small constant, only when the need to evaluate the function at time $t = 0$ arises.

The same thing could happen at $t = T$ if the cosine schedule and DDIM were used simultaneously, but this can be handled in the same way.

If we use the sigmoid eq. (78) or modified tanh schedules, eq. (79) these problems do not occur unless the hyperparameters $A$ and $k$ are chosen to be very extreme values.

# E    Supplement on Fundamentals

For convenience, let us summarize some basics behind the ideas in this paper. The contents of this section are not particularly novel, but the authors expect that this section will give a better understanding of the ideas of this paper and the continuous-time approach to diffusion generative models.

## E.1    Taylor Expansion and Itô-Taylor Expansion

### E.1.1    Taylor Expansion of Deterministic Systems

**1-dimensional case**    Let us first consider a 1-dim deterministic system $\dot{x}(t) = a(x(t), t)$, where $a(\cdot, \cdot)$ is sufficiently smooth, and let us derive the Taylor series expression of the solution of this ODE. Let $\varphi(x(t), t)$ be a differentiable function. Its total derivative is written as

$$d\varphi = \frac{\partial \varphi}{\partial t} dt + \frac{\partial \varphi}{\partial x} dx = \frac{\partial \varphi}{\partial t} dt + \frac{\partial \varphi}{\partial x} \frac{dx}{dt} dt = \left( \frac{\partial \varphi}{\partial t} + \frac{\partial \varphi}{\partial x} a(x, t) \right) dt$$

$$= \underbrace{\left( \frac{\partial}{\partial t} + a(x, t) \frac{\partial}{\partial x} \right)}_{=:L_\flat} \varphi dt. \tag{80}$$

By integrating both sides from $0$ to $t$, we have

$$\varphi(x(t), t) = \varphi(x(0), 0) + \int_0^t (L_\flat \varphi)(x(s), s) ds. \tag{81}$$

We use this formula recursively to obtain the Taylor series of the above system. Let $\varphi(x(t), t) = x(t)$, then we have

$$x(t) = x(0) + \int_0^t (L_\flat x)(x(s), s) ds$$

$$= x(0) + \int_0^t a(x(s), s) ds. \tag{82}$$

Let $\varphi(x(t), t) = a(x(t), t)$, then we have

$$a(x(t), t) = a(x(0), 0) + \int_0^t (L_\flat a)(x(s), s) ds. \tag{83}$$

Using the above two equations, we have

$$x(t) = x(0) + \int_0^t \left( a(x(0), 0) + \int_0^{t_1} (L_\flat a)(x(t_2), t_2) dt_2 \right) dt_1$$

$$= x(0) + \int_0^t a(x(0), 0) dt_1 + \int_0^t \int_0^{t_1} (L_\flat a)(x(t_2), t_2) dt_2 dt_1.$$

$$= x(0) + t a(x(0), 0) + \int_0^t \int_0^{t_1} (L_\flat a)(x(t_2), t_2) dt_2 dt_1. \tag{84}$$

We can expand again the term inside of the integral by using the following relation,

$$(L_\flat a)(x(t), t) = (L_\flat a)(x(0), 0) + \int_0^t (L_\flat^2 a)(x(s), s) ds, \tag{85}$$

and obtain the expansion as follows,

$$x(t) = x(0) + t a(x(0), 0) + \frac{t^2}{2} (L_\flat a)(x(0), 0) + \int_0^t \int_0^{t_1} \int_0^{t_2} (L_\flat^2 a)(x(t_3), t_3) dt_3 dt_2 dt_1. \tag{86}$$

Applying this argument recursively, we can obtain the Taylor series of the deterministic system, if $a(\cdot, \cdot)$ is sufficiently smooth.

$$x(t) = x(0) + t a(x(0), 0) + \sum_{k=2}^\infty \frac{t^k}{k!} (L_\flat^{k-1} a)(x(0), 0). \tag{87}$$

Since the above discussion is valid when the integration interval is $(t, t + h)$ instead of $(0, t)$, it can be written as follows,

$$x(t + h) = x(t) + ha(x(t), t) + \sum_{k=2}^{\infty} \frac{h^k}{k!}(L_\flat^{k-1} a)(x(t), t). \tag{88}$$

**Multi-dimensional case**   Let us consider the multi-dimensional ODE $\dot{\mathbf{x}} = \mathbf{a}(\mathbf{x}, t)$. The total derivative of a smooth scalar function $\varphi(\mathbf{x}, t)$ is written as

$$d\varphi = \frac{\partial \varphi}{\partial t} dt + \sum_i \frac{\partial \varphi}{\partial x_i} dx_i = \frac{\partial \varphi}{\partial t} dt + \sum_i \frac{\partial \varphi}{\partial x_i} \frac{dx_i}{dt} dt = \frac{\partial \varphi}{\partial t} dt + \left(\sum_i a_i(\mathbf{x}, t) \frac{\partial}{\partial x_i}\right) \varphi dt$$

$$= \left(\frac{\partial \varphi}{\partial t} + (\mathbf{a}(\mathbf{x}, t) \cdot \nabla)\varphi\right) dt = \left(\frac{\partial}{\partial t} + \mathbf{a}(\mathbf{x}, t) \cdot \nabla\right) \varphi dt. \tag{89}$$

Let $\boldsymbol{\varphi}(\mathbf{x}, t)$ be a vector-valued smooth function, then we immediately have

$$d\boldsymbol{\varphi} = (\partial_t + \mathbf{a}(\mathbf{x}, t) \cdot \nabla) \, \boldsymbol{\varphi} dt. \tag{90}$$

Using the scalar operator $L_\flat = (\partial_t + \mathbf{a}(\mathbf{x}, t) \cdot \nabla)$, we can obtain the following Taylor expansion similarly to the 1-dim case,

$$\mathbf{x}(t) = \mathbf{x}(0) + t(L_\flat \mathbf{x})\Big|_{t=0} + \frac{t^2}{2}(L_\flat^2 \mathbf{x})\Big|_{t=0} + \cdots$$

$$= \mathbf{x}(0) + t(\partial_t + \mathbf{a}(\mathbf{x}, t) \cdot \nabla)\mathbf{x}\Big|_{t=0} + \frac{t^2}{2}(\partial_t + \mathbf{a}(\mathbf{x}, t) \cdot \nabla)^2 \mathbf{x}\Big|_{t=0} + \cdots$$

$$= \mathbf{x}(0) + t\mathbf{a}(\mathbf{x}(0), 0) + \frac{t^2}{2}(\partial_t + \mathbf{a}(\mathbf{x}, t) \cdot \nabla)\mathbf{a}(\mathbf{x}, t)\Big|_{t=0} + \cdots \tag{91}$$

The second order term $(\mathbf{a} \cdot \nabla)\mathbf{a}$ is written as follows,

$$(\mathbf{a} \cdot \nabla) \mathbf{a} = \left(\sum_i a_i \partial_{x_i}\right) \begin{bmatrix} a_1(\mathbf{x}, t) \\ \vdots \\ a_d(\mathbf{x}, t) \end{bmatrix} = \begin{bmatrix} a_1 \partial_{x_1} a_1 & \cdots & a_d \partial_{x_d} a_1 \\ \vdots & \ddots & \vdots \\ a_1 \partial_{x_1} a_d & \cdots & a_d \partial_{x_d} a_d \end{bmatrix} \begin{bmatrix} 1 \\ \vdots \\ 1 \end{bmatrix}. \tag{92}$$

In a special case where each dimension is separable, i.e. $\partial_{x_i} a_j = 0 (i \neq j)$, the above $d \times d$ matrix is diagonal, and we have

$$(\mathbf{a} \cdot \nabla) \mathbf{a} = \begin{bmatrix} a_1 \partial_{x_1} a_1 \\ \vdots \\ a_d \partial_{x_1} a_d \end{bmatrix} = \begin{bmatrix} a_1 \\ \vdots \\ a_d \end{bmatrix} \odot \begin{bmatrix} \partial_{x_1} \\ \vdots \\ \partial_{x_d} \end{bmatrix} \odot \begin{bmatrix} a_1 \\ \vdots \\ a_d \end{bmatrix} = \mathbf{a} \odot \nabla \odot \mathbf{a}. \tag{93}$$

where $\odot$ is the element-wise product or operation. In this case, it is sufficient to consider each dimension separately, and it is formally equivalent to the 1-dim case.

### E.1.2   Itô-Taylor Expansion of Stochastic Systems

For the above reasons, it is sufficient to consider the 1-dim case even in the stochastic case. As is well known, Taylor expansion is not valid for stochastic systems $x_t = a(x_t, t)dt + b(x_t, t)dB_t$. This is because of the relation "$dB_t^2 \sim dt$". This effect is taken into account in the celebrated Itô's lemma, i.e., the stochastic version of eq. (80),

$$d\varphi = \underbrace{\left(\frac{\partial}{\partial t} + a(x, t)\frac{\partial}{\partial x} + \frac{b(x, t)^2}{2}\frac{\partial^2}{\partial x^2}\right)}_{=:L_\sharp} \varphi dt + \underbrace{\left(b(x, t)\frac{\partial}{\partial x}\right)}_{=:G_\sharp} \varphi dB_t. \tag{94}$$

By using Itô's formula recursively, we can obtain the following higher-order expansion of a stochastic system, which is called the Itô-Taylor expansion.

$$x_h = x_0 + \int_0^h a(x_t, t)dt + \int_0^h b(x_t, t)dB_t \tag{95}$$

$$
\begin{aligned}
= x_0 &+ \int_0^h \left( a(x_0, 0) + \int_0^t (L_\sharp a)(x_s, s)ds + \int_0^t (G_\sharp a)(x_s, s)dB_s \right) dt \\
&+ \int_0^h \left( b(x_0, 0) + \int_0^t (L_\sharp b)(x_s, s)ds + \int_0^t (G_\sharp b)(x_s, s)dB_s \right) dB_t \\
= x_0 &+ a(x_0, 0) \int_0^h dt + \int_0^h \int_0^t (L_\sharp a)(x_s, s)dsdt + \int_0^h \int_0^t (G_\sharp a)(x_s, s)dB_sdt \\
&+ b(x_0, 0) \int_0^h dB_t + \int_0^h \int_0^t (L_\sharp b)(x_s, s)dsdB_t + \int_0^h \int_0^t (G_\sharp b)(x_s, s)dB_sdB_t \\
= x_0 &+ a(x_0, 0) \int_0^h dt + b(x_0, 0) \int_0^h dB_t \\
&+ \int_0^h \int_0^t \left( (L_\sharp a)(x_0, 0) + \cdots \right) dsdt + \int_0^h \int_0^t \left( (G_\sharp a)(x_0, 0) + \cdots \right) dB_sdt \\
&+ \int_0^h \int_0^t \left( (L_\sharp b)(x_0, 0) + \cdots \right) dsdB_t + \int_0^h \int_0^t \left( (G_\sharp b)(x_0, 0) + \cdots \right) dB_sdB_t \\
= x_0 &+ a(x_0, 0)h + b(x_0, 0)B_h \\
&+ (L_\sharp a)(x_0, 0) \int_0^h \int_0^t dsdt + (G_\sharp a)(x_0, 0) \int_0^h \int_0^t dB_sdt \\
&+ (L_\sharp b)(x_0, 0) \int_0^h \int_0^t dsdB_t + (G_\sharp b)(x_0, 0) \int_0^h \int_0^t dB_sdB_t \\
&+ Remainder,
\end{aligned}
\tag{96}
$$

where the remainder consists of triple integrals as follows,

$$
Remainder = \int_0^h \int_0^t \int_0^s (L_\sharp^2 a)(x_u, u)dudsdt + \cdots .
\tag{97}
$$

We ignore these terms now. If we also ignore the double integrals, we obtain the Euler-Maruyama scheme.

**Evaluation of Each Integral in eq. (96)**  Let us evaluate the double integrals.

$$
\begin{cases}
\int_0^h \int_0^t dsdt & \cdots \text{(deterministic)} \\
\int_0^h \int_0^t dB_sdt & \cdots \text{(stochastic 1)} \\
\int_0^h \int_0^t dsdB_t & \cdots \text{(stochastic 2)} \\
\int_0^h \int_0^t dB_sdB_t & \cdots \text{(stochastic 3)}
\end{cases}
$$

■ *Deterministic*:  The deterministic one, $\int_0^h \int_0^t dsdt$, is easy to evaluate.

$$
\int_0^h \int_0^t dsdt = \int_0^h tdt = \frac{1}{2}h^2.
\tag{98}
$$

■ *Stochastic 1*:  Other integrals contain stochastic integrations. Let us denote the first one by $\tilde{z}$.

$$
\tilde{z} := \int_0^h \int_0^t dB_sdt = \int_0^h B_tdt.
\tag{99}
$$

As $\tilde{z}$ is the limit of a sum of Gaussian variables, i.e.,

$$\tilde{z} = \lim_{n\to\infty} \sum_{i=0}^{n-1} \frac{h}{n} B_{hi/n} = \lim_{n\to\infty} \frac{h}{n} \sum_{i=0}^{n-1} \sum_{j=1}^{i} (B_{hj/n} - B_{h(j-1)/n)})$$

$$= \lim_{n\to\infty} \frac{h}{n} \sum_{i=0}^{n-1} \sum_{j=1}^{i} W_j, \quad W_j \sim \mathcal{N}(0, \frac{h}{n}), \tag{100}$$

so $\tilde{z}$ is also a Gaussian, whose mean is 0. The variance, however, requires some discussions, which shall be seen later. In addition, we shall also see that $\tilde{z}$ is correlated with $B_h$,

$$\mathbb{E}[\tilde{z} \cdot B_h] \neq 0. \tag{101}$$

■ *Stochastic 2*: The second one has the correlation with the first one as follows. Here, we use the integral-by-parts formula. See e.g. (Øksendal, 2013, Theorem 4.1.5).

$$\int_0^h \int_0^t ds dB_t = \int_0^h t dB_t = t B_t \Big|_0^h - \int_0^h B_t dt = h B_h - \tilde{z}. \tag{102}$$

■ *Stochastic 3*: The third one is computed as follows, using a famous formula of Itô integral.

$$\int_0^h \int_0^t dB_s dB_t = \int_0^h B_t dB_t = \frac{1}{2} \left( B_h^2 - h \right). \tag{103}$$

This is derived by substituting $\varphi(x,t) = x^2$ and $x_t = B_t$ (i.e., $dx_t = 0 \cdot dt + 1 \cdot dB_t$) into Itô's formula eq. (94).

■ *Substituting the Integrals to the Itô-Taylor Expansion*: Let us denote $\tilde{w} := B_h \sim \mathcal{N}(0, h)$, and let us shift the integral interval from $(0, h)$ to $(t, t + h)$. Then, we may rewrite the above second order expansion as follows, which we have already seen in the main text,

$$x_{t+h} = x_t + h a(x_t, t) + \tilde{w} b(x_t, t)$$
$$+ (L_\sharp a)(x_t, t) \cdot h^2/2 + (G_\sharp a)(x_t, t) \cdot \tilde{z}$$
$$+ (L_\sharp b)(x_t, t) \cdot (\tilde{w}h - \tilde{z}) + (G_\sharp b)(x_t, t) \cdot (\tilde{w}^2 - h)/2. \tag{104}$$

**Covariance of the Random Variables** $\tilde{w}, \tilde{z}$ Next, let us evaluate the the variance of $\tilde{z}$, and the correlation between the Gaussian variables $\tilde{w}$ and $\tilde{z}$. Let us first calculate the variance of $(\tilde{w}h - \tilde{z})$. By Itô's isometry, we have

$$\mathbb{E}[(\tilde{w}h - \tilde{z})^2] = \mathbb{E}\left[ \left( \int_0^h s dB_s \right)^2 \right] = \mathbb{E}\left[ \int_0^h s^2 ds \right] = \frac{1}{3} h^3. \tag{105}$$

The correlation between $\tilde{w}$ and $\tilde{z}$ is similarly evaluated by Itô's isometry (see (Øksendal, 2013, Proof of Lemma 3.1.5)) as follows,

$$\mathbb{E}[\tilde{w}\tilde{z}] = \mathbb{E}\left[ B_h \int_0^h s dB_s \right] = \mathbb{E}\left[ \left( \int_0^h dB_s \right) \left( \int_0^h s dB_s \right) \right] = \mathbb{E}\left[ \int_0^h s ds \right] = \frac{1}{2} h^2. \tag{106}$$

Using the above variance and covariance, we can calculate the variance of $\tilde{z}$ as follows,

$$\mathbb{E}[\tilde{z}^2] = \mathbb{E}[(\tilde{w}h - \tilde{z})^2 - h^2 \tilde{w}^2 + 2h \tilde{w}\tilde{z}] = \frac{1}{3}h^3 - h^2 \cdot h + 2h \cdot \frac{h^2}{2} = \frac{1}{3}h^3. \tag{107}$$

We need to find random variables $\tilde{w}, \tilde{z}$ that satisfy the requirements for (co)variances that $\mathbb{E}[\tilde{w}^2] = h$, $\mathbb{E}[\tilde{w}\tilde{z}] = h^2/2$ and $\mathbb{E}[\tilde{z}^2] = h^3/3$, and we can easily verify that the following ones do,

$$\begin{bmatrix} \tilde{w} \\ \tilde{z} \end{bmatrix} = \begin{bmatrix} \sqrt{h} & 0 \\ h\sqrt{h}/2 & h\sqrt{h}/2\sqrt{3} \end{bmatrix} \begin{bmatrix} u_1 \\ u_2 \end{bmatrix} \tag{108}$$

where $u_1, u_2 \overset{\text{i.i.d.}}{\sim} \mathcal{N}(0, 1)$. Let us compute the covariance matrix just to be sure.

$$
\begin{aligned}
\mathbb{E}\left[\begin{bmatrix} \tilde{w} \\ \tilde{z} \end{bmatrix} \begin{bmatrix} \tilde{w} & \tilde{z} \end{bmatrix}\right] &= \mathbb{E}\left[\begin{bmatrix} \sqrt{h} & 0 \\ h\sqrt{h}/2 & h\sqrt{h}/2\sqrt{3} \end{bmatrix} \begin{bmatrix} u_1 \\ u_2 \end{bmatrix} \begin{bmatrix} u_1 & u_2 \end{bmatrix} \begin{bmatrix} \sqrt{h} & h\sqrt{h}/2 \\ 0 & h\sqrt{h}/2\sqrt{3} \end{bmatrix}\right] \\
&= \begin{bmatrix} \sqrt{h} & 0 \\ h\sqrt{h}/2 & h\sqrt{h}/2\sqrt{3} \end{bmatrix} \mathbb{E}\left[\begin{bmatrix} u_1^2 & u_1 u_2 \\ u_1 u_2 & u_2^2 \end{bmatrix}\right] \begin{bmatrix} \sqrt{h} & h\sqrt{h}/2 \\ 0 & h\sqrt{h}/2\sqrt{3} \end{bmatrix} \\
&= \begin{bmatrix} \sqrt{h} & 0 \\ h\sqrt{h}/2 & h\sqrt{h}/2\sqrt{3} \end{bmatrix} \begin{bmatrix} \sqrt{h} & h\sqrt{h}/2 \\ 0 & h\sqrt{h}/2\sqrt{3} \end{bmatrix} \\
&= \begin{bmatrix} h & h^2/2 \\ h^2/2 & h^3/3 \end{bmatrix}.
\end{aligned}
\tag{109}
$$

### E.2 DIFFUSION PROCESS AND FOKKER-PLANCK EQUATION

**Derivation of Fokker-Planck Equation**   In this section, let us derive the Fokker-Planck equation

$$\partial_t p(\mathbf{x}_t, t) = -\nabla_{\mathbf{x}_t} (f(\mathbf{x}_t, t)p(\mathbf{x}_t, t)) + \frac{1}{2}g(t)^2 \nabla_{\mathbf{x}_t}^2 p(\mathbf{x}_t, t). \tag{110}$$

from the Itô SDE,

$$d\mathbf{x}_t = f(\mathbf{x}_t, t)dt + g(t)d\mathbf{B}_t. \tag{111}$$

For simplicity, we consider a 1-dim case here. The following approach using the Fourier methods (characteristic function) will be easy and intuitive. See also e.g. (Cox & Miller, 1965, § 5),(Karlin & Taylor, 1981, § 15),(Shreve, 2004, § 6) and (Särkkä & Solin, 2019, § 5).

Let us consider an infinitesimal time step $h$. Then $x_{t+h}$ is written as follows,

$$x_{t+h} = x_t + w_t, \quad w_t \sim \mathcal{N}(f(x_t, t)h, g(t)^2 h). \tag{112}$$

Let us consider the characteristic function $\phi_x(\omega) := \mathbb{E}_{p(x)}[\exp i\omega x]$ (where $i = \sqrt{-1}$), which is the Fourier transform of the density function $p(x)$. Because of the convolution theorem $\phi_{x+y}(\omega) = \phi_x(\omega) \cdot \phi_y(\omega)$, the characteristic functions of $p(x_{t+h}, t+h)$, $p(x_t, t)$, and $p(w_t)$ have the following relation,

$$\phi_{x_{t+h}}(\omega) = \phi_{w_t}(\omega)\phi_{x_t}(\omega). \tag{113}$$

It is easily shown that the characteristic function of the above Gaussian is given by

$$\phi_{w_t}(\omega) = \exp\left(i\omega f(x_t, t)h - \frac{1}{2}g(t)^2 h\omega^2\right). \tag{114}$$

Expanding the r.h.s. up to the first order terms of $h$, we have

$$\phi_{w_t}(\omega) = 1 + i\omega f(x_t, t)h - \frac{1}{2}g(t)^2 h\omega^2 + O(h^2). \tag{115}$$

Thus we obtain

$$\frac{\phi_{x_{t+h}}(\omega) - \phi_{x_t}(\omega)}{h} = \left(-(-i\omega)f(x_t, t) + (-i\omega)^2 \frac{g(t)^2}{2}\right)\phi_t(\omega) + O(h). \tag{116}$$

When $h \to 0$,

$$\partial_t \phi_{x_t}(\omega) = -(-i\omega)f(x_t, t)\phi_t(\omega) + (-i\omega)^2 \frac{g(t)^2}{2}\phi_t(\omega). \tag{117}$$

Since $(-i\omega)$ in the Fourier domain corresponds to the spatial derivative $\partial_{x_t}$ in the real domain [6], it translates as follows,

$$\partial_t p(x_t, t) = -\partial_{x_t}(f(x_t, t)p(x_t, t)) + \partial_{x_t}^2 \frac{g(t)^2}{2} p(x_t, t), \tag{118}$$

and thus we have obtained the Fokker-Planck equation. In particular, if $f = 0$, this equation is called the heat equation, which was also first developed by Fourier.

**Example: Overdamped Langevin Dynamics**   When the drift term is the gradient of a potential function $U(\cdot)$, the SDE is often called the overdampled Langevin equation[7],

$$d\mathbf{x}_t = -\nabla_{\mathbf{x}_t} U(\mathbf{x}_t)dt + \sqrt{2D}d\mathbf{B}_t, \tag{119}$$

---

[6]Integral by parts and $p(-\infty) = p(\infty) = 0$. Formally writing,

$$\mathbb{E}_{\frac{d}{dx}p(x)}[e^{i\omega x}] = \int_{-\infty}^{\infty} e^{i\omega x} \frac{d}{dx} p(x)dx = -\int_{-\infty}^{\infty} i\omega e^{i\omega x} p(x)dx = (-i\omega)\mathbb{E}_{p(x)}[e^{i\omega x}].$$

[7]The Langevin equation should actually be the following equation system, which is called the underdamped Langevin equation,

$$\dot{\mathbf{x}}_t = \mathbf{v}_t, \quad M\dot{\mathbf{v}}_t = -\gamma\mathbf{v}_t - \nabla_{\mathbf{x}_t}U(\mathbf{x}_t) + \sqrt{2D}\frac{d\mathbf{B}_t}{dt},$$

where $\mathbf{v}_t$ is the velocity (momentum) variable, $M$ is the mass of particle, and $\gamma$ is a constant called *friction* or *viscosity* coefficient. In this case, the energy function should also be modified as $E = \frac{1}{2}M\|\mathbf{v}\|^2 + U(\mathbf{x})$. Assuming that the mass is very small compared to the friction, the derivative of momentum $M\dot{\mathbf{v}}_t$ can be ignored (i.e., if the force $F$ is constant, the ODE $M\dot{v} = -\gamma v + F$ has the solution $v = Ce^{-t\gamma/M} + F/\gamma$, and the velocity immediately converges to $F/\gamma$ if $M \ll \gamma$) and the overdamped equation is obtained.

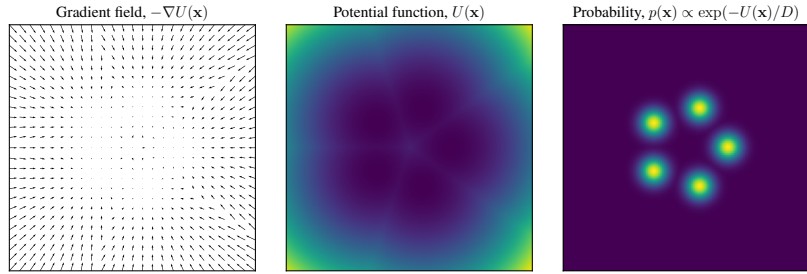

Figure 13: Force field (potential gradient, score function) $-\nabla U(\mathbf{x})$, potential function $U(\mathbf{x})$, and the Boltzmann distribution $p(\mathbf{x}) = \frac{1}{Z}e^{-U(\mathbf{x})/D}$. The scalar potential $U(\mathbf{x})$ is given by eq. (122)

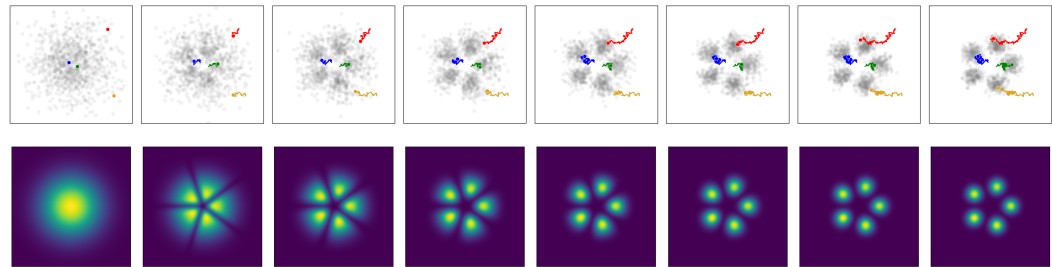

Figure 14: Numerical simulation of Langevin and Fokker-Planck equations. (Top: microscopic particle-wise dynamics) Time evolution of 1000 random samples $\{\mathbf{x}_t^{(i)}\}_{i=1}^{1000}$ obeying the Langevin equation. Paths of four particular samples are shown in color. The time evolves from left to right. (Bottom: macroscopic population dynamics) Time evolution of the density function $p(\mathbf{x}, t)$ obeying the Fokker-Planck equation.

where $D$ is a scalar constant. Its associated Fokker-Planck equation is

$$
\begin{aligned}
\partial_t p(\mathbf{x}_t, t) &= \nabla_{\mathbf{x}_t} \cdot (\nabla_{\mathbf{x}_t} U(\mathbf{x}_t) \cdot p(\mathbf{x}_t, t) + \nabla_{\mathbf{x}_t} D p(\mathbf{x}_t, t)) \\
&= \nabla_{\mathbf{x}_t} \cdot [\nabla_{\mathbf{x}_t} (U(\mathbf{x}_t) + D \log p(\mathbf{x}_t, t)) \cdot p(\mathbf{x}_t, t)] .
\end{aligned}
\tag{120}
$$

If $U(\mathbf{x}_t) + D \log p(\mathbf{x}_t, t)$ is constant, the r.h.s. will be zero, and therefore, $\partial_t p(\mathbf{x}_t, t) = 0$. That is,

$$
p(\mathbf{x}) = \frac{1}{Z}e^{-U(\mathbf{x})/D}, \quad \text{where } Z = \int_{\mathbb{R}^d} e^{-U(\mathbf{x})/D} d\mathbf{x}
\tag{121}
$$

is the stationary solution of the FPE, and it does no longer evolve over time. Therefore, it is expected that the particles obeying the Langevin equation will eventually follow this Boltzmann distribution.

Let us compare the microscopic dynamics of each particle obeying the Langevin equation with the macroscopic dynamics of the population obeying the FPE using a 2-dim toy model. The energy function we use here is the following Gaussian mixture model,

$$
U(x, y) = -\log \sum_{k=1}^{5} \exp\left[ -\frac{(x - \cos\frac{2k\pi}{5})^2 + (y - \sin\frac{2k\pi}{5})^2}{2\sigma^2} \right],
\tag{122}
$$

where $\sigma = 0.1$. The diffusion parameter was $D = 5$. Figure 13 shows the force field $-\nabla U(\mathbf{x})$ where $\mathbf{x} = (x, y)$, potential function $U(\mathbf{x})$, and the Boltzmann distribution $p(\mathbf{x}) = \frac{1}{Z}e^{-U(\mathbf{x})/D}$.

Figure 14 shows the time evolution of Langevin and Fokker-Planck equations, where the initial density was $p(\mathbf{x}_0, 0) = \mathcal{N}(\mathbf{0}, \mathbf{I})$. The time step size was $h = 5 \times 10^{-5}$, and the figures are plotted every 50 steps. These figures will be helpful to intuitively understand that the SDE (Langevin equation) and the FPE are describing the same phenomenon from different perspectives: individual description or population description.

### E.3 DERIVATION OF THE PROBABILITY FLOW ODE AND REVERSE-SDE

#### E.3.1 DERIVATION OF THE PROBABILITY FLOW ODE

For convenience, the derivation of the PF-ODE is briefly described here. For more general case details where $g(t)$ is a matrix-valued coefficient which is dependent on $\mathbf{x}_t$, i.e. $\mathbf{G}(\mathbf{x}_t, t)$, please refer to the original paper (Song et al., 2020b).

Firstly, let us consider an SDE,

$$d\mathbf{x}_t = \mathbf{f}(\mathbf{x}_t, t)dt + g(t)d\mathbf{B}_t. \tag{123}$$

The associated FPE is,

$$\partial_t p(\mathbf{x}_t, t) = -\nabla \cdot (\mathbf{f}(\mathbf{x}_t, t)p(\mathbf{x}_t, t)) + \Delta \frac{1}{2}g(t)^2 p(\mathbf{x}_t, t). \tag{124}$$

Here, let us forcibly incorporate the diffusion term (i.e., the second-order derivative) into the drift term (first-order derivative). Noting that the Laplacian is the inner product of two $\nabla$-s, i.e. $\Delta = \nabla \cdot \nabla$,

$$
\begin{aligned}
\text{r.h.s.} &= -\nabla \cdot \left[ \mathbf{f}(\mathbf{x}_t, t)p(\mathbf{x}_t, t) - \nabla \frac{1}{2}g(t)^2 p(\mathbf{x}_t, t) \right] \\
&= -\nabla \cdot \left[ \left( \mathbf{f}(\mathbf{x}_t, t) - \frac{1}{p(\mathbf{x}_t, t)}\nabla \frac{1}{2}g(t)^2 p(\mathbf{x}_t, t) \right) p(x_t, t) \right] \\
&= -\nabla \cdot \left[ \Big( \underbrace{f(\mathbf{x}_t, t) - \frac{1}{2}g(t)^2 \nabla \log p(\mathbf{x}_t, t)}_{=: \bar{\mathbf{f}}_\flat(\mathbf{x}_t, t)} \Big) p(\mathbf{x}_t, t) \right].
\end{aligned}
\tag{125}
$$

Thus we have obtained a diffusion-free version of the FPE of the form,

$$\partial_t p(\mathbf{x}_t, t) = -\nabla \cdot \left( \bar{\mathbf{f}}_\flat(\mathbf{x}_t, t)p(\mathbf{x}_t, t) \right) - \cancel{\Delta (0 \cdot p(\mathbf{x}_t, t))}. \tag{126}$$

Since it is an 'FPE', it has an associated 'SDE' as follows,

$$d\mathbf{x}_t = \bar{\mathbf{f}}_\flat(\mathbf{x}_t, t)dt + \cancel{0 \cdot d\mathbf{B}_t}. \tag{127}$$

Although this 'SDE' does not give the same particle-wise dynamics as the original SDE, its macroscopic dynamics of population is exactly the same as the original one, since the associated FPE has not changed from the original one at all. That is, the density evolves from $p(\mathbf{x}_0, 0)$ to $p(\mathbf{x}_T, T)$ exactly the same way.

Since the obtained 'SDE' is actually a deterministic ODE, its time-reversal is simply obtained by flipping the sign, i.e.,

$$d\mathbf{x}_t = (-\bar{\mathbf{f}}_\flat(\mathbf{x}_t, t))(-dt). \tag{128}$$

Thus the Probability Flow ODE is obtained. If the terminal random variables are drawn from $p(\mathbf{x}_T, T)$, then the particles obeying this ODE will reconstruct the initial density $p(\mathbf{x}_0, 0)$ as a population.

Note that the vector field of the form $\mathbf{j} := \rho\mathbf{v}$ is often referred to as the *flux*, particularly in physical contexts e.g. fluid dynamics, where $\rho$ is density and $\mathbf{v}$ is a vector field (such as the velocity field), and the PDE of the form

$$\partial_t \rho + \nabla \cdot \mathbf{j} = 0 \tag{129}$$

is often called the *continuity equation*, which is closely related to the conservation laws; in the fluid case, the mass conservation law. Another well known example is the charge conservation law in electromagnetism, where $\rho$ is the charge density and $\mathbf{j}$ is the current density. In the present case eq. (126), $\mathbf{j} := p\bar{\mathbf{f}}_\flat$ is understood as the flux, and therefore, $\bar{\mathbf{f}}_\flat$ could be understood as a sort of velocity field, and the conservation law corresponds to the fact that the sum of probability mass is constant. This will give an intuitive understanding why the method is called the Probability "*Flow*" ODE.

#### E.3.2 DERIVATION OF R-FPE FROM KBE

In this section, let us show that R-FPE eq. (4) is the same as the KBE eq. (3). Let us consider the case where the diffusion coefficient is independent of the spatial variable, i.e., $g(\mathbf{x}_t, t) = g(t)$. The overall discussion is based on (Anderson, 1982, § 5), though the following calculation is not explicitly written in the paper.

Firstly, the KBE is

$$-\partial_s p(\mathbf{x}_t, t \mid \mathbf{x}_s, s) = \mathbf{f}(\mathbf{x}_s, s) \cdot \nabla_{\mathbf{x}_s} p(\mathbf{x}_t, t \mid \mathbf{x}_s, s) + \frac{g(s)^2}{2} \Delta_{\mathbf{x}_s} p(\mathbf{x}_t, t \mid \mathbf{x}_s, s),$$

and what we need to show is that this is equivalent as the R-FPE,

$$-\partial_s p(\mathbf{x}_s, s \mid \mathbf{x}_t, t) = \nabla_{\mathbf{x}_s} \cdot (\mathbf{f}(\mathbf{x}_s, t) - g(s)^2 \nabla_{\mathbf{x}_s} \log p(\mathbf{x}_s, s)) p(\mathbf{x}_s, s \mid \mathbf{x}_t, t) + \Delta_{\mathbf{x}_s} \frac{g(s)^2}{2} p(\mathbf{x}_s, s \mid \mathbf{x}_t, t)$$

For simplicity, we derive the former from the latter. By the Bayes theorem, we may substitute each $p(\mathbf{x}_s, s \mid \mathbf{x}_t, t)$ with $p(\mathbf{x}_t, t \mid \mathbf{x}_s, s) p(\mathbf{x}_s, s) / p(\mathbf{x}_t, t)$, and obtain

$$- \partial_s \frac{p(\mathbf{x}_t, t \mid \mathbf{x}_s, s) p(\mathbf{x}_s, s)}{p(\mathbf{x}_t, t)} =$$
$$\nabla_{\mathbf{x}_s} \cdot (\mathbf{f}(\mathbf{x}_s, t) - g(s)^2 \nabla_{\mathbf{x}_s} \log p(\mathbf{x}_s, s)) \frac{p(\mathbf{x}_t, t \mid \mathbf{x}_s, s) p(\mathbf{x}_s, s)}{p(\mathbf{x}_t, t)}$$
$$+ \Delta_{\mathbf{x}_s} \frac{g(s)^2}{2} \frac{p(\mathbf{x}_t, t \mid \mathbf{x}_s, s) p(\mathbf{x}_s, s)}{p(\mathbf{x}_t, t)}. \quad (130)$$

To simplify the notation, let us denote $p_s := p(\mathbf{x}_s, s)$, $p_{t|s} := p(\mathbf{x}_t, t \mid \mathbf{x}_s, s)$, and drop the variables $\mathbf{x}_s$ and $s$ (that is, $\partial$ always means $\partial_s$, and $\nabla$ always means $\nabla_{\mathbf{x}_s}$). Let us also recall the Leibniz rule of the divergence of vector field. Let $\mathbf{a}$ be a vector-valued function and let $u$ be a scalar function, then the following relation holds[8],

$$\text{Leibniz rule:} \qquad \nabla \cdot (\mathbf{a}u) = \mathbf{a} \cdot \nabla u + (\nabla \cdot \mathbf{a})u \qquad (131)$$

Now let us evaluate each term. The l.h.s. is

$$- \partial_s (p(\mathbf{x}_t, t \mid \mathbf{x}_s, s) p(\mathbf{x}_s, s)) = -\partial(p_{t|s} p_s) = -(\partial p_{t|s})p_s - p_{t|s}\partial p_s$$
$$= -(\partial p_{t|s})p_s - p_{t|s}\left(-\nabla \cdot (\mathbf{f}p_s) + \Delta \frac{g^2}{2} p_s\right) \quad (\because \text{FPE})$$
$$= -(\partial p_{t|s})p_s - p_{t|s}\left(-\mathbf{f} \cdot (\nabla p_s) - (\nabla \cdot \mathbf{f})p_s + \Delta \frac{g^2}{2} p_s\right) \quad (\because \text{Leibniz rule}). \quad (132)$$

The drift term is

$$\nabla \cdot \left[(\mathbf{f} - g^2 \nabla \log p_s) p_{t|s} p_s\right]$$
$$= \nabla \cdot \left[\left(\mathbf{f} - g^2 \frac{\nabla p_s}{p_s}\right) p_{t|s} p_s\right]$$
$$= \nabla \cdot \left[\mathbf{f}p_{t|s} p_s - (\nabla p_s) g^2 p_{t|s}\right]$$
$$= \left[\nabla \cdot \mathbf{f}p_{t|s} p_s\right] - \left[\nabla \cdot (\nabla p_s) g^2 p_{t|s}\right]$$
$$= \left[\mathbf{f} \cdot \nabla(p_{t|s} p_s) + (\nabla \cdot \mathbf{f})p_{t|s} p_s\right] - \left[(\nabla p_s) \cdot (\nabla g^2 p_{t|s}) + (\Delta p_s) g^2 p_{t|s}\right]$$
$$= \left[\mathbf{f} \cdot (p_s \nabla p_{t|s} + p_{t|s} \nabla p_s) + (\nabla \cdot \mathbf{f})p_{t|s} p_s\right] - \left[g^2 (\nabla p_s) \cdot (\nabla p_{t|s}) + g^2 (\Delta p_s) p_{t|s}\right]. \quad (133)$$

Finally, the diffusion term is

$$\Delta(\frac{g^2}{2} p_{t|s} p_s) = \nabla \cdot \nabla(\frac{g^2}{2} p_{t|s} p_s)$$
$$= \frac{g^2}{2} \nabla \cdot \left((\nabla p_{t|s})p_s + (\nabla p_s)p_{t|s}\right)$$
$$= \frac{g^2}{2}(p_s \Delta p_{t|s} + p_{t|s} \Delta p_s + 2\nabla p_s \cdot \nabla p_{t|s}). \quad (134)$$

---

[8]This is also written as $\text{div}(\mathbf{a}u) = \mathbf{a} \cdot \text{grad}u + u(\text{div}\mathbf{a})$. This is just expressing the relation, $\sum_i \partial_i(a_i u) = \sum_i (a_i(\partial_i u) + (\partial_i a_i)u)$.

Now let us simplify the R-FPE using the above relations.

$$\partial p_{s|t} + \nabla \cdot \left[ (\mathbf{f} - g^2 \nabla \log p_s) p_{s|t} \right] + \Delta \frac{g^2}{2} p_{s|t}$$

$$\propto \partial(p_{t|s} p_s) + \nabla \cdot \left[ (\mathbf{f} - g^2 \nabla \log p_s) p_{t|s} p_s \right] + \Delta(\frac{g^2}{2} p_{t|s} p_s)$$

$$= (\partial p_{t|s}) p_s + p_{t|s} \left( -\cancel{\mathbf{f} \cdot (\nabla p_s)} - \cancel{(\nabla \cdot \mathbf{f}) p_s} + \cancel{\Delta \frac{g^2}{2} p_s} \right)$$

$$+ \left[ \mathbf{f} \cdot (p_s \nabla p_{t|s} + \cancel{p_{t|s} \nabla p_s}) + \cancel{(\nabla \cdot \mathbf{f}) p_{t|s} p_s} \right] - \left[ g^2 (\nabla p_s) \cdot (\nabla p_{t|s}) + \cancel{g^2 (\Delta p_s) p_{t|s}} \right]$$

$$+ \frac{g^2}{2} (p_s \Delta p_{t|s} + \cancel{p_{t|s} \Delta p_s} + 2 \nabla p_s \cdot \nabla p_{t|s})$$

$$= (\partial p_{t|s}) p_s + \mathbf{f} \cdot (p_s \nabla p_{t|s}) + \frac{g^2}{2} (\Delta p_{t|s}) p_s$$

$$= p_s \left( \partial p_{t|s} + \mathbf{f} \cdot \nabla p_{t|s} + \frac{g^2}{2} (\Delta p_{t|s}) \right). \tag{135}$$

It implies that if the following relation holds, then the R-FPE also holds,

$$\partial_s p_{t|s} + \mathbf{f} \cdot \nabla p_{t|s} + \frac{g^2}{2} (\Delta p_{t|s}) = 0.$$

This equation is nothing other than the KBE. Hence, KBE implies R-FPE.

We may also do the same thing for the case where the diffusion coefficient $g(\mathbf{x}_t, t)$ is dependent on the spatial variable $\mathbf{x}_t$. In this case, the target R-FPE is modified as

$$-\partial p_{s|t} = \nabla \cdot \left[ \left( \mathbf{f} - \frac{1}{p_s} \nabla(g^2 p_s) \right) p_{s|t} \right] + \Delta \left( \frac{g^2}{2} p_{s|t} \right)$$

$$= \nabla \cdot \left[ \left( \mathbf{f} - \nabla g^2 - g^2 \nabla \log p_s \right) p_{s|t} \right] + \Delta \left( \frac{g^2}{2} p_{s|t} \right), \tag{136}$$

but we can also show that the KBE implies this relation. The use of computer algebra systems such as SymPy can reduce the calculation effort for checking this fact.

### E.4 On the Convergence of Numerical SDE Schemes.

Let us introduce two convergence concepts which are commonly used in numerical SDE studies. See also (Kloeden et al., 1994, § 3.3, § 3.4)

**Definition 1** (Strong Convergence). *Let $\tilde{\mathbf{x}}_t$ be the path at the continuous limit $h \to 0$, and $\mathbf{x}_t$ be the discretized numerical path, computed by a numerical scheme with the step size $h > 0$. Then, it is said the numerical scheme has the strong order of convergence $\gamma$ if the following inequality holds for a certain constant $K_s > 0$,*

$$\mathbb{E}[|\mathbf{x}_t - \tilde{\mathbf{x}}_t|] \le K_s h^\gamma. \tag{137}$$

**Definition 2** (Weak Convergence). *Similarly, it is said that the scheme has the weak order of convergence $\beta$, if the following inequality holds for any test functions $\phi(\cdot)$ in a certain class of functions, and a certain constant $K_w > 0$,*

$$|\mathbb{E}[\phi(\mathbf{x}_t)] - \mathbb{E}[\phi(\tilde{\mathbf{x}}_t)]| \le K_w h^\beta. \tag{138}$$

It is known that the Euler-Maruyama scheme has the strong convergence of order $\gamma = 0.5$, and weak order of $\beta = 1$, in general. However, for more specific cases including the diffusion generative models that the diffusion coefficient $g(t)$ is not dependent on $\mathbf{x}_t$, the Euler-Maruyama scheme has a little better strong convergence of order $\gamma = 1$.

The strong convergence is concerned with the precision of the path, while the weak convergence is with the precision of the moments. In our case, we are not much interested in whether a data $\hat{\mathbf{x}}_0$ generated using a finite $h > 0$ approximates the continuous limit $\tilde{\mathbf{x}}_0, (h \to 0)$ driven by the same Brownian motion. Instead, we are more interested in whether the density $p(\hat{\mathbf{x}}_0)$ of the samples generated with a finite step size $h > 0$ approximates the ideal density $p(\tilde{\mathbf{x}}_0), (h \to 0)$ which is supposed to approximate the true density $p(\mathbf{x}_0)$. In this sense, the concept of strong convergence is not much important for us, but the weak convergence would be sufficient.

### E.5 RUNGE-KUTTA METHODS

Let us briefly introduce the derivation of Runge-Kutta methods for a 1-dim ODE $\dot{x} = f(x,t)$. We are interested in deriving the following derivative-free formula

$$x_{t+h} = x_t + \sum_{i=1}^{n} hb_i k_i + o(h^p), \quad \text{where } k_i = f(x_t + \sum_{j<i} ha_{ij} k_j, t + hc_i), \quad n \geq p. \quad (139)$$

The array of coefficients are often shown in the following form, which is called the Butcher tableau.

$$\begin{array}{c|ccccc}
0 & & & & & \\
c_2 & a_{21} & & & & \\
c_3 & a_{31} & a_{32} & & & \\
\vdots & \vdots & \vdots & \ddots & & \\
c_n & a_{n1} & a_{n2} & \cdots & a_{n,n-1} & \\
\hline
& b_1 & b_2 & \cdots & b_{n-1} & b_n
\end{array} \quad (140)$$

Now the problem is how to design each value in this tableau. First, for simplicity, let us consider the following 2nd order case,

$$\begin{aligned}
x_{t+h} &= x_t + hb_1 k_1 + hb_2 k_2 + o(h^2) \\
k_1 &= f(x_t, t) \\
k_2 &= f(x_t + ha_{21} k_1, t + hc_2),
\end{aligned}$$

or,

$$x_{t+h} = x_t + hb_1 f(x_t, t) + hb_2 f(x_t + ha_{21} f(x_t, t), t + hc_2) + o(h^2). \quad (141)$$

Noting that the Taylor expansions of the l.h.s. and the third term in r.h.s. are written as follows,

$$(\text{l.h.s.}) = x_t + hf(x_t, t) + \frac{h^2}{2}(\dot{f} + ff')(x_t, t) + o(h^2), \quad (142)$$

$$(\text{third term of r.h.s.}) = hb_2 \left( f(x_t, t) + ha_{21} f(x_t, t) f'(x_t, t) + hc_2 \dot{f}(x_t, t) + o(h) \right), \quad (143)$$

we obtain the following relation.

$$x_t + hf(x_t, t) + \frac{h^2}{2}(\dot{f} + ff')(x_t, t) =$$
$$x_t + hb_1 f(x_t, t) + hb_2 \left( f(x_t, t) + ha_{21} f(x_t, t) f'(x_t, t) + hc_2 \dot{f}(x_t, t) \right) + o(h^2). \quad (144)$$

Comparing both sides, we can find that the derivatives in both sides can be eliminated if the following equation are satisfied,

$$hf(x_t, t) = hb_1 f(x_t, t) + hb_2 f(x_t, t), \quad (145)$$

$$\frac{h^2}{2}(\dot{f} + ff')(x_t, t) = hb_2 \left( ha_{21} f(x_t, t) f'(x_t, t) + hc_2 \dot{f}(x_t, t) \right). \quad (146)$$

By simplifying the above equations, we obtain the following relations of coefficients.

$$b_1 + b_2 = 1, \quad b_2 a_{21} = \frac{1}{2}, \quad b_2 c_2 = \frac{1}{2}, \quad (147)$$

and thus we obtain the following Butcher tableau.

$$\begin{array}{c|cc}
0 & & \\
c_2 & a_{21} & \\
\hline
& b_1 & b_2
\end{array}
=
\begin{array}{c|cc}
0 & & \\
c_2 & c_2 & \\
\hline
& 1 - 1/(2c_2) & 1/(2c_2)
\end{array} \quad (148)$$

(When $c_2 = 1$, the method is particularly called Heun's method.) Thus we have confirmed that the second order Taylor expansion of $x_{t+h}$ is expressed as eq. (141), which is independent of any derivatives of $f$.

We can similarly consider higher-order methods. In the 3rd-order case, following relations are automatically obtained after a little effort of symbolic math programming using e.g. SymPy,

$$b_1 + b_2 + b_3 = 1, \quad b_2 c_2 + b_3 c_3 = \frac{1}{2}, \quad b_2 c_2^2 + b_3 c_3^2 = \frac{1}{3}, \quad a_{32} a_{21} b_3 = \frac{1}{6}, \quad a_{32} b_3 c_2 = \frac{1}{6},$$

$$a_{21} b_2 + (a_{31} + a_{32}) b_3 = \frac{1}{2}, \quad a_{21}^2 b_2 + (a_{32} + a_{31})^2 b_3 = \frac{1}{3}, \quad a_{21} b_2 c_2 + (a_{31} + a_{32}) b_3 c_3 = \frac{1}{3}.$$

By solving this equation system using SymPy, the following Butcher tableau is obtained.

$$
\begin{array}{c|ccc}
0 \\
c_2 & a_{21} \\
c_3 & a_{31} & a_{32} \\
\hline
& b_1 & b_2 & b_3
\end{array}
=
\begin{array}{c|ccc}
0 \\
c_2(\neq 2/3) & c_2 \\
c_3(\neq c_2) & \frac{c_3(3c_2^2 - 3c_2 + c_3)}{c_2(3c_2 - 2)} & \frac{c_3(c_2 - c_3)}{c_2(3c_2 - 2)} \\
\hline
& 1 - \frac{1}{2c_2} - \frac{1}{2c_3} + \frac{1}{3c_2 c_3} & \frac{2 - 3c_3}{6c_2(c_2 - c_3)} & \frac{3c_2 - 2}{6c_3(c_2 - c_3)}
\end{array}
\tag{149}
$$

It will naturally become much more complicated when we consider higher-order methods, though it can be simpler if substituting specific values. In the 4th-order case, when the Butcher tableau is written as follows, the method is particularly called the Classical 4th-order Runge-Kutta (RK4) method.

$$
\begin{array}{c|cccc}
0 \\
c_2 & a_{21} \\
c_3 & a_{31} & a_{32} \\
c_4 & a_{41} & a_{42} & a_{43} \\
\hline
& b_1 & b_2 & b_3 & b_4
\end{array}
=
\begin{array}{c|cccc}
0 \\
1/2 & 1/2 \\
1/2 & 0 & 1/2 \\
1 & 0 & 0 & 1 \\
\hline
& 1/6 & 1/3 & 1/3 & 1/6
\end{array}
\tag{150}
$$

**Numerical Example** Let us consider a toy example $\dot{x} = x \sin t, x_0 = 1$. The exact solution is $x_t = e^{1 - \cos t}$. Figure 15 compares the numerical solutions of the Euler, Heun, Classical RK4 and Taylor 2nd methods, where the step size is $h = 0.5$. It is clearly observed that the Euler method immediately deviates significantly from the exact solution but higher-order methods follow it for longer periods; while 2nd-order methods (Heun and Taylor 2nd) gradually deviate from the exact solution, the Classical 4th-order Runge-Kutta method more finely approximates the exact solution.

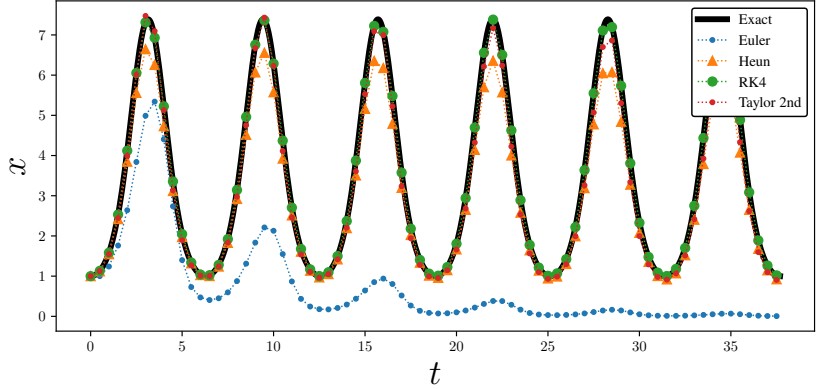

Figure 15: Comparison of numerical solutions of the ODE $\dot{x} = x \sin t, x_0 = 1$.

Note that the Taylor 2nd method is given as follows,

$$x_{t+h} = x_t + h(x_t \sin t) + \frac{h^2}{2} \underbrace{(x_t \cos t + x_t \sin^2 t)}_{\text{derivative of } x \sin t}. \tag{151}$$

In this case the computation of the derivative $(\partial_t + x \sin t \partial_x)(x \sin t)$ is tractable. However, it will be infeasible if this term is a little more complicated. The Runge-Kutta method is advantageous if the derivatives are not easy to compute.

# F  PSEUDOCODE

## F.1  QUASI-TAYLOR SAMPLER

The following pseudocode shows the proposed Quasi-Taylor sampler.

---

**Algorithm 1** Quasi-Taylor Sampling Scheme with Ideal Derivatives

---

**Require:**
  Trained neural network model $\mathbf{S}_\theta(\mathbf{x}_t, t, \mathbf{c})$             ♯ The conditioning information $\mathbf{c}$ is optional.
  Data size $d > 0$    $\cdots$ *Int*
  Total time $T > 0$   $\cdots$ *Float*
  Number of steps $N > 0$   $\cdots$ *Int*
  Variance schedule function $\nu_t$    $\cdots$ *Float* $\rightarrow$ *Float*
  Diffusion schedule function $\beta_t = -\frac{d}{dt}\log(1 - \nu_t)$    $\cdots$ *Float* $\rightarrow$ *Float*
  (Optional) Conditioning information $\mathbf{c}$
  Step size schedule $h_1, h_2, \cdots, h_N$    $\cdots$ *List of Float*
**Begin**

  $\mathbf{x} \sim \mathcal{N}(\mathbf{0}_d, \mathbf{I}_d)$                ♯ Draw a $d$-dimensional Gaussian noise with unit variance

  **for** $n = 1$ **to** $N$ **do**

    **[Step size and time parameter]**

    $h \leftarrow h_n$

    $t \leftarrow T - \sum\limits_{i=1}^{n-1} h_i$

    **[Compute the Noise Level]**

    Compute, or get from lookup table, the values of $\nu_t, \beta_t, \dot{\beta}_t, \ddot{\beta}_t$

    **[Compute Coefficients]**

    Compute $\rho^\flat$ using eq. (23)

    Compute $\mu^\flat$ using eq. (24)

    **[Update Data]**

    $\mathbf{x} \leftarrow \rho^\flat \mathbf{x} + \mu^\flat \mathbf{S}_\theta(\mathbf{x}, t, \mathbf{c})/\sqrt{\nu}$                ♯ eq. (22)

  **end for**
  Clip outliers of $\mathbf{x}$ so that e.g. $-1 \leq \mathbf{x} \leq 1$.
**End**
**Output:** $\mathbf{x}$

---

Note that, $\rho_\flat$ and $\mu_\flat$, as well as $\nu_t, \lambda_t, \beta_t$ and their derivatives, are only dependent on $T$ and $h$. Therefore, they can be pre-computed before the actual synthesis.

## F.2 QUASI-ITÔ-TAYLOR SAMPLER

The following pseudocode shows the proposed Quasi-Itô-Taylor sampler.

---

**Algorithm 2** Quasi-Itô-Taylor Sampling Scheme with Ideal Derivatives

---

**Require:**
  Trained neural network model $\mathbf{S}_\theta(\mathbf{x}_t, t, \mathbf{c})$          $\sharp$ The conditioning information $\mathbf{c}$ is optional.
  Data size $d > 0$    $\cdots Int$
  Total time $T > 0$    $\cdots Float$
  Number of steps $N > 0$    $\cdots Int$
  Variance schedule function $\nu_t$    $\cdots Float \to Float$
  Diffusion schedule function $\beta_t = -\frac{d}{dt} \log(1 - \nu_t)$    $\cdots Float \to Float$
  (Optional) Conditioning information $\mathbf{c}$
  Step size schedule $h_1, h_2, \cdots, h_N$    $\cdots List\ of\ Float$
**Begin**

  $\mathbf{x} \sim \mathcal{N}(\mathbf{0}_d, \mathbf{I}_d)$          $\sharp$ Draw a $d$-dimensional Gaussian noise with unit variance

  **for** $n = 1$ **to** $N$ **do**

    **[Step size and time parameter]**

    $h \leftarrow h_n$

    $t \leftarrow T - \sum_{i=1}^{n-1} h_i$

    **[Compute the Noise Level]**

    Compute, or get from lookup table, the values of $\nu_t, \beta_t, \dot{\beta}_t$

    **[Compute Coefficients]**

    Compute $\rho^\sharp$ using eq. (26)

    Compute $\mu^\sharp$ using eq. (26)

    **[Draw a Correlated Driving Noise]**
    **if** $n = N$ **then**
      $\mathbf{n}^\sharp = \mathbf{0}_d$          $\sharp$ No noise is injected at the final step

    **else**
      $\mathbf{w} \sim \mathcal{N}(\mathbf{0}_d, \mathbf{I}_d), \quad \mathbf{u} \sim \mathcal{N}(\mathbf{0}_d, \mathbf{I}_d), \quad \mathbf{z} = \frac{1}{2}\mathbf{w} + \frac{1}{2\sqrt{3}}\mathbf{u}$    $\sharp$ See eq. (108) and Theorem 1

      Compute $\mathbf{n}^\sharp$ using eq. (27) and $\mathbf{w}, \mathbf{z}$ above.

    **end if**

    **[Update Data]**
    $\mathbf{x} \leftarrow \rho^\sharp \mathbf{x} + \mu^\sharp \mathbf{S}_\theta(\mathbf{x}, t, \mathbf{c})/\sqrt{\nu} + \mathbf{n}^\sharp$          $\sharp$ eq. (25)

  **end for**
  Clip outliers of $\mathbf{x}$ so that e.g. $-1 \leq \mathbf{x} \leq 1$.
**End**
**Output:** $\mathbf{x}$

---

## F.3 TRAINING

The following pseudocode shows an example for the training of diffusion-based generative models.

---

**Algorithm 3** Training of Diffusion models

---

**Require:**
  Data size $d > 0$ $\quad \cdots Int$
  Training Data $\mathcal{D} = \{\mathbf{x}_0^{(i)}, \mathbf{c}^{(i)}\}$, where $\mathbf{x}_0^{(i)} \in \mathbb{R}^d$ $\qquad \sharp$ The conditioning information $\mathbf{c}$ is optional.
  Neural network model $\mathbf{S}_\theta(\mathbf{x}_t, t, \mathbf{c})$, parameterized by $\theta$.
  Neural network optimizer (*Adam*) and its parameters (e.g. learning rate)
  Total time $T > 0$ $\quad \cdots Float$ $\qquad\qquad\qquad\qquad \sharp$ Not necessarily the same as the one for the synthesis.
  Variance schedule function $\nu_t$ $\quad \cdots Float \to Float$ $\quad \sharp$ Not necessarily the same as the one for the synthesis.
  Batch size $b > 0$ $\quad \cdots Int$
**Begin**
  **for** sufficiently many times until convergence **do**
    **[Draw a Batch]**
    $batch \leftarrow [\,]$ $\qquad\qquad\qquad\qquad\qquad\qquad\qquad\qquad\qquad\qquad \sharp$ Empty list
    **for** $b$ times (batch size) **do**
      $(\mathbf{x}_0, \mathbf{c}) \sim \mathcal{D}$ $\qquad\qquad\qquad \sharp$ Draw a data from the set of training data; the conditioning $\mathbf{c}$ is optional.

      $t \sim \text{Uniform}(0, T)$ $\qquad\qquad\qquad \sharp$ Draw a time parameter $t$ from the uniform distribution.

      Compute $\nu_t$

      $\mathbf{w} \sim \mathcal{N}(\mathbf{0}_d, \mathbf{I}_d)$ $\qquad\qquad\qquad \sharp$ Draw a $d$-dimensional Gaussian noise with unit variance.

      Append the tuple $(\mathbf{x}_0, \mathbf{c}, t, \nu, \mathbf{w})$ to $batch$
    **end for**

    **[Forward Computation]**
    $([\mathbf{x}_0], [\mathbf{c}], [t], [\nu], [\mathbf{w}] \leftarrow batch$
    $[\mathbf{s}] \leftarrow \mathbf{S}_\theta([\sqrt{1-\nu}] \odot [\mathbf{x}_0] + [\sqrt{\nu}] \odot [\mathbf{w}], [t], [\mathbf{c}])$ $\qquad\qquad\qquad \sharp$ See also eq. (10).

    **[Back-propagation]**
    $\mathcal{L} \leftarrow \mathbb{E}_{\text{batch}}\left[\|[\mathbf{w}] - [\mathbf{s}]\|_2^2\right]$ $\qquad\qquad\qquad\qquad \sharp$ Loss. See also eq. (10).
    Compute $\nabla_\theta \mathcal{L}$ by the back-propagation.
    $\theta \leftarrow Adam(\boldsymbol{\theta}, \nabla_\theta \mathcal{L})$ $\qquad\qquad\qquad\qquad \sharp$ Update the parameters of $\mathbf{S}_\theta(\cdot, \cdot, \cdot)$.
  **end for**
**End**
**Output:** The network parameter $\theta$.

---

# G  ADDITIONAL RESULTS OF IMAGE SYNTHESIS

## G.1  SAMPLING EXAMPLES

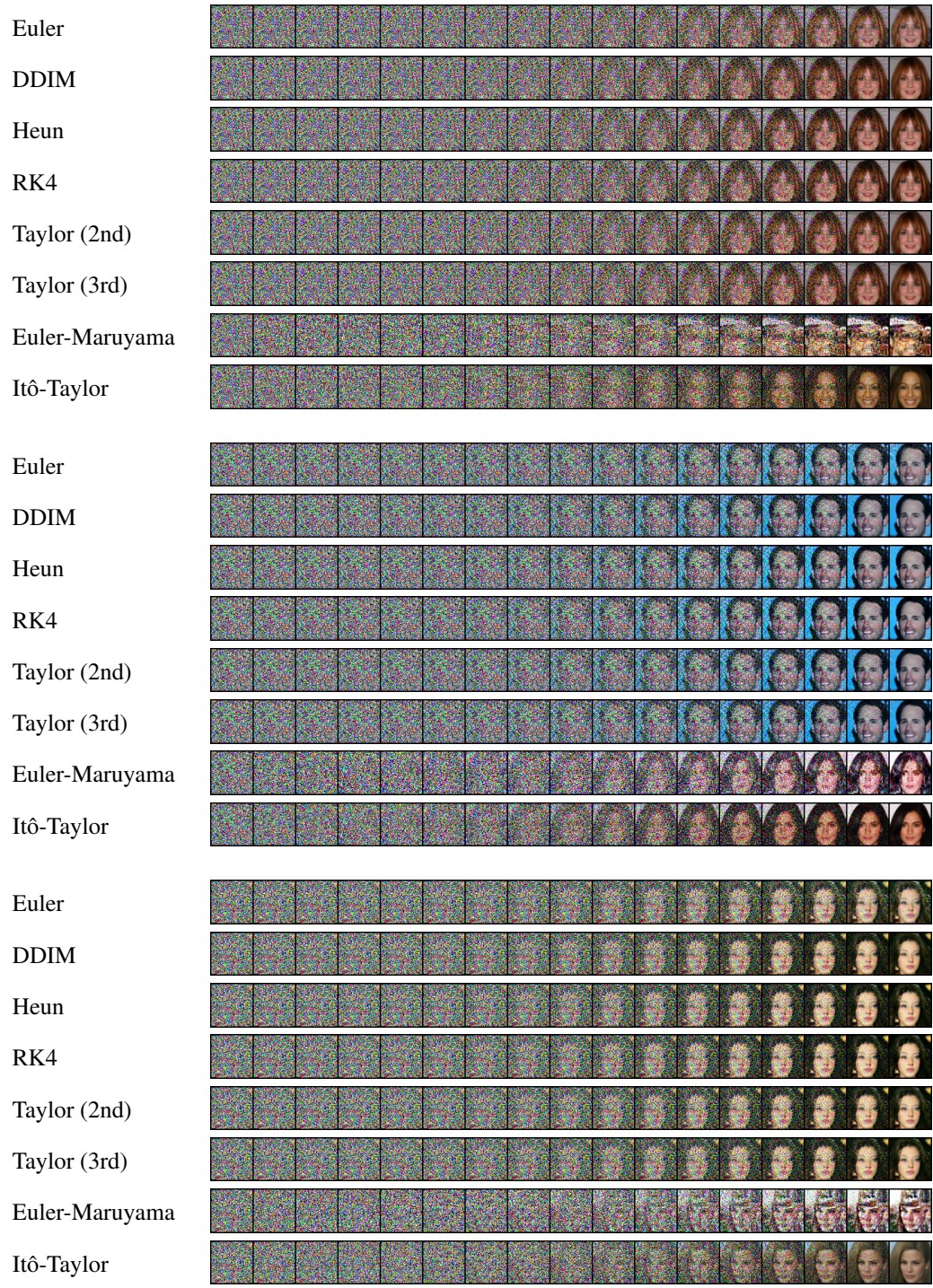

Figure 16: Comparison of the image synthesis process. The dataset is CelebA ($64 \times 64$). All the conditions but the sampling algorithm are the same including the random seed. The number of refinement steps is $N = 16$.

## G.2 RANDOM SAMPLES

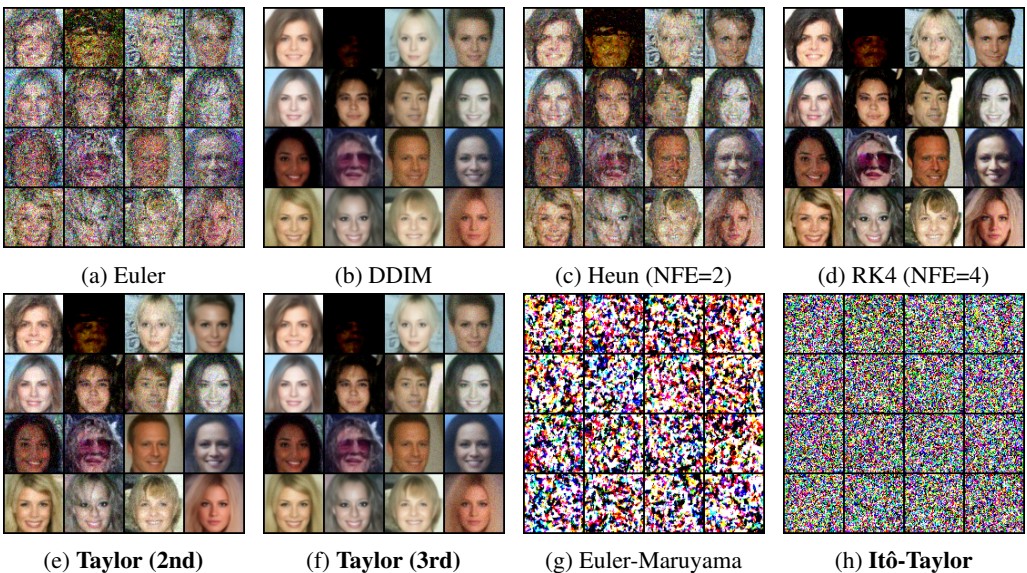

(a) Euler        (b) DDIM        (c) Heun (NFE=2)        (d) RK4 (NFE=4)

(e) **Taylor (2nd)**    (f) **Taylor (3rd)**    (g) Euler-Maruyama    (h) **Itô-Taylor**

Figure 17: CelebA ($64 \times 64$) synthesis samples. $\underline{N = 4}$.

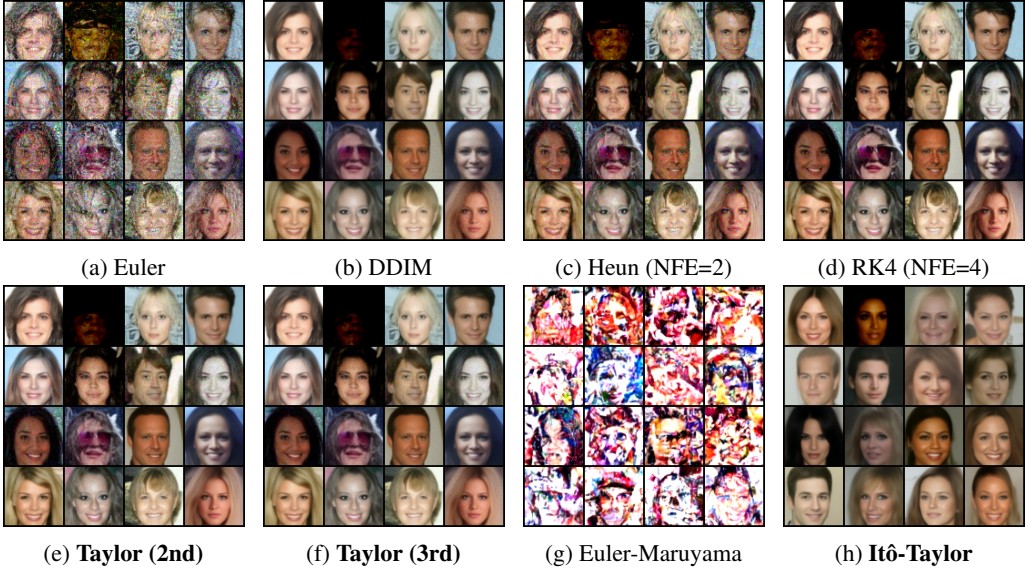

(a) Euler        (b) DDIM        (c) Heun (NFE=2)        (d) RK4 (NFE=4)

(e) **Taylor (2nd)**    (f) **Taylor (3rd)**    (g) Euler-Maruyama    (h) **Itô-Taylor**

Figure 18: CelebA ($64 \times 64$) synthesis samples. $\underline{N = 8}$.

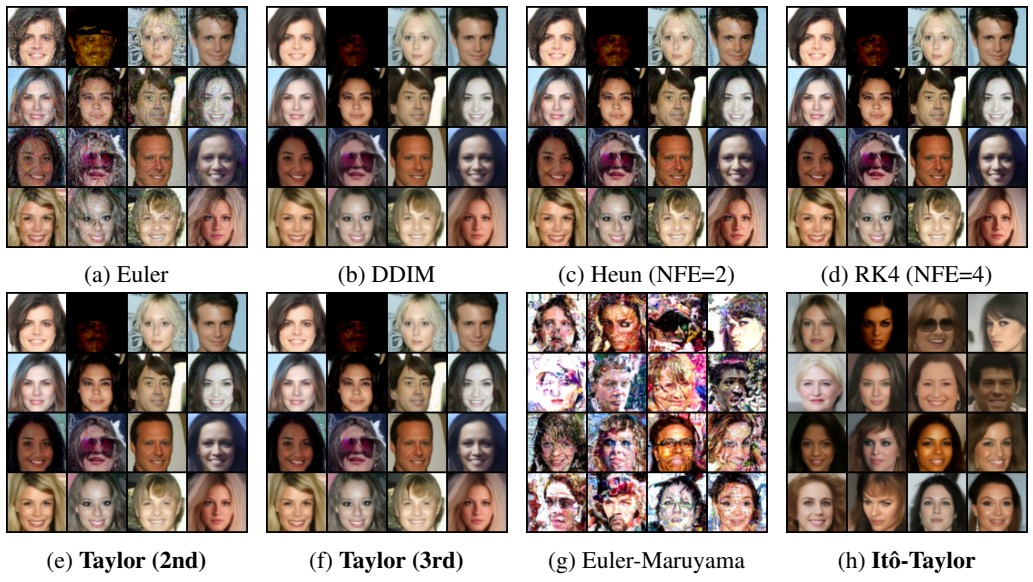

| (a) Euler | (b) DDIM | (c) Heun (NFE=2) | (d) RK4 (NFE=4) |
| (e) **Taylor (2nd)** | (f) **Taylor (3rd)** | (g) Euler-Maruyama | (h) **Itô-Taylor** |

Figure 19: CelebA ($64 \times 64$) synthesis samples. $\underline{N = 16}$.

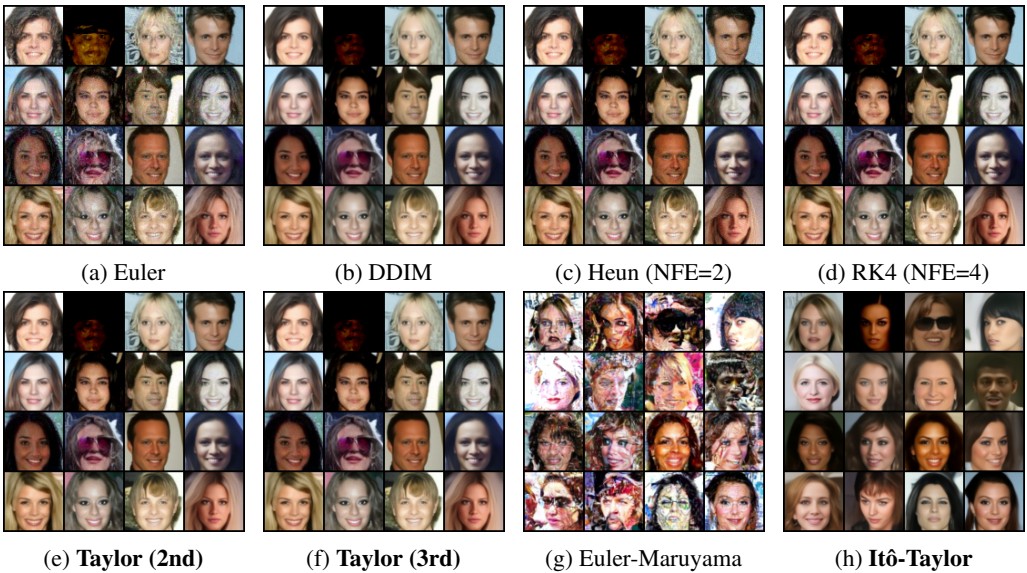

| (a) Euler | (b) DDIM | (c) Heun (NFE=2) | (d) RK4 (NFE=4) |
| (e) **Taylor (2nd)** | (f) **Taylor (3rd)** | (g) Euler-Maruyama | (h) **Itô-Taylor** |

Figure 20: CelebA ($64 \times 64$) synthesis samples. $\underline{N = 20}$.

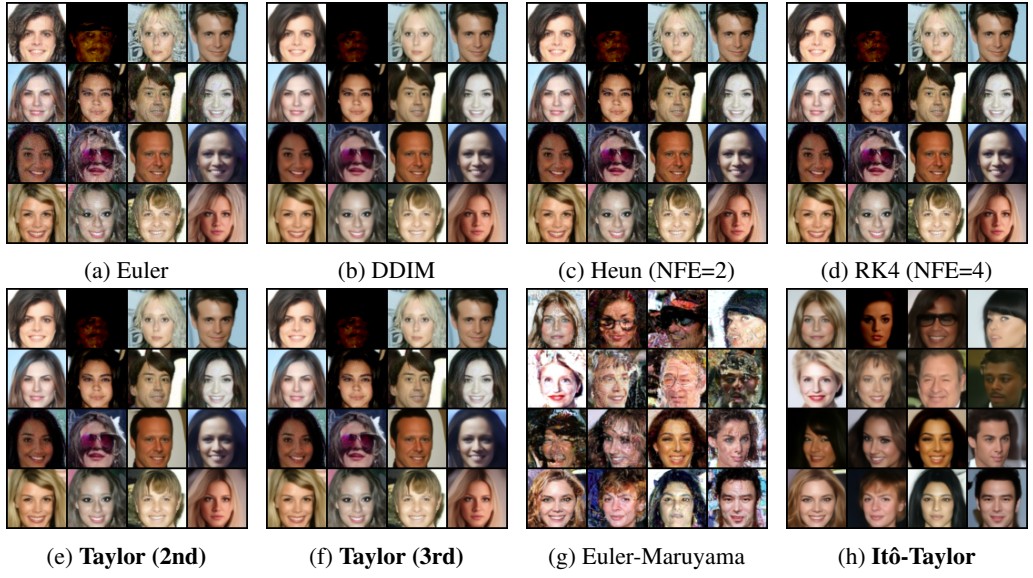

(a) Euler      (b) DDIM      (c) Heun (NFE=2)      (d) RK4 (NFE=4)

(e) **Taylor (2nd)**      (f) **Taylor (3rd)**      (g) Euler-Maruyama      (h) **Itô-Taylor**

Figure 21: CelebA ($64 \times 64$) synthesis samples. $\underline{N = 30}$.

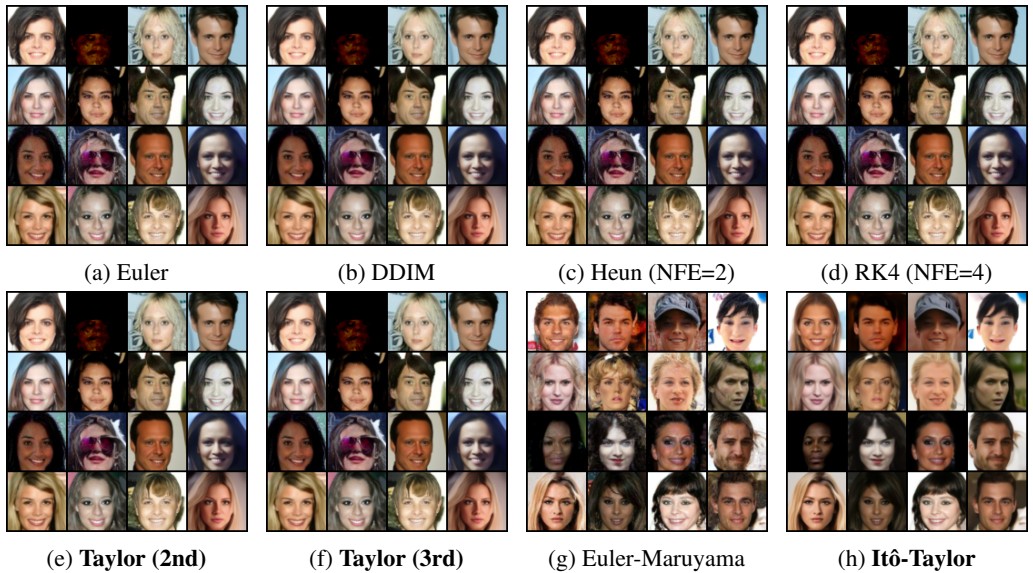

(a) Euler      (b) DDIM      (c) Heun (NFE=2)      (d) RK4 (NFE=4)

(e) **Taylor (2nd)**      (f) **Taylor (3rd)**      (g) Euler-Maruyama      (h) **Itô-Taylor**

Figure 22: CelebA ($64 \times 64$) synthesis samples. $\underline{N = 100}$.

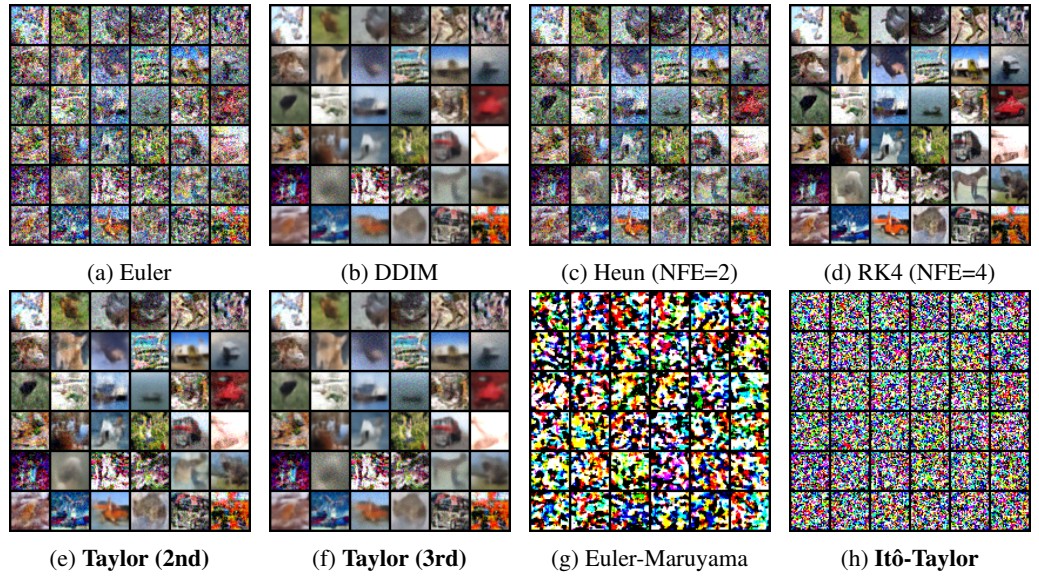

(a) Euler  (b) DDIM  (c) Heun (NFE=2)  (d) RK4 (NFE=4)

(e) **Taylor (2nd)**  (f) **Taylor (3rd)**  (g) Euler-Maruyama  (h) **Itô-Taylor**

Figure 23: CIFAR-10(32 × 32) synthesis samples. $\underline{N = 4}$.

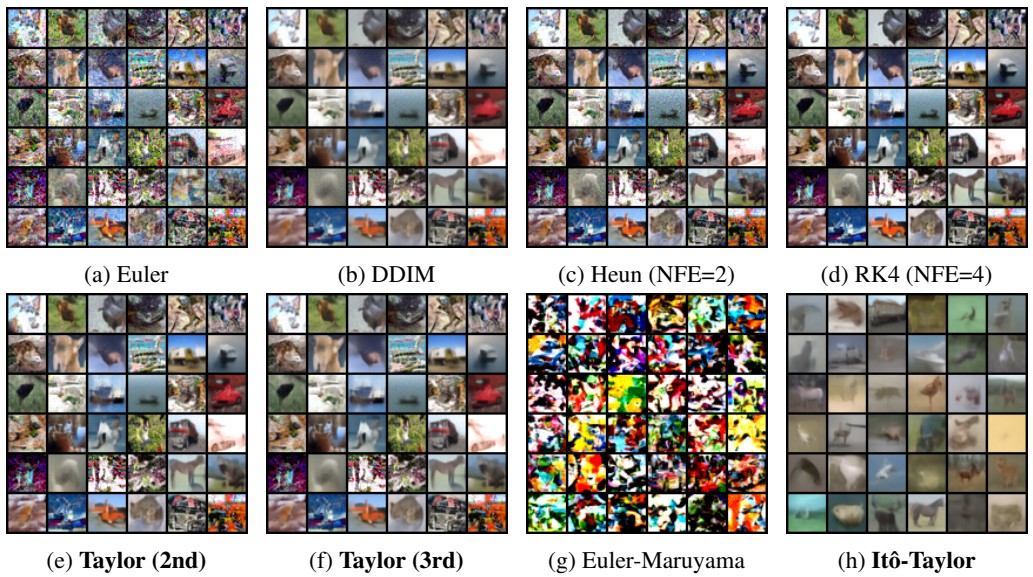

(a) Euler  (b) DDIM  (c) Heun (NFE=2)  (d) RK4 (NFE=4)

(e) **Taylor (2nd)**  (f) **Taylor (3rd)**  (g) Euler-Maruyama  (h) **Itô-Taylor**

Figure 24: CIFAR-10 (32 × 32) synthesis samples. $\underline{N = 8}$.

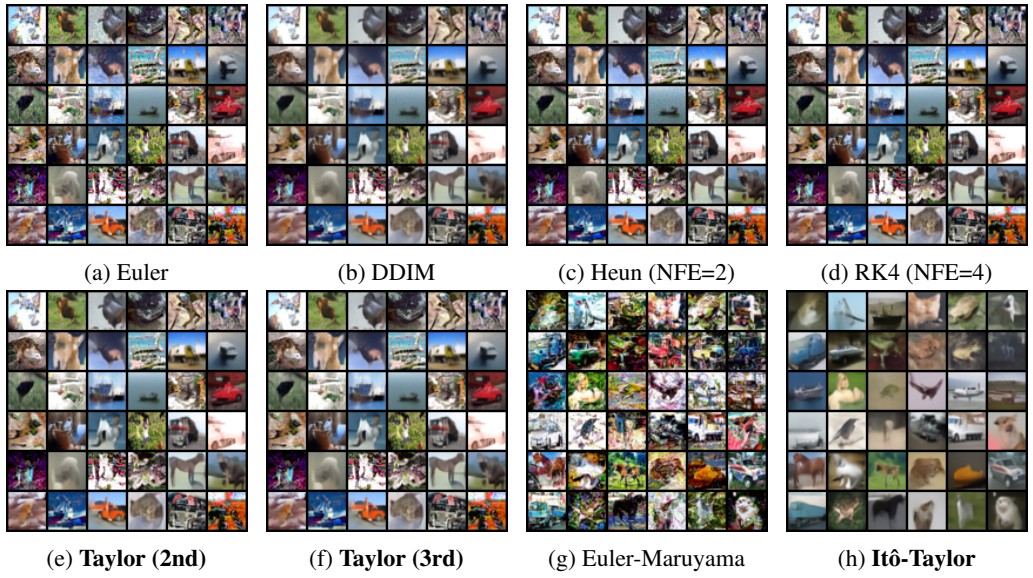

| | | |
|---|---|---|
| (a) Euler | (b) DDIM | (c) Heun (NFE=2) |

(a) Euler  (b) DDIM  (c) Heun (NFE=2)  (d) RK4 (NFE=4)

(e) **Taylor (2nd)**  (f) **Taylor (3rd)**  (g) Euler-Maruyama  (h) **Itô-Taylor**

Figure 25: CIFAR-10 ($32 \times 32$) synthesis samples. $\underline{N = 20}$.

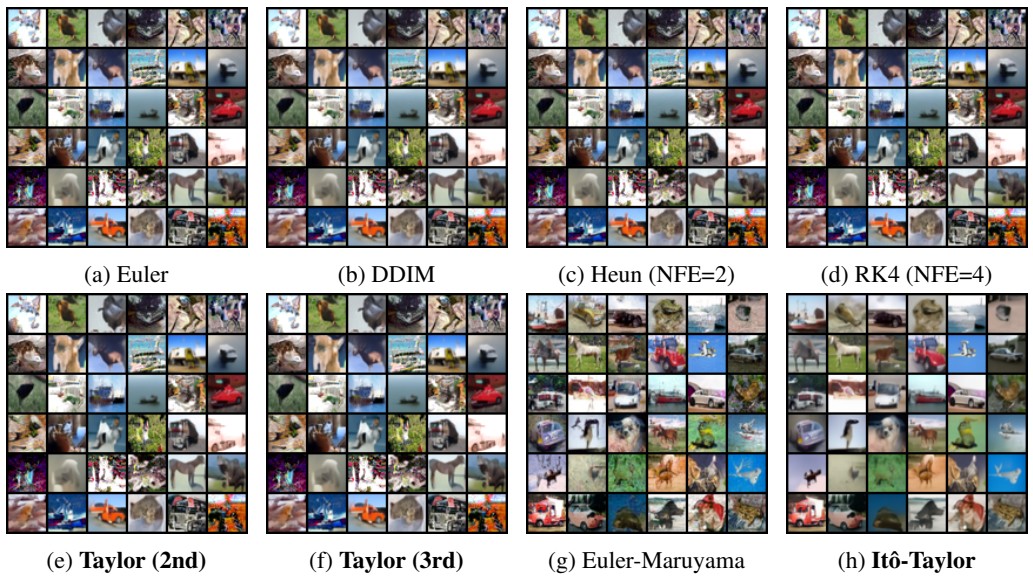

(a) Euler  (b) DDIM  (c) Heun (NFE=2)  (d) RK4 (NFE=4)

(e) **Taylor (2nd)**  (f) **Taylor (3rd)**  (g) Euler-Maruyama  (h) **Itô-Taylor**

Figure 26: CIFAR-10 ($32 \times 32$) synthesis samples. $\underline{N = 100}$.

