# OpenReview forum: "Quasi-Taylor Samplers for Diffusion Generative Models based on Ideal Derivatives"
_ICLR.cc/2023/Conference — Submitted to ICLR 2023_

### Official Review · Reviewer_Qqb3 · 2022-10-18

**Confidence:** 4
**Correctness:** 3
**Technical Novelty And Significance:** 2
**Empirical Novelty And Significance:** 2
**Recommendation:** 5

**Clarity, Quality, Novelty And Reproducibility:**

The paper is written clear. Since the idea of approximating high order gradients exists, such as [3], this work is relatively less novel.

**Strength And Weaknesses:**

Strength:

This work proposes high order methods for solving both SDE and ODE. It seems that this method can be extended to arbitrary order without much extra cost.

Weakness:

1. The approximation in Eq.(19) and Eq.(21) looks rough. Is there any bound of the approximation error?

2. It seems this method needs to use a specific noise schedule function in Eq.(28), which makes it less flexible. Would this method work on other noise schedule functions?

3. There are only experiments on CIFAR10 and CelebA 64. Besides, the used diffusion model is also weak, which has a FID around 10 on CIFAR10.

4. Missing related works on speeding up diffusion models, such as [1, 2, 3].

[1] Analytic-DPM: an Analytic Estimate of the Optimal Reverse Variance in Diffusion Probabilistic Models

[2] Estimating the Optimal Covariance with Imperfect Mean in Diffusion Probabilistic Models

[3] DPM-Solver: A Fast ODE Solver for Diffusion Probabilistic Model Sampling in Around 10 Steps


**Summary Of The Paper:**

This work proposes high order methods for solving both SDE and ODE, by approximating high order gradients in Taylar expansion. It seems that this method can be extended to arbitrary order by computing some constants. Experiments are conducted on CIFAR10 and CelebA 64.


**Summary Of The Review:**

---

> ### Author Response · Authors · 2022-11-18
> **Response to Reviewer Qqb3**
>
> Thank you for taking the time to review our paper and for your comments. We would like to respond to your comments.
>
> > The approximation in Eq.(19) and Eq.(21) looks rough. Is there any bound of the approximation error?
>
> - Approximation bound of Eq. (19) is evaluated in Appendix A. Summary is written just after Eq. (19)
> - Approximation bound of Eq. (21) has not been mathematically evaluated, but we empirically evaluated the accuracy of this approximation in the revised Appendix B.
>
> > It seems this method needs to use a specific noise schedule function in Eq.(28), which makes it less flexible. Would this method work on other noise schedule functions?
> - Now this section is removed, and the noise function is changed to the standard linear schedule. See also our general comments.
> - (Adding a mathematical condition inevitably leads to some degree of inflexibility.  While "for a solution to a differential equation to exist uniquely, the coefficients must satisfy the Lipschitz constraint" is essentially a necessary condition for any noise schedule, some of the noise schedules in use today do not satisfy this condition. Whether such a schedule would fit into this framework will depend on cases.)
>
> > There are only experiments on CIFAR10 and CelebA 64. Besides, the used diffusion model is also weak, which has a FID around 10 on CIFAR10.
>
> - Because we did not have enough computation resources to scale up our experiments, we only performed the experiments using relatively small datasets and relatively small network.
> - Of course, with the rapid expansion of real-world applications of diffusion models in the past few months and the availability of many open-source models and trained checkpoints, it could be argued that these should be used to evaluate samplers.
> - However, our intention in this paper in the first place was not to conduct such a thorough experiment and claim a slight margin of improvement, but rather to deepen our understanding of the diffusion models. Therefore, it would be regrettable if our findings are buried simply because the scale of the experiment is small.
>
>
> > Missing related works on speeding up diffusion models, such as [1, 2, 3].
>
> - Thank you for your suggestion. We cited these papers in the new version.
>
> > The paper is written clear. Since the idea of approximating high order gradients exists, such as [3], this work is relatively less novel.
> - I disagree with the argument that this paper is less novel because there are already studies using approximate calculations of higher derivatives. I think that is an argument that goes too far.
> - An important claim of our paper is that we find that the approximate higher-order derivative is obtained by simple arithmetic operations of a noise schedule and a single-step evaluation of the learned score function. And it does not require extra computation of any deep networks. Thus, our approximation method does not allow the NFE (neural function evaluations) to be larger than 1. On the other hand, in paper [3], several algorithms have multiple NFEs in one step, and there seems to be a difference in the computational complexity to obtain the same approximation of derivatives.

---

### Official Review · Reviewer_rnag · 2022-10-23

**Confidence:** 4
**Correctness:** 3
**Technical Novelty And Significance:** 3
**Empirical Novelty And Significance:** 2
**Recommendation:** 3

**Clarity, Quality, Novelty And Reproducibility:**

The presentation of the paper is quite clear (with the exception of the discussion on BKE which I found misleading as emphasized before).

The methodological contribution of the paper is quite interesting but I did not find the theory and experiments of the paper to be compelling.

The work is quite novel and I think that the use of the approximation  $\nabla \log p(x_t)$ with the conditional score $\nabla \log p(x_t|x_0)$  could be useful.

Experimental details to reproduce the introduced method are provided.

**Details Of Ethics Concerns:**

No ethical concern

**Strength And Weaknesses:**

STRENGTHS:
* One of the strenght of this paper is that it introduces a novel way to sample from diffusion models using the theoretical of higher order integrator. To the best of my knowledge this approach is novel.
* I also found the use of the approximation of the score by the conditional score to be quite interesting.
* The paper is well-written and the ODE, SDE and reverse ODE, SDE are clearly introduced.

WEAKNESSES:
* I think the authors make misleading claim regarding the Backward Kolmogorov Equation (BKE) (by the way, maybe I missed it in the main text but FPE and BKE are never defined, I just assumed that FPE was Fokker-Planck Equation and BKE was Backward Kolmogorov Equation). I could not find in [1] any reference to the backward Kolmgorov Equation but maybe I missed something here. However I disagree that the reverse-time dynamics is somehow associated with BKE. In fact FPE and BKE are dual of each other but there is no connection here with the time-reversal. The time-reversal dynamics also satisfies FPE and BKE evolutions but these are not related to the BKE of the forward process (even though one can use the BKE to establish the time-reversal SDE, see [1]).
* More important maybe, I don't really understand why the authors say that the derivatives of the learned score functions cannot be computed? It is easy to perform autodifferenciation w.r.t. the (one-dimensional) time variable and one can use vector-Jacobian-product to compute the  term $a . \nabla s_\theta$. I might be missing something but it seems that these issues can be leveraged by the careful use of automatic differenciation.
* Proposition 1 is very poorly worded. What are these "many cases"? I understand that here the authors are trying to justify their method but this is too hand-wavey. As of now the statement is too imprecise. What are the edge cases? How could we extend this proposition to hold in a more general setting?
* The deterioriation of the method noted by the author "the proposed Quasi-Taylor methods tend to give good results around $N=16,20$ and the FID deteriorates from there" is quite worrisome. This deterioriation which to me is a key limitation of the work should have been investigated in greater depth.
* The FID results provided by the authors are quite surprising. For CelebA a FID of 20 something is very large. In the original DDIM paper [2], the authors reported way better results with FID around 17 even for 10 steps and FID around 6 for 100 steps. Can the authors explain this stricking discrepancy between the announced numbers and the reported numbers in [2]?
* I think that the authors do not discuss important part of the literature dealing with acceleration of diffusion models. I understand that the authors are not going to compare themselves with knowledge distillation approaches like the one of [3] because these approaches are quite different from the one considered in that paper which is focused in better samplers but improved samplers have already been proposed in the literature, like [4,5] for instance. Comparisons with these methods are important and omitted here.

[1] Song, Sohl-Dickstein, Kingma, Kumar, Ermon, Poole - Score-Based Generative Modeling through Stochastic Differential Equations

[2] Song, Meng, Ermon - Denoising Diffusion Implicit Models

[3] Luhman, Luhman - Knowledge distillation in iterative generative models for improved sampling speed

[4] Liu, Ren, Lin, Zhao - Pseudo Numerical Methods for Diffusion Models on Manifolds

[5] Zhang, Chen - Fast Sampling of Diffusion Models with Exponential Integrator

**Summary Of The Paper:**

In this paper, the authors deal with the acceleration of denoising diffusion models. In particular, they propose improved samplers with higher order in order to reduce the number of steps requires at sampling times. The acceleration proposed in that paper is described for both Ordinary Differential Equation (ODE) and Stochastic Differential Equation (SDE) flows. It is based on the higher-order integrators [1]. In order to efficiently compute the derivatives the authors identify the score $\nabla \log p(x_t)$ with the conditional score $\nabla \log p(x_t|x_0)$ which has a tractable expression. Doing so they are able to compute efficient approximations of the derivatives. The theoretical and methodological study is complemented with experiments on CelebA 64x64 and CIFAR10.

[1] Kloeden, Platen, Schurz - Numerical Solution of SDE through computer experiments

**Summary Of The Review:**

While I think that the issue of accelerating diffusion models is interesting, I think that the paper has several important flaws from a methodological and experimental point of view.

---

> ### Author Response · Authors · 2022-11-13
> **Response to the reviwer rnag**
>
> Thank you for taking the time to carefully read and comment on our paper. I would like to make a few comments.
>
> > FPE and BKE are never defined
>
> - FPE and BKE are defined in eq (2) and (3).
> - If you meant to clarify the meaning when the acronym first appeared, we have addressed that point in the revised version.
>
> > I could not find in [1] any reference to the backward Kolmgorov Equation but maybe I missed something here. However I disagree that the reverse-time dynamics is somehow associated with BKE.
>
> - Probability-flow ODE is derived from the forward Kolmogorov equation (FPE), and the reverse time SDE is related to another FPE (R-FPE in our paper) obtained by deforming BKE using the Bayes theorem and unconditional FPE. The literature [1] may not have mentioned the latter fact, but we can derive the backward SDE from BKE. Please see (Anderson 1982, section 5) in which it is stated that the backward SDE can be derived from BKE. We added the derivation in **Appendix E.3.2** of the revised manuscript.
> - We modified the statements a little to avoid the confusion.
>
> > I don't really understand why the authors say that the derivatives of the learned score functions cannot be computed? It is easy to perform autodifferenciation w.r.t. the (one-dimensional) time variable and one can use vector-Jacobian-product to compute the term.
>
> - Yes, you are right. We have used a little too strong words. Calculating the exact derivatives of learned score function is not "intractable", but "costly and should be avoided if possible".
> - It can be calculated by numerical difference, by back-propagation, or by using JAX AutoDiff functions. But in any case, extra calculations are required. Numerical differencing requires the score function to be evaluated at least two points such as $\frac{S(x+\epsilon, t) - S(x, t)}{\|\epsilon\|}$, back-propagation requires the network to be computed in the reverse direction, and the same would be true for the Jacobian-product methods in JAX AutoDiff requiring the same or three times the cost of forward function evaluation.
> - In contrast, the strength of the proposed method lies in the fact that the "ideal derivatives" can be computed only using known values, and no extra evaluation of a deep function is required.
> - Please see the revised **section 3.4**
>
> > Proposition 1 is very poorly worded. What are these "many cases"? I understand that here the authors are trying to justify their method but this is too hand-wavey. As of now the statement is too imprecise. What are the edge cases? How could we extend this proposition to hold in a more general setting?
>
> - In fact, this may be more of a "conjecture" rather than a mathematically rigorous "proposition." We have elaborated the condition in the revised version in **Conjecture 1**
> - In Appendix A, we test theoretically and experimentally to what extent the single-point approximation holds under the following assumptions.
>   - The data space $\mathbb{R}^d$ are sufficiently high-dimensional, $d\gg 1$
>   - The data manifold $M \subset \mathbb{R}^d$ is bounded.
>   - The data manifold $M$ can be approximated locally by a Euclidean space, $1 \ll \dim M \ll d$.
>   - The variance parameter $\nu_t$ is close to 0 or 1.

---

> ### Author Response · Authors · 2022-11-18
> **Response (2) to the reviewer rnag**
>
> > The paper is well-written and the ODE, SDE and reverse ODE, SDE are clearly introduced.
>
> > The presentation of the paper is quite clear (with the exception of the discussion on BKE which I found misleading as emphasized before).
> - Thank you!
> - On the BKE, please see **Appendix E.3.2** and our another response.
>
> > The deterioriation of the method noted by the author "the proposed Quasi-Taylor methods tend to give good results around  and the FID deteriorates from there" is quite worrisome. This deterioriation which to me is a key limitation of the work should have been investigated in greater depth. The FID results provided by the authors are quite surprising. For CelebA a FID of 20 something is very large. In the original DDIM paper [2], the authors reported way better results with FID around 17 even for 10 steps and FID around 6 for 100 steps. Can the authors explain this stricking discrepancy between the announced numbers and the reported numbers in [2]?
> - As we have written in the general comment above,  There are several possible reasons.
>    - One is the possibility that there was a mistake somewhere in the FID calculation, which was carefully debugged in the revised version.
>    - Another is that the noise schedule may not have been very good, and this was replaced with an existing standard one.
>    - Another possibility is that the amount of learning was not sufficient. This is due to the limited computational resources available to us. However, we could not find any description of the computational resources used for training in the original DDIM paper, so we cannot say anything about this.
> - Now we have modified our experimental condition. Please see the revised **section 4**.
>
> > I think that the authors do not discuss important part of the literature dealing with acceleration of diffusion models. I understand that the authors are not going to compare themselves with knowledge distillation approaches like the one of [3] because these approaches are quite different from the one considered in that paper which is focused in better samplers but improved samplers have already been proposed in the literature, like [4,5] for instance. Comparisons with these methods are important and omitted here.
> > [5] Zhang, Chen - Fast Sampling of Diffusion Models with Exponential Integrator
>
> - Thank you for you suggestion. We cited these papers in the revised version.
> - It is right that comparisons should be made with existing studies, but the reference [5] seems to be a paper from the NeurIPS 2022 workshop, which has not yet been held (i.e., not yet officially published). I do not think it is a very fair to require a comparison experiment with such a paper.
> - Please do not misunderstand that the comparative experiments are lacking. That is not the case. We have conducted experiments using the DDIM and Runge-Kutta methods, which we believe have already been established as standard methods, as a baseline for comparison experiments.

---

### Official Review · Reviewer_jtaQ · 2022-10-25

**Confidence:** 4
**Correctness:** 2
**Technical Novelty And Significance:** 2
**Empirical Novelty And Significance:** 2
**Recommendation:** 3

**Clarity, Quality, Novelty And Reproducibility:**

Writing is unclear: For example,

"[DDIM] is not necessarily derived directly from PF-ODEs, and its relationship to PF-ODE was revealed through a little argumentation"

"Nevertheless, in diffusion models, the derivatives are expected to have good structure, and are effectively evaluated."

"The Jacobian matrix is diagonal assuming that each dimension is independent of each other." <--- Is this really a valid assumption?

"To date, this approximation has often been understood as a "tractable surrogate". <--- What does this mean? Citations?

"50,000 images were generated for each condition to compute the FID scores." Why 50,000? The standard is 10,000 [5].

Correctness and Novelty:
The proposed solver hinges on the use of "ideal derivatives", which replace the true higher order terms in the Taylor expansion of the ODE. While the idea of applying an idealized derivative substitution to a Taylor expansion of the diffusion ODE is novel, its correctness is unclear.
Moreover, the general derivation of the solver requires assuming that the diffusion in each data dimension is independent. This seems like a very strong (and unrealistic) assumption to me.

[5] Heusel, M., Ramsauer, H., Unterthiner, T., Nessler, B. and Hochreiter, S., 2017. Gans trained by a two time-scale update rule converge to a local nash equilibrium. Advances in neural information processing systems, 30.

**Strength And Weaknesses:**

Strengths:
* The work tackles a relevant problem in current score-based diffusion models: slow data generation.
Weaknesses:
* The empirical results in the main Figure 2 are not very compelling, compared to, e.g., [2, 3, 4].
* There is limited discussion on why the "ideal derivatives" are a suitable replacement for the true derivatives. Arguments (in the appendix) are mainly intuitive, with no proofs.
* The extra theory is dense, and it is unclear whether it is worth the empirical improvements.

[2] Salimans, T. and Ho, J., 2022. Progressive distillation for fast sampling of diffusion models. arXiv preprint arXiv:2202.00512.
[3] Song, J., Meng, C. and Ermon, S., 2020. Denoising diffusion implicit models. arXiv preprint arXiv:2010.02502.
[4] Kong, Z. and Ping, W., 2021. On fast sampling of diffusion probabilistic models. arXiv preprint arXiv:2106.00132.

**Summary Of The Paper:**

The authors propose a novel solver of the Probability Flow ODE for diffusion models introduced in [1]. By taking a Taylor expansion of the ODE and including higher-order terms, the solver can take larger steps. This speeds up sampling, which is a well-known computational bottleneck in diffusion models. However, the higher order terms in the Taylor expansion are themselves expensive to compute. Thus the authors propose substituting these terms with "ideal derivatives" which involve

[1] Y. Song, J. Sohl-Dickstein, D. P. Kingma, A. Kumar, S. Ermon, and B. Poole. Score-based generative modeling through stochastic differential equations. 2021


**Summary Of The Review:**

Overall, I find the motivation of the paper relevant. However, I find key steps of the derivation of the method problematic (i.e., the "idealized derivative" assumption, the "independent dimensions" assumption). Moreover, empirical evaluations are lacking. From Figures 2 and 4, it appears that image quality in terms of FID is not competitive compared to other faster diffusion models.

---

> ### Author Response · Authors · 2022-11-14
> **Response to the reviewer jtaQ**
>
> We appreciate your careful reading of our paper and your comments, but we disagree with some of your comments.
>
> > Writing is unclear: For example, "[DDIM] is not necessarily derived directly from PF-ODEs, and its relationship to PF-ODE was revealed through a little argumentation"
> - What we mean here is as follows: From the current way of thinking, the natural approach is to develop a solver of a given PF-ODE (which is the position of this study). However, since DDIM was developed in parallel with PF-ODE, it was not proposed as a method for solving a given PF-ODE. Nevertheless, it can be confirmed after a little mathematical argument that DDIM agrees with the Euler method for PF-ODE up to the 1st order term. So we said that a little argumentation is required to understand the connection between PF-ODE and DDIM.
>
> > "Nevertheless, in diffusion models, the derivatives are expected to have good structure, and are effectively evaluated."
>
> - The term "good structure" is used in the sense of "easy to compute and satisfies properties that are convenient for performing certain mathematical operations."
> - In this context, one of the main reasons why the Taylor method has not become a mainstream numerical method is that derivatives are not always easy to compute and are generally very complicated expressions ("bad structure"), but in our case, we argue that the coefficient of our ODE has a "good structure" such that derivatives are easily obtained though in an approximate manner.
>
> > "The Jacobian matrix is diagonal assuming that each dimension is independent of each other." <--- Is this really a valid assumption?
>
> > Moreover, the general derivation of the solver requires assuming that the diffusion in each data dimension is independent. This seems like a very strong (and unrealistic) assumption to me.
>
> - I do not think the independence assumption is unrealistic, since the neural network is predicting the Gaussian random variable with unit variance. This is logically equivalent as our assumption of single point approximation.
> - In variational Bayesian inference and many-body problems, it is a common technique to assume statistical independence of each variable, usually called the "mean-field approximation". This has the advantage that complex, almost unsolvable problems involving many variables, can be written down to a product of simpler problems, and a solution that is meaningful in practical terms can be obtained even by approximation. Not limited to this study, there are countless studies in which variables that are not inherently independent are assumed to be independent variables and solved by the mean-field approximation. Making this more rigorous is of course an important research direction, but this is a remarkably large leap forward.
>
> > "To date, this approximation has often been understood as a "tractable surrogate". <--- What does this mean? Citations?
>
> - I think it would be a common understanding in machine learning community that the Variational Bayesian loss function is used as a "surrogate" for some true training target which is intractable. The variational Bayesian logic of "being intractable to maximize the likelihood (because of the integral), ELBO is optimized instead, which is the lower bound of the likelihood," is probably universally used in many contexts. Indeed, Ho et al. states in their DDPM paper, the objective function of the diffusion model can be understood as the ELBO if some constants are dropped. I think saying "ELBO is optimized" implicitly means "this is not a true training target but a tractable surrogate".
>
> > "50,000 images were generated for each condition to compute the FID scores." Why 50,000? The standard is 10,000 [5].
> - 10,000 is the minimum requirement, and more images will improve the accuracy of FID. Even in the paper you mentioned [5] it is stated that "**We computed the (mw, Cw) on all CelebA images, while for computing (m, C)
> we used 50,000 randomly selected sample**"
> - Their official repository (https://github.com/bioinf-jku/TTUR) also says "**We recommend using a minimum sample size of 10,000 to calculate the FID otherwise the true FID of the generator is underestimated**."
> - The initial SDE paper (Song et al ICLR 2021 Score-Based Generative Modeling... ) says "**All FIDs are computed on 50k samples with tensorflow gan**". So I do not think it strange to use more than 10000 samples.

---

> ### Author Response · Authors · 2022-11-18
> **Response (2) to Reviewer jtarQ**
>
> > The empirical results in the main Figure 2 are not very compelling, compared to, e.g., [2, 3, 4].
> - We have updated our experiment in our revised version. Please see the revised **section 4**.
>
> > There is limited discussion on why the "ideal derivatives" are a suitable replacement for the true derivatives.
> - We added some discussion in **section 3.4** why our ideal derivatives are advantageous over other methods to derive derivatives.
>    - To compute finite differences, perturbations must be added and the score functions at two different points must be evaluated.
>    - Obviously, the back propagation -based methods require the evaluation of a deep function.
>    - On the other hand, in the ideal deriv method, the evaluation of the score function can be omitted (more precisely, values evaluated elsewhere can be used) and no extra evaluation of the deep neural network is required. This is an advantage.
>
> > Arguments (in the appendix) are mainly intuitive, with no proofs.
> - It is true that this paper does not take the form of "theorem--proof", and that it is based on intuitive arguments (albeit it may appear mathematically dense). However, I do not see why this is a weakness. There are plenty of papers that do not take the appearance of "theorem--proof", and it would be rather favorable that the arguments can be followed intuitively.
>
> > The extra theory is dense, and it is unclear whether it is worth the empirical improvements.
> - I am not sure which part you are referring to, but would like to comment individually.
>   - Appendix A, as you probably already pointed out, is an intuitive argument (albeit seemingly dense).
>   - Appendix B is not particularly difficult, as once we assume that the approximation in Appendix A is "equal", the rest is just calculating the derivative. The second half part is about automatic differentiation using a computer, which is also not that difficult.
>   - The former Appendix C was removed, as we do not use this noise schedule in the revised version.
>   - The former Appendix D (now Appendix E) is a summary of basic knowledge which will be useful for understanding the theoretical background of this paper and diffusion models in general. The very basic questions of what exactly it means to solve ordinary differential equations by Taylor expansion, what is the Fokker-Planck equation, and how is the Runge-Kutta method derived, are not relevant to the subject of this paper, but will be useful to readers who are not familiar with the mathematical background in this field. These are things that could have been left out, but are provided as a brief introduction to background knowledge for readers unfamiliar with these topics. I believe these are just summaries of known mathematical knowledge and not particularly esoteric.

---

### Official Review · Reviewer_hEQV · 2022-11-03

**Confidence:** 4
**Correctness:** 2
**Technical Novelty And Significance:** 2
**Empirical Novelty And Significance:** 2
**Recommendation:** 5

**Clarity, Quality, Novelty And Reproducibility:**

Although this paper considers a very interesting direction that could potentially have novelty, the clarity of the writing and the quality of the experiments need to be improved.

**Strength And Weaknesses:**

**Strength:**
1. This work considers an important direction of diffusion models: as diffusion models are known for their inefficiency in sampling, improving the sampling speed is of great interest to the community.
2. Using high-order derivates to improve the sampling speed is an interesting and promising direction.


**Weakness:**
1. The writing needs to be improved. There are many equations with parameters/notations that are not properly defined in Section 3.4. The main algorithms (Algorithms 1, 2) are not properly justified or explained with rigorous proof. The method formulation is hard to follow.
2. Proposition 1 is not rigorously stated. For instance, what does "in many cases" mean? What does an "arbitrary vector" mean? What would be the dimension of the vector? What would be the domain?
3. Although in the left figure in figure 4, the proposed approach (Taylor 2nd, 3rd) is able to achieve better performance than the baselines, the performance degrades after 20 steps. Why is that the reason? At the same time, the DDIM performance reported in figure 2, 4 is much worse than the one reported in the original DDIM paper: in the original paper, DDIM is able to achieve an FID of 13.36 on CIFAR-10, and 17.33 on CelebA 64 using 10 steps---a performance better than all of the proposed methods using 10 steps. However, in both figure 2 and figure 4, the reported performance for DDIM is much worse. My understanding is that this can be caused by using different noise scheduling. If that is the case, the comparison in figure 2 and 4 might not be fair for the baselines.
5. It seems that noise scheduling will affect the performance. How do you select the noise scheduling parameters?
6. It is hard to tell the difference between the samples from DDIM and the ones from the proposed method in Figure 1. At the same time, the samples seem to have shifted color. If it is because of not having enough sampling steps, then it would be better to visualize samples with higher quality but using more steps.

**Summary Of The Paper:**

This paper proposes an approach to improving the sample efficiency of diffusion models by incorporating high-order derivatives. As high-order derivatives are often expensive to compute, the authors propose an approximation. Empirically, the proposed approach is able to generate images using small number of sampling steps.

**Summary Of The Review:**

Given that the clarity of the paper is limited, the experiments are not very impressive or convincing, the entire paper needs to be improved for acceptance.

---

> ### Author Response · Authors · 2022-11-18
> **Response to the Reviewer hEQV**
>
> Thank you for your careful reading of our manuscript. We would like to make some comments.
>
> > The writing needs to be improved. There are many equations with parameters/notations that are not properly defined in Section 3.4.
> - We have carefully checked the symbols so that there are no undefined symbols except for the common ones. But it is possible that we are not aware of them. We would like to ask you to point out unclear symbols in specific if you found any.
>
> > The main algorithms (Algorithms 1, 2) are not properly justified or explained with rigorous proof. The method formulation is hard to follow.
> - At first glance, these algorithms may seem difficult, but as pointed out in the text, these are merely substitutions of ideal derivatives into the Taylor expansion. Since it is very hard to type LaTeX code to write out derivation (proof?) completely, we have substituted them by presenting a method to check the calculations with SymPy in **Appendix B3**. It can be checked that these equations are obtained by completing and executing the detailed parts of the SymPy code.
>
> > Proposition 1 is not rigorously stated. For instance, what does "in many cases" mean? What does an "arbitrary vector" mean? What would be the dimension of the vector? What would be the domain?
> - In the revised version we renamed this claim as "conjecture" and elaborated the conditions more in detail. Please see **Conjecture 1** in the revised text.
>
> > Although in the left figure in figure 4, the proposed approach (Taylor 2nd, 3rd) is able to achieve better performance than the baselines, the performance degrades after 20 steps. Why is that the reason?
> - There are several possible causes. For example,
>   - It could be a bug in the FID evaluation script.
>   - It may reflect the fact that increasing the number of steps increases the chance of incorporating various errors, such as approximation errors of derivatives.
> - But this phenomenon was not observed in the revised experiment.
>
> > At the same time, the DDIM performance reported in figure 2, 4 is much worse than the one reported in the original DDIM paper: in the original paper, DDIM is able to achieve an FID of 13.36 on CIFAR-10, and 17.33 on CelebA 64 using 10 steps---a performance better than all of the proposed methods using 10 steps.
> > My understanding is that this can be caused by using different noise scheduling. If that is the case, the comparison in figure 2 and 4 might not be fair for the baselines.
> - As commented in the general comments, the FID calculation method may have been incorrect, and this may have been an effect. That point has been corrected.
> - Your point that the schedule may have been adversely affected is probably true, and we changed it to the existing standard noise schedule, instead of our original one. We believe this will result in a fair comparison, with nothing new except the sampler.
> - Now, since all the experiments were conducted according to existing conditions (except for a significantly smaller number of training epochs due to computational resource constraints) and are only comparing samplers, we would like to ask for a comparison within the paper, not with other papers.
>
> > It seems that noise scheduling will affect the performance. How do you select the noise scheduling parameters?
> - As we have written in the general comment above, we have modified the noise scheduling function in the revised version. Now the schedule is the same as the existing one. But your point is right. Thank you for your comment.
>
> > It is hard to tell the difference between the samples from DDIM and the ones from the proposed method in Figure 1. At the same time, the samples seem to have shifted color.
> - As a matter of fact, it is important to note that this method very close to  DDIM. This is because, as we have written more clearly in the revised version, we can confirm that the proposed Quasi-Taylor 3rd agrees with the Taylor expansion of the DDIM coefficients at least up to 3rd order of $h$. Therefore, it is not surprising that DDIM and Quasi-Taylor produce similar results.
> - "So, is the proposed method simply a rediscovery of DDIM and not an academic novelty?" We disagree with such potential criticism. We believe that the discovery that the higher-order terms of DDIM can be derived from a completely different route than the derivation of DDIM, and that it is based on the seemingly dubious "ideal deriv approx," has significant implications for DDIM users.
> - In this paper, we also derive a stochastic solver for SDE using the logic of the ideal deriv approx. In other words, we obtain another generalization (SDE), although there is some overlap of results with existing studies. In this respect, we believe that our ideas truly expand the general knowledge on diffusion-generated models.
>
> > It would be better to visualize samples with higher quality but using more steps.
> - Please see Appendix G.

---

### Author Response · Authors · 2022-11-18
**General Response from the Authors**

Thank you all for taking the time to review our paper. We have incorporated your comments into the revised version of the paper and have submitted it again.

Important changes are noted in blue. (Note that typos and wording have been changed in places other than those in blue. Please note that we have not colored in these minor changes.)

The most important changes are as follows.

### Experiment
- Based on your comments that the experimental results were not very good, we have reviewed the experimental conditions. Specifically, we have changed the experiment to use the standard linear schedule $\beta_t = \beta_\min + (\beta_\max - \beta_\min)t$ instead of our original noise schedule ("modified tanh", in the revised version, which is inherently preferable because it satisfies the Lipschitz constraint, but it does not seem to work experimentally). As a result, there are significant changes in the experiment section (**section 4** and **Appendix G**), which you are encouraged to review. We believe that the results of the revised experiment are closer to SOTA than the previous one.
- **FID**: Corrections have been made to the FID evaluation method. In the previous experiment, there may have been a problem with the conditions used to calculate the statistics (e.g., resizing function, dynamic range conversion of images, the version of inception network, etc.). In this revision, the conditions for calculating FID have been carefully examined to ensure that there are no discrepancies in the way the statistics are calculated for the ground-truth images and the generated images.
- One point we would like to ask is that this experiment is not based on existing checkpoints, but a new full-scratch learning process, and the evaluation is being conducted before the learning process has fully converged due to constraints of our computational resources. Therefore, we would like to ask for a relative evaluation within a paper, rather than comparing absolute values of FIDs from other papers.  (Also please note that FIDs can vary due to subtle differences in implementation.)

### Accuracy of Ideal Derivatives
- We have added a section to the Appendix that directly evaluates the accuracy of our claimed ideal differential approximation. Please see **Section 3.4**, **Appendix B**

### Relationship with DDIM
- We emphasized more explicitly than the previous version that our method is equivalent to that of the DDIM at least up to 3rd order terms of $h$.
- This suggests that our derivation has approximately the same reliability as the DDIM derivation process. We have not evaluated the specific error, but it is of the order of $O(h^4)$ at most. Conversely, one could argue that "the DDIM is not reliable because its results are nearly identical to the outcome of dubious ideal derivative approximation", but of course we do not believe this to be the case.
- Please see the **Introduction**, **section 5** and **Appendix C**.

Thank you in advance.

---

### Comment · Area_Chair_RSDp · 2022-11-19
**Please respond to author feedback**

Dear reviewers,

The authors have provided their feedback. Please respond, and at least acknowledge you've read them.

Best, AC

---

### Decision · Program_Chairs · 2023-01-20

**Decision:**

Reject

**Justification For Why Not Higher Score:**

While the method could potentially be useful, reviewers found that in the paper's current form, the method is not presented in good clarity, and the experimental results are not significant.

**Justification For Why Not Lower Score:**

N/A

**Metareview: Summary, Strengths And Weaknesses:**

This work proposes high order methods for solving both SDE and ODE, by approximating high order gradients in Taylor expansion. The idea is applied to diffusion models, with experiments conducted on comparing different integrators of the backward SDE.

Reviewers agreed that the paper aims to address an important research question for diffusion models -- speeding up the generation process -- and they thought the proposed approach is novel.

On the downside, however, reviewers found the presentation of their main approach to be confusing, and they have concerns on the reported experimental results as compared with results in previous paper (e.g., the overall FID results seem to be too bad). Even beyond that, after revision with the updated results, the proposed approach are not significantly better than prior arts.

Although the paper is not good enough in its current form for acceptance at ICLR, I would encourage the authors to revise and resubmit this paper to other ML venues. Perhaps better presentations for the insights behind the ideal derivative approach, plus a check up on the experimental implementations, e.g., the FID evaluation framework, could help for the next submission.

**Summary Of Ac-Reviewer Meeting:**

N/A